# The importance of regulated resource reallocation during dynamic environmental shifts in yeast

Rachel A Kocik [1], Eli G Cytrynbaum [2], Jamie M Ahrens[1], Megan N McClean[1,2] & Audrey P Gasch [1,3]✉

## Abstract

Many organisms maintain generalized stress responses activated by adverse conditions. A common theme is the induction of stress-defense proteins with reduced production of growth-promoting proteins, including ribosomes. Yet the precise roles of these coupled programs are difficult to dissect. Here, we investigated *Saccharomyces cerevisiae* responding to salt as a model stressor. We used molecular, genomic, and single-cell microfluidic methods to examine the interplay between transient induction of stress-defense genes and coordinated repression of growth-promoting genes in the yeast environmental stress response (ESR). Loss of transcriptional inducers Msn2/4 accelerates growth during multiple mild stress doses, at the expense of acquired tolerance to subsequent severe stresses. In contrast, loss of Dot6/Tod6 repressors of growth-promoting genes delays stress acclimation, showing that gene repression accommodates the cost of the Msn2/4 response. Msn2/4 bind the *DOT6* promoter, influence Dot6 abundance and activation dynamics, and are required for full repression of Dot6 targets and other growth-promoting genes. Thus, Msn2/4 participate in regulating resource reallocation needed to induce their transcripts, underscoring a common theme in stress responses utilized in other organisms.

**Keywords** Stress; Resource Allocation; Regulatory Dynamics
**Subject Categories** Chromatin, Transcription & Genomics; Evolution & Ecology

## Introduction

Cells have evolved intricate systems to allocate limited intracellular resources according to the demands of an often-fluctuating environment. When conditions are favorable, many microbes maximize their growth rate by directing resources toward growth and proliferation. Much of the transcriptional and translational capacity goes toward producing ribosomes, which under optimal conditions fuel rapid growth (Scott and Hwa, 2011, 2023; Warner, 1999). However, in suboptimal conditions, especially in response to acute stress, resources including transcriptional and translational capacity are reallocated toward survival, often at the expense of growth and growth-promoting processes. In fact, rapid growth and high stress tolerance represent a well-known tradeoff: fast-growing cells directing resources toward division are often the most sensitive to stress, whereas slow growing cells are typically highly stress tolerant (Balaban et al, 2004; Basu et al, 2022; Levy et al, 2012; Pontes and Groisman, 2019; Zakrzewska et al, 2011; Zhang et al, 2020). This is true across organisms, including bacteria, yeast, plants, and mammalian cells. However, it remains poorly understood how cells regulate changes in resource allocation during times of stress and which cellular objectives (maximizing growth versus high stress tolerance) dictate those changes. This is important for understanding how cells thrive in natural environments that are dynamic and often suboptimal. Presumably, cells must coordinate multiple facets of physiology as they respond and acclimate to changing conditions.

Budding yeast *Saccharomyces cerevisiae* has been an excellent model to understand principles of growth-versus-defense responses. Upon an acute shift to suboptimal conditions, yeast activate condition-specific responses customized for each condition, along with a common transcriptomic response known as the environmental stress response (ESR) (Causton et al, 2001; Gasch et al, 2000). The ESR is activated in response to diverse types of stress, including nutrient limitation, shifts in environmental conditions like osmolarity or temperature, and exposure to toxic compounds. The program includes ~300 transcriptionally induced genes (iESR genes) involved in wide-ranging defense processes such as redox balance, protein folding and degradation, carbohydrate and energy metabolism, trehalose and glycogen biosynthesis, and other processes (Gasch, 2002a; Gasch et al, 2000). Induced transcription of iESR genes is coordinated with reduced expression of ~600 genes (rESR genes) that encode ribosomal proteins (RP) and proteins involved in ribosome biogenesis, translation, and other growth-promoting processes (RiBi genes) (Gasch, 2002a; Gasch et al, 2000). In optimal conditions, cells devote significant resources to transcribing and translating rESR transcripts, which are required for rapid growth, while maintaining low production of iESR and defense proteins (Fig. 1A). Notably, other organisms maintain analogous, if not orthologous, responses to balance stress-defense versus growth-promoting processes, including the Integrated Stress Response (ISR) in mammals (Costa-Mattioli and Walter, 2020; Harding et al, 2003) and the Stringent/SOS responses in bacteria (Gourse et al, 2018; Irving et al, 2021). Despite differences in the regulation of these programs across species, many of the underlying themes are conserved, including the redirection

---

[1]Center for Genomic Science Innovation, University of Wisconsin-Madison, Madison, WI 53706, USA. [2]Department of Biomedical Engineering, University of Wisconsin-Madison, Madison, WI 53706, USA. [3]Department of Medical Genetics, University of Wisconsin-Madison, Madison, WI 53706, USA. ✉E-mail: agasch@wisc.edu

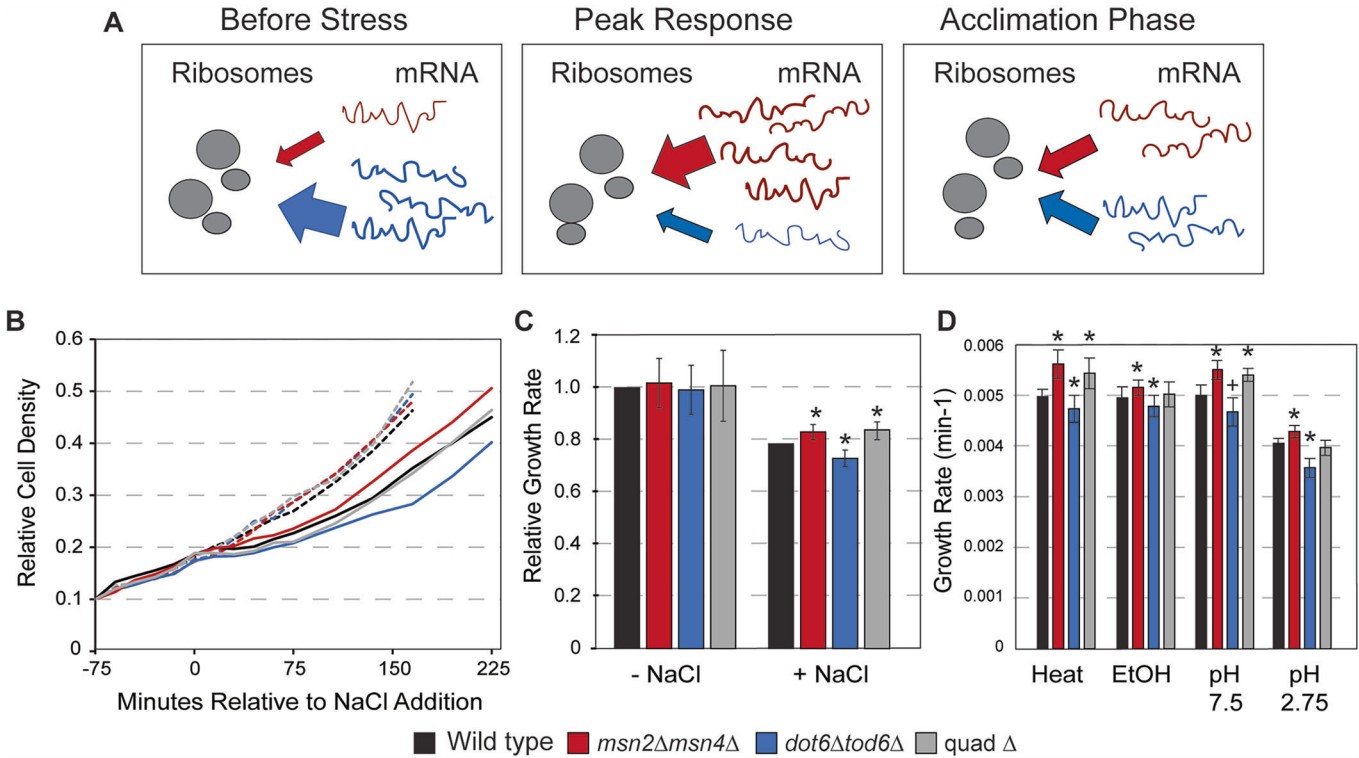

**Figure 1. Msn2/4 versus Dot6/Tod6 have opposing influences on post-stress growth rate.**

(A) A model for the dynamics of stress-defense mRNAs induced by stress (red) versus growth-promoting mRNAs highly expressed in optimal conditions but transiently repressed during stress acclimation (blue). (B) Representative relative cell density ($OD_{600}$) for wild-type (black), *msn2Δmsn4Δ* (red), *dot6Δtod6Δ* (blue), and quadruple mutant ("quadΔ") (gray) growing in the absence (dashed lines) and presence (solid lines) of 0.7 M NaCl added at 0 min. (C) Average and standard deviation ($n = 7$) of growth rates in the absence (left) and presence (right) of 0.7 M NaCl (calculated from 75 to 225 min timepoints). Post NaCl growth rates were calculated relative to their paired wild-type, then scaled to the average wild-type post-stress versus pre-stress relative rate. Exact *P* values from replicate-paired, two-sided *T* tests are listed in Dataset EV1. (D) Average and standard deviation ($n = 6$) of growth rates after 30–40 °C heat shock, 3% ethanol, pH 7.5 or 2.5 shift. *P* value < 0.01, +P value = 0.07, two-tailed, replicate-paired *t* test relative to the corresponding treated wild-type. Exact *P* values are listed in Dataset EV1. Source data are available online for this figure.

of cellular resources away from growth-promoting processes and toward stress-induced transcripts. Yet decoupling the role of translational suppression from the functions of induced proteins has remained challenging.

iESR induction and rESR repression are highly correlated in bulk cultures, at least in part due to coordinated regulation (Gasch, 2002a). iESR gene induction is partially orchestrated by the paralogous "general stress" transcription factors Msn2 and Msn4 (Causton et al, 2001; Estruch and Carlson, 1993; Gasch et al, 2000), overlaid with condition-specific factors that customize expression levels in specific environments (Gasch, 2002a). Msn2 and/or Msn4 (Msn2/4) bind the stress response promoter element (STRE, CCCCT) present in one to many copies upstream of hundreds of target genes (Martínez-Pastor et al, 1996; Stewart-Ornstein et al, 2013). Like many stress-regulated factors, Msn2/4 are regulated by nuclear translocation: phosphorylation at specific residues, including by growth-promoting Protein Kinase A (PKA) and TOR kinases, restricts the factors to the cytoplasm during optimal conditions (Beck and Hall, 1999; Boy-Marcotte et al, 1998; Görner et al, 1998; Jacquet et al, 2003; Smith et al, 1998). During stress, dephosphorylation of those residues coupled with other activating mechanisms prompts Msn2/4 nuclear localization and gene induction (De Wever et al, 2005; Garreau et al, 2000; González et al, 2009; Görner et al, 1998; Lenssen et al, 2005;

Santhanam et al, 2004; Smith et al, 1998). The dynamics of nuclear translocation can impart important differences in gene expression, depending on the amount of protein that enters the nucleus, the duration of nuclear accumulation, and gene-promoter architecture including the number of upstream STRE elements of each gene (Gasch, 2002a; Hansen and O'Shea, 2013, 2015a, 2015b, 2016; Hansen and Zechner, 2021; Hao and O'Shea, 2012; Purvis et al, 2012; Purvis and Lahav, 2013; Stewart-Ornstein et al, 2013; Sweeney and McClean, 2023). rESR subgroups are modulated by different regulators (Gasch, 2002a; Jorgensen et al, 2004; Marion et al, 2004; Schawalder et al, 2004; Shore and Nasmyth, 1987). Many of the RiBi genes are repressed during stress by Dot6 and its paralog Tod6, each of which can dominate the response depending on the conditions (Bergenholm et al, 2018; Cheng and Brar, 2019; Lippman and Broach, 2009). Dot6 and/or Tod6 (Dot6/Tod6) bind the GATGAG motif, which is present in about ~70% of RiBi gene promoters (Badis et al, 2008; Zhu et al, 2009). Furthermore, Dot6/Tod6 are also regulated by nuclear translocation, where phosphorylation by PKA and Sch9/TOR maintains the factors in the cytoplasm and dephosphorylation leads to nuclear accumulation (Huber et al, 2011; Lippman and Broach, 2009).

During many stress responses, ESR activation, and in particular rESR repression, coincides with growth reduction; however, the

ESR is not an indirect response to growth as previously proposed (see (Brauer et al, 2008; Castrillo et al, 2007; Lee et al, 2011; Lu et al, 2009; O'Duibhir et al, 2014; Regenberg et al, 2006)). Cells already arrested in division and with reduced biomass production still show rESR repression upon stress exposure (Ho et al, 2018). We proposed that rESR repression during acute stress helps to redirect transcriptional and translational capacity toward induced mRNAs (Bergen et al, 2022; Ho et al, 2018; Lee et al, 2011). Cells lacking *DOT6/TOD6* fail to fully repress rESR genes during acute salt stress, leading to the over-abundance of rESR transcripts that remain associated with ribosomes. In turn, iESR transcripts show reduced ribosome binding and delayed production of their proteins (Ho et al, 2018; Lee et al, 2011). Indeed, the *dot6Δtod6Δ* mutant also shows delayed synthesis of iESR protein Ctt1 encoding cytosolic catalase, despite higher induction of *CTT1* mRNA (Bergen et al, 2022; Ho et al, 2018). Thus, repression of the rESR genes may indirectly influence the production of induced proteins simply by decreasing rESR mRNAs from the translating pool.

Studying cell cultures in bulk can obscure causal relationships that vary across individual cells in a population. Thus, investigating cell-to-cell heterogeneity has been a useful tool in deciphering co-varying phenotypes that can reflect on cellular coordination (Bagamery et al, 2020; Barber et al, 2021; Gasch et al, 2017; Levy et al, 2012; Li et al, 2018). We previously used microfluidic live-cell imaging to study single-cell heterogeneity in cells responding to an acute dose of sodium chloride (NaCl) stress (Bergen et al, 2022). Our system enabled characterization of multiple phenotypes in single cells, including growth rate, colony size, cell-cycle phase, and nuclear-translocation dynamics of fluorescently tagged Msn2-mCherry and Dot6-GFP expressed in the same cells. Somewhat counterintuitively, we found that wild-type cells with larger Dot6 nuclear translocation response, predicted to produce stronger repression of growth-promoting rESR genes, in fact acclimate with faster post-stress growth rates following acute salt stress. Wild-type cells with stronger Dot6 activation also displayed faster production of Ctt1 protein compared to cells with weaker activation, consistent with our model that rESR repression helps to accelerate production of induced proteins (Bergen et al, 2022). We proposed that Dot6-dependent transcriptional repression, and by extension repression of the rESR as a whole, is important to reallocate resources for faster acclimation to stress conditions. But if and how a faster response is important for stress survival has not been tested. Furthermore, how resource reallocation through gene repression is coordinated with genes induced in the ESR was not clear from past work.

Here, we investigated the interplay between Msn2/4 and Dot6/Tod6 activation dynamics, transcriptional regulation, and growth versus defense objectives. We show that activation of the Msn2/4 response comes at a significant cost during a single-stress treatment, in a manner that is accommodated by Dot6/Tod6-dependent repression. Yet both programs are required for normal acquisition of subsequent stress tolerance when cells are shifted between two stressful conditions. By analyzing our own and previously published datasets, we show that Msn2/4 contribute to the regulation of Dot6 and many other genes repressed in the rESR, demonstrating that Msn2/4 help manage the resource reallocation needed for their own response. We discuss reminiscent programs in the stress responses of other species, even when the precise mechanisms have evolved.

# Results

## Msn2/4 versus Dot6/Tod6 have opposing influences on growth rate after a single-stress treatment

We set out to test the separable effects of iESR and rESR expression changes by studying cells lacking their respective regulators. We previously showed that cells lacking Dot6 and Tod6 (*dot6Δtod6Δ*) grow indistinguishably from wild-type in the absence of stress but display slower growth after salt stress (Bergen et al, 2022). We confirmed here that *dot6Δtod6Δ* cells show a similar lag phase but reduced growth rate compared to wild-type cells only after stress treatment ($P = 0.002$, replicate-paired $T$ test (Fig. 1B,C). The reduced post-stress growth rate was also seen for other stresses, including heat shock, ethanol, basic or acid pH treatments (Fig. 1D), and was previously reported after nitrogen and glucose deprivation (Lippman and Broach, 2009). A remaining question was the role of Msn2/4 and iESR induction on stress acclimation. Like the *dot6Δtod6Δ* mutant, *msn2Δmsn4Δ* cells grew indistinguishably from wild-type in the absence of stress (Fig. 1B). Surprisingly, however, the *msn2Δmsn4Δ* mutant grew significantly faster than wild-type cells after salt stress ($P = 0.008$) and with shorter lag, since they had a significantly greater percent change in cell density over 60 min compared to the wild-type ($P = 0.010$, replicate and time-paired two-tailed $T$ test). The growth rate trend was reproducible across other stresses as well (Fig. 1D). Thus, mounting the iESR comes with a significant cost to post-stress growth rate after a single-stress treatment.

We reasoned that if Dot6/Tod6-dependent repression serves to release resources for induced protein production, then loss of the costly Msn2/4 response may recover growth rate in *dot6Δtod6Δ* cells. Alternatively, if Dot6/Tod6 plays a different unrecognized role, deletion of *MSN2/4* would not alleviate its post-stress growth requirement. To test this, we generated a strain lacking all four transcription factors (referred to as the quad mutant or *quadΔ*). Like both double mutants, the quad mutant grew similarly to wild-type in the absence of stress. However, during 75–225 min post-salt addition, the quad mutant grew faster than the wild-type, similar to cells lacking Msn2/4. The result held for other stresses as well, where the quad mutant recovered with post-stress growth that was equal to or better than the wild-type strain. Thus, the reduced post-stress growth rate of *dot6Δtod6Δ* cells acclimating to stress can be complemented by loss of Msn2/4, suggesting that the cost of iESR induction explains the fitness defect when RiBi gene repression is lost.

## Both Msn2/4 and Dot6/Tod6 responses benefit future-stress survival

The cost of Msn2/4 activity to post-stress growth rate raised questions about why cells would maintain this response. One explanation is acquired stress resistance, in which cells that mount a stress response during a mild dose of one stress can survive what would otherwise be a lethal dose of subsequent stress treatment (Berry and Gasch, 2008). Past work from our lab showed that Msn2/4 are essential for acquired resistance to severe peroxide stress after salt-stress pretreatment (Berry et al, 2011; Berry and Gasch, 2008), which we confirmed here. At varying times before and after NaCl treatment, an aliquot of culture was removed and

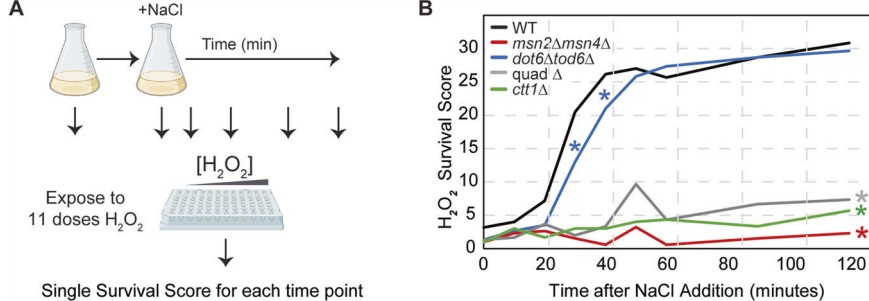

**Figure 2. Msn2/4 and Dot6/Tod6 responses are important for acquired stress resistance.**

(A) Schematic of acquired stress resistance protocol, see text. (B) The average change in $H_2O_2$ survival scores for wild-type (black), $msn2\Delta msn4\Delta$ (red), $dot6\Delta tod6\Delta$ (blue), quad Δ (gray), and $ctt1\Delta$ (green) cells. $n = 3$ replicates except for $dot6\Delta tod6\Delta$ where $n = 6$. *P value < 0.03 (*), one-tailed, replicate-paired $t$ test at each timepoint (see Fig. EV1 for paired datasets used in statistics). P values for $dot6\Delta tod6\Delta$ versus WT: 0.030 (30 min), 0.0186 (40 min). The $msn2\Delta msn4\Delta$, $quad\Delta$, and $ctt1\Delta$ mutants were highly significant at all time points after 20 min (represented by a single asterisk at the end of the curve, see Dataset EV3 for exact P values). Source data are available online for this figure.

cells were exposed to a panel of $H_2O_2$ doses for 2 h, after which colony forming units were assessed (Fig. 2A). The relative viability at each dose was normalized to the side-by-side treated wild-type, and a single $H_2O_2$ survival score was calculated as the sum of those scores across doses (see "Methods", Fig EV1A–H). As expected, cells lacking *MSN2/4* had a major defect acquiring peroxide tolerance after salt treatment, as did the quad mutant that also lacks *MSN2/4* (Fig. 2B). As shown previously, acquisition of peroxide tolerance was also dependent on Msn2/4 target *CTT1* encoding cytosolic catalase, since a strain lacking *CTT1* acquired little tolerance (Berry and Gasch, 2008; Guan et al, 2012) (Fig. 2B). Thus, the Msn2/4 response, at least in part via Ctt1, is essential for acquired peroxide tolerance under these conditions.

The role of Dot6/Tod6 and rESR repression in acquired stress resistance had not been previously investigated. Here, we found that Dot6/Tod6 are also required for normal acquisition of peroxide treatment. While the $dot6\Delta tod6\Delta$ mutant acquired wild-type levels of hydrogen peroxide resistance after exposure to salt, they did so with a significant delay (Fig. 2B). Wild-type cells acquired maximal resistance by ~40 min; however, the $dot6\Delta tod6\Delta$ cells took over ~60 min to reach maximal tolerance. This delayed acquisition of peroxide tolerance cannot be explained by the reduced growth rate of the mutant, which was observed at later time points (see Fig. 1). Instead, the timing of the delay correlates with delayed Ctt1 protein production in the $dot6\Delta tod6\Delta$ mutant (Bergen et al, 2022; Ho et al, 2018). These results are consistent with the model that Dot6/Tod6 gene repression helps to accelerate production of stress-defense proteins needed for acquired stress resistance.

## Interplay between Msn2 and Dot6 activation dynamics

While response couplings are often difficult to identify in bulk cultures, co-varying phenotypes become apparent when scoring single-cell heterogeneity within a population. We previously developed a microfluidics assay to explore heterogeneity in the nuclear translocation dynamics of Msn2-mCherry and Dot6-GFP expressed in the same cells (Bergen et al, 2022). In that study, we found that wild-type cells with a larger peak in Dot6 nuclear accumulation acclimated with faster post-stress growth rates than

cells with a smaller peak. Here, we characterized how each ESR response impacted activation of the counterpart ESR regulators.

To explore this, we generated mutants in which one or the other set of paralogous transcription factors was deleted. Msn2 and Dot6 represent the primary paralogs during NaCl stress (Berry and Gasch, 2008) (Figs. EV2A–D and EV3A–F). In particular, Tod6 showed only weak nuclear translocation during NaCl stress (Fig. EV2) and had little contribution to RiBi gene repression on its own (Fig. EV3). Thus, we generated one strain expressing genomically integrated Msn2-mCherry in the absence of *DOT6/TOD6*, while another strain expressed integrated Dot6-GFP in the absence of *MSN2/4*. We compared the response of Dot6-GFP or Msn2-mCherry in each mutant to wild-type cells that expressed both Msn2-mCherry and Dot6-GFP, as well as a third constitutive fluorescence marker (Nhp6a-iRFP, see "Methods"). This enabled mixing each mutant with the wild-type and distinguishing strains based on Nhp6a-iRFP, providing a sensitive comparison of strain behaviors in the same microfluidics chamber.

Loss of *DOT6/TOD6* did not appreciably influence Msn2-mCherry nuclear dynamics (Fig. 3A–C). Mutant and wild-type cells displayed similar distributions of Msn2 localization dynamics and Msn2-mCherry localization peak heights (Fig. 3B,C). Furthermore, clustering of the individual cells based on Msn2 dynamics revealed that the two cell types cluster together and are not distinguishable by gross differences in Msn2 behavior (Fig. 3A). We conclude that the presence of Dot6 does not significantly impact the behavior of Msn2.

In contrast, loss of Msn2/4 had a major impact on Dot6 behavior. Clustering cells based on Dot6-GFP translocation dynamics clearly delineated cell types: many of the $msn2\Delta msn4\Delta$ cells had a weaker Dot6-GFP nuclear translocation response that was well below the median of all cells in the analysis, causing many of the $msn2\Delta msn4\Delta$ cells to fall in a separate cluster (Fig. 4A). The weaker response can also be seen in the distributions of nuclear translocation dynamics, where cells lacking *MSN2/4* displayed a lower Dot6 translocation response than the wild-type (Figs. 4B and EV4A,B). We noticed that, beyond the translocation dynamics during acute stress, cells lacking *MSN2/4* showed significantly lower levels of Dot6-GFP signal overall, both before and after stress (Fig. 4C). This cannot be explained by differences in cell size (which could change signal intensity over a changing area), since

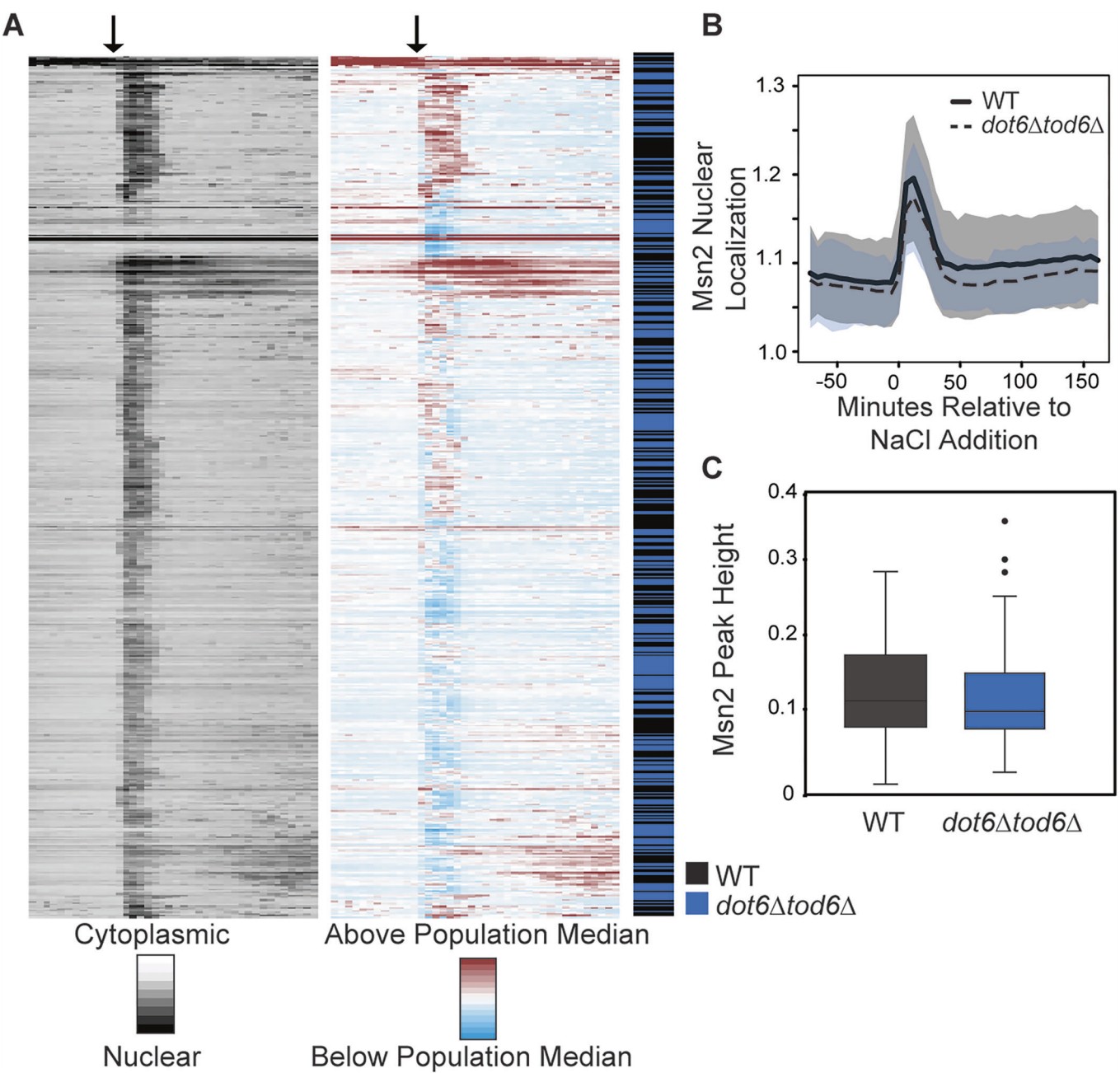

**Figure 3. Msn2 behavior is not affected by a loss of Dot6 and Tod6.**

(A) Left: Nuclear/cytoplasmic ratio was plotted for $n = 559$ cells (wt, 272; $dot6\Delta tod6\Delta$, 287) as rows across timepoints (columns) before and after salt addition, indicated by the arrow. Right: the same data shown on the left normalized to the median of each column (population median). Cells were hierarchically clustered based on population-centered Msn2 nuclear translocation dynamics, after which cell identity was mapped onto the figure, indicating wild-type (black) or $dot6\Delta tod6\Delta$ (blue) cells. (B) The average Msn2 nuclear/cytoplasmic ratio in wild-type (black line) and $dot6\Delta tod6\Delta$ (dashed line) cells +/− one standard deviation. (C) Distribution of Msn2 acute stress peak heights for wild-type (black) and $dot6\Delta tod6\Delta$ (blue) cells from (A). $P = 0.2$, Wilcoxon rank-sum test. Boxplots show median (line) and 0.25 and 0.75 quartiles (box), with whiskers extending from minimum to maximum excluding outliers (circles) that are <0.25 quartile–1.5× interquartile range or >0.75 quartile–1.5× interquartile range. Source data are available online for this figure.

$msn2\Delta msn4\Delta$ cell size is indistinguishable from wild-type both before or after salt stress ($P = 0.7$ and 0.2, respectively, Wilcoxon rank-sum test). The $msn2\Delta msn4\Delta$ mutant showed no difference in Tod6-GFP levels or activation, which remained weak (Fig. EV2). Thus, cells lacking $MSN2/4$ have less Dot6-GFP.

One question is if Msn2/4 has a direct impact on Dot6 levels/response or if the effect is indirect, perhaps simply due to the loss of iESR induction. We turned to past Msn2 chromatin-immunoprecipitation data to investigate. Remarkably, both Msn2 and Msn4 bind the Dot6 promoter after multiple stress conditions

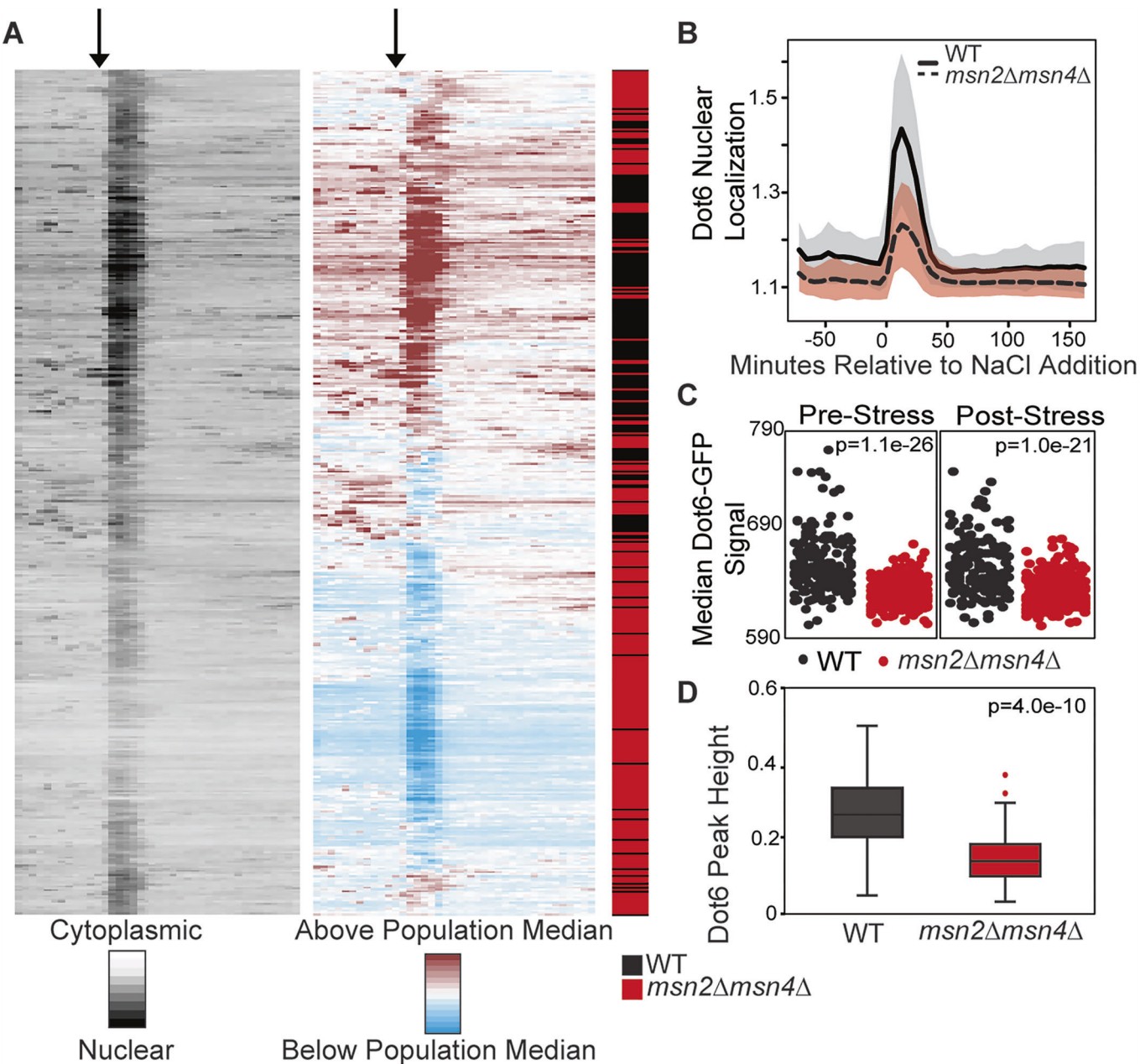

**Figure 4. Loss of *MSN2/4* leads to decreased Dot6 abundance and nuclear localization.**

(A) Wild-type (WT) and *msn2Δmsn4Δ* cells from 4 replicates were clustered based on population-centered Dot6 nuclear translocation dynamics, as described in Fig. 3. (B) The population average of Dot6 nuclear/cytoplasmic ratio in wild-type (black line) and *msn2Δmsn4Δ* (dashed line) cells +/− one standard deviation. (C) Distribution of median Dot6-GFP signal scored before (0-72 min) or after (120–216 min) NaCl treatment (see "Methods" for details) for WT and *msn2Δmsn4Δ* cells; p, Wilcoxon rank-sum test. (D) Distribution of Dot6 acute-stress peak heights across 147 cells (WT, 43, *msn2Δmsn4Δ*, 104) with similar Dot6-GFP levels, see "Methods". $P = 2.2e{-}10$, Wilcoxon rank-sum test. Boxplots show median (line) and 0.25 and 0.75 quartiles (box), with whiskers extending from minimum to maximum excluding outliers (circles) that are <0.25 quartile–1.5× interquartile range or >0.75 quartile–1.5× interquartile range. Source data are available online for this figure.

in ours and other studies (Brodsky et al, 2020; Elfving et al, 2014; Huebert et al, 2012; Kuang et al, 2017; Ni et al, 2009) (Fig. EV5A,B). The Dot6 promoter harbors one perfect-match (CCCCT) to the Msn2 binding site at -580 bp and multiple other similar C-rich sequences within ~600 bp upstream (Brodsky et al, 2020; Elfving et al, 2014), although Msn2 can localize to the promoter through protein interactions, without its binding domain (Brodsky et al, 2020; Mindel et al, 2024). These results suggest a direct conduit

between Msn2 that contributes to iESR induction and regulation of Dot6 that participates in rESR repression. It is also possible that loss of *MSN2/MSN4* affects Dot6 protein levels indirectly, perhaps through degradation (Kusama et al, 2022).

We wondered if the apparent weaker nuclear translocation of Dot6-GFP in this mutant is artifactually influenced by having less GFP signal (see "Methods"). To test this, we investigated a subset of *msn2Δmsn4Δ* and wild-type cells who's starting Dot6-GFP abundance

was in the same range. Across 147 wild-type and *msn2Δmsn4Δ* cells with indistinguishable levels of Dot6-GFP (*P* = 0.5, Wilcoxon test, see "Methods"), the *msn2Δmsn4Δ* cells displayed significantly smaller Dot6-GFP nuclear translocation peak heights (Fig. 4D, *P* = 2.2e-10, Wilcoxon test). Thus, cells lacking *MSN2/4* have lower Dot6 abundance overall, possibly through both direct regulation of *DOT6* transcription and other indirect effects, and weaker Dot6 activation during salt stress, likely indirectly.

## Msn2/4 influences both iESR induction and rESR repression

To further explore the interplay between Msn2/4 and Dot6/Tod6, we followed dynamic changes to the transcriptome before and in ten-minute increments after salt stress in the different strains. We first defined RiBi targets under our growth conditions, identifying 489 genes whose response to salt treatment was altered in the *dot6Δtod6Δ* strain compared to wild-type in at least two timepoints (FDR < 0.05), Fig. EV3A, see "Methods"). Consistent with prior results (Ho et al, 2018; Huber et al, 2011; Kunkel et al, 2019; Kusama et al, 2022; Lippman and Broach, 2009), 82% of the affected genes showed a repression defect, and 82% of those harbored matches to the known Dot6/Tod6 consensus 'GATGAG' within 500 bp upstream (*P* = 3.3e-58 hypergeometric test). The remaining 18% of differentially expressed genes were induced by salt, but these generally showed subtle differences (and in some cases greater induction) compared to wild-type. As expected based on the microscopy data, deletion of *TOD6* alone had little effect on expression, whereas deletion of *DOT6* produced a repression defect that was exacerbated in the absence of both regulators (Fig. EV3B,C).

In contrast to *DOT6/TOD6*, loss of *MSN2/4* produced broader expression effects. We partitioned 1306 genes significantly affected by *MSN2/4* deletion (FDR < 0.05, see "Methods") into 10 groups by k-means clustering (Fig. 5A,B). Three of the clusters showed little expression change in wild-type cells but were weakly induced in the absence of *MSN2/4* (Fig. 5A,B, Clusters h-j), likely due to indirect effects. Three other clusters (a-c) were induced in wild-type cells but at reduced levels in the *msn2Δmsn4Δ* strain (Fig. 5A-C, point 1). As expected, these clusters were heavily enriched for genes with Msn2/4 binding elements within 500 bp upstream of the gene ('STRE', CCCCT) and for genes whose promoters are physically bound by Msn2 and/or Msn4 in response to multiple stresses including NaCl, as summarized in the Yeastract database (Elfving et al, 2014; Huebert et al, 2012; Kuang et al, 2017; Ni et al, 2009; Teixeira et al, 2023). The clusters were distinguished from one another by subtle differences in expression patterns that correlated with regulatory architecture. For example, in addition to Msn2/4 targets, Cluster a was enriched for targets of other environment-responsive factors (many of which regulate only subsets of these genes, Dataset EV2) (Gasch, 2002a; Gasch et al, 2000). Cluster b was enriched for metabolic genes involved in respiration and carbon metabolism, as well as targets of Msn2/4 and of several metabolic regulators, including Hap4, Stb5, and Oaf1 (Dataset EV2). Cluster c was very strongly induced in wild-type cells and only slightly dependent on Msn2/4 for induction; this group was also enriched for targets of many environmentally responsive regulators, especially of osmo-responsive factors Hot1 and Sko1 that can super-induce osmo-responsive genes beyond Msn2/4-regulated levels, specifically during osmotic stress (Gasch, 2002a;

Gasch et al, 2000; Rep et al, 1999). Collectively, promoters of genes in clusters a and c are bound by more regulators than the average salt-responsive gene (FDR < 0.05, Wilcoxon rank-sum test, Fig. 5D). This is consistent with the known combinatorial fine-tuning of the genes in the ESR: although Msn2/4 provide a general backdrop for iESR gene induction, condition-specific regulators converge to modulate expression of subsets of genes depending on demands (Gasch, 2002a; Gasch et al, 2000).

Previous studies reported a repression defect in *msn2Δmsn4Δ* cells, but whether this was indirectly due to a defective stress response was not clear (Chasman et al, 2014; Elfving et al, 2014). Msn2/4 bind the *DOT6* promoter after multiple stresses, including salt stress (Elfving et al, 2014; Huebert et al, 2012; Kuang et al, 2017; Ni et al, 2009), and we showed here that Msn2/4 are required for normal Dot6 protein levels (Fig. 4A–D). Thus, we hypothesized that the *msn2Δmsn4Δ* strain would have a defect in *DOT6* mRNA abundance and, in turn, Dot6-dependent gene repression. Indeed, the *msn2Δmsn4Δ* strain had a significant defect repressing genes dependent on Dot6, which fell into Clusters d and e (Fig. 5A–C, Clusters, point 2). The *TOD6* gene (but not *DOT6*) belongs to Cluster d and harbors three upstream GATGAG sequences in its promoter, suggesting that it may be a direct target of Dot6/Tod6 repression. Clusters d and e were both heavily enriched for upstream GATGAG sequences, but no other transcription factor targets aside of general regulator Abf1 for Cluster d that is known to influence RiBi genes (Bosio et al, 2017a; Bosio et al, 2017b) (we note that known RiBi regulator Stb3 (Huber et al, 2011; Liko et al, 2007, 2010) is not included in Yeastract). In fact, clusters d and e comprised genes with a dearth of annotated transcriptional regulators compared to the total set of salt-responsive genes (Fig. 5D). Msn2/4 do not bind most of the promoters of Clusters d and e genes, and Cluster d is actually statistically significantly under-enriched for genes with upstream STRE elements (*P* = 1e-3, hypergeometric test), supporting the model that Msn2/4 dependence is indirect, perhaps in part via direct Dot6 regulation.

Msn2/4 are well-known and characterized transcriptional activators, and both move to the *DOT6* promoter after NaCl and other stresses. We found that wild-type cells maintain *DOT6* mRNA after multiple stresses, whereas the *msn2Δmsn4Δ* cells expressed significantly less *DOT6* mRNA after NaCl, studied here as well as heat, peroxide stress, and glucose starvation studied previously (Elfving et al, 2014; Gasch et al, 2000; Huebert et al, 2012) (Fig. 6A,B). Given that Msn2/4 bind the *DOT6* promoter after stress, affect both *DOT6* mRNA and Dot6 protein levels, and are necessary for normal repression of Dot6 targets, we propose that the most parsimonious explanation is that Msn2/4 counteract repressive features to maintain *DOT6* expression. Because Msn2/4 can bind the *DOT6* promoter even without direct DNA binding, we were unable to ablate binding at the upstream STRE element. We did, however, generate a mutant in which genomic expression of *DOT6* was controlled by a TET-inducible promoter. Cells expressing reduced Dot6 protein levels matching those seen in the *msn2Δmsn4Δ* strain produced a defect in RiBi gene repression (Fig. EV3D–F). Msn2/4 did not have the same impact on *TOD6*: the gene was strongly repressed with RiBi genes in the wild-type responding to multiple stresses and showed weaker repression in the *msn2Δmsn4Δ* mutant, as did other RiBi genes. Furthermore, Msn2/4 are not known to bind its promoter. While evidence suggests that Msn2/4 directly regulates *DOT6*, additional indirect effects may also be at play.

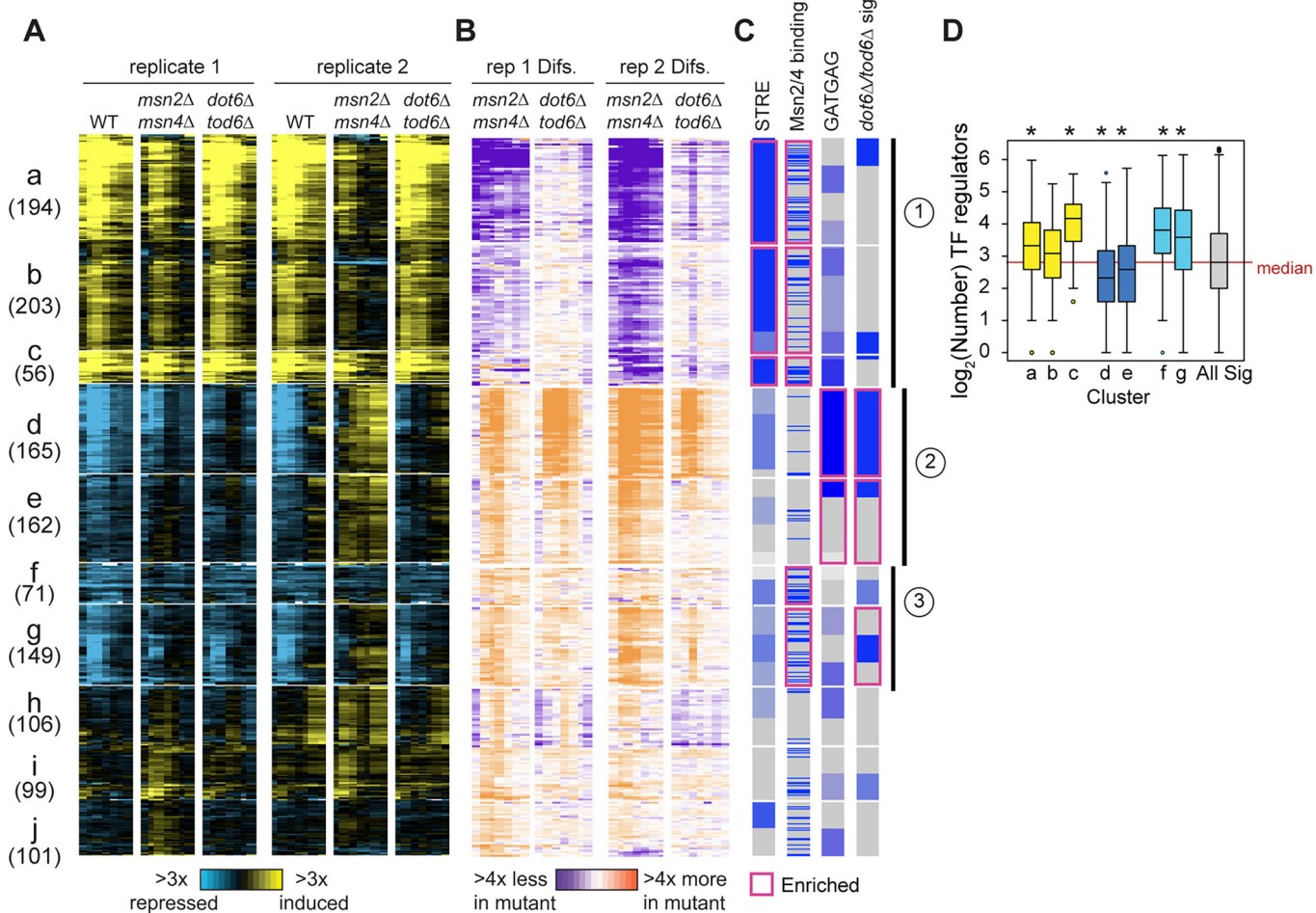

**Figure 5. Msn2/4 influence iESR induction and rESR repression.**

(A, B) In total, 1306 genes (rows) differentially expressed between wild-type and *msn2Δmsn4Δ* cells (FDR < 0.05) in each timepoint (columns) are shown, for 10 clusters (a–j) identified by k-means clustering, where cluster size is indicated in parentheses. (A) Values represent log₂(change) in expression compared to unstressed cells (blue-yellow plot) or (B) the log₂(difference) in fold-change values in each mutant compared to wild-type cells, according to the keys. (C) Blue lines aligned with each gene from (A, B) indicate upstream STRE or GATGAG sequences, genes whose promoters are physically bound by Msn2/4, and genes whose expression is defective in the *dot6tod6Δ* mutant (FDR < 0.05). Features enriched with statistical significance (FDR < 0.02, hypergeometric test) are outlined in magenta. Circled numbers represent discussion points in the text. (D) Distributions of log₂ of the number of promoter-binding transcription factor regulators from Yeastract for genes in each cluster from (A) and 3504 genes with salt-responsive expression changes in the wild-type ('All Sig', FDR < 0.05) at two timepoints, see "Methods". *FDR < 0.05, Wilcoxon rank-sum test compared to all significant genes, exact *P* values provided in Dataset EV1. The median of all significant genes is highlighted with a red line. Boxplots show median (line) and 0.25 and 0.75 quartiles (box), with whiskers extending from minimum to maximum excluding outliers (circles) that are <0.25 quartile–1.5× interquartile range or >0.75 quartile–1.5× interquartile range. Source data are available online for this figure.

In the course of analysis, we realized that loss of *MSN2/4* affected other salt-repressed genes, most notably those in Clusters f and g (Fig. 5A–C, point 3). Cluster f was enriched for genes linked to amino acid transport, methionine synthesis, and the cell periphery (FDR < 0.02, gProfiler (Kolberg et al, 2023)), whereas Cluster g comprised additional rESR genes, including several affected by Dot6/Tod6 deletion and harboring upstream GATGAG elements as well as a subset of ribosomal protein genes and others in the rESR (Fig. 5A–C, point 3). In contrast to the above repressed clusters, these groups were statistically significantly enriched for genes whose promoters are bound by Msn2/4 after multiple stresses (FDR < 0.02, (Elfving et al, 2014; Huebert et al, 2012; Kuang et al, 2017; Ni et al, 2009)). Surprisingly, however, despite increased

upstream Msn2/4 binding, there was no enrichment of upstream STRE elements; in fact Cluster g was strongly under-enriched for genes with upstream STRE sequences (*P* = 3.7e-7, hypergeometric test). This was in stark contrast to induced Msn2/4 targets in clusters a–c (Fig. 5A–C, point 1) that are enriched for both Msn2/4 promoter binding and upstream binding elements. As cited above, Msn2 can bind DNA and regulate gene expression independent of its binding element or even its DNA-binding domain, through molecular interactions with its intrinsically disordered domains (Brodsky et al, 2020; Mahendrawada et al, 2025; Mindel et al, 2024). Our results suggest that repression of these genes, including many in the rESR, is affected by Msn2/4 but through a different mechanism than at iESR genes, perhaps one modulated strictly

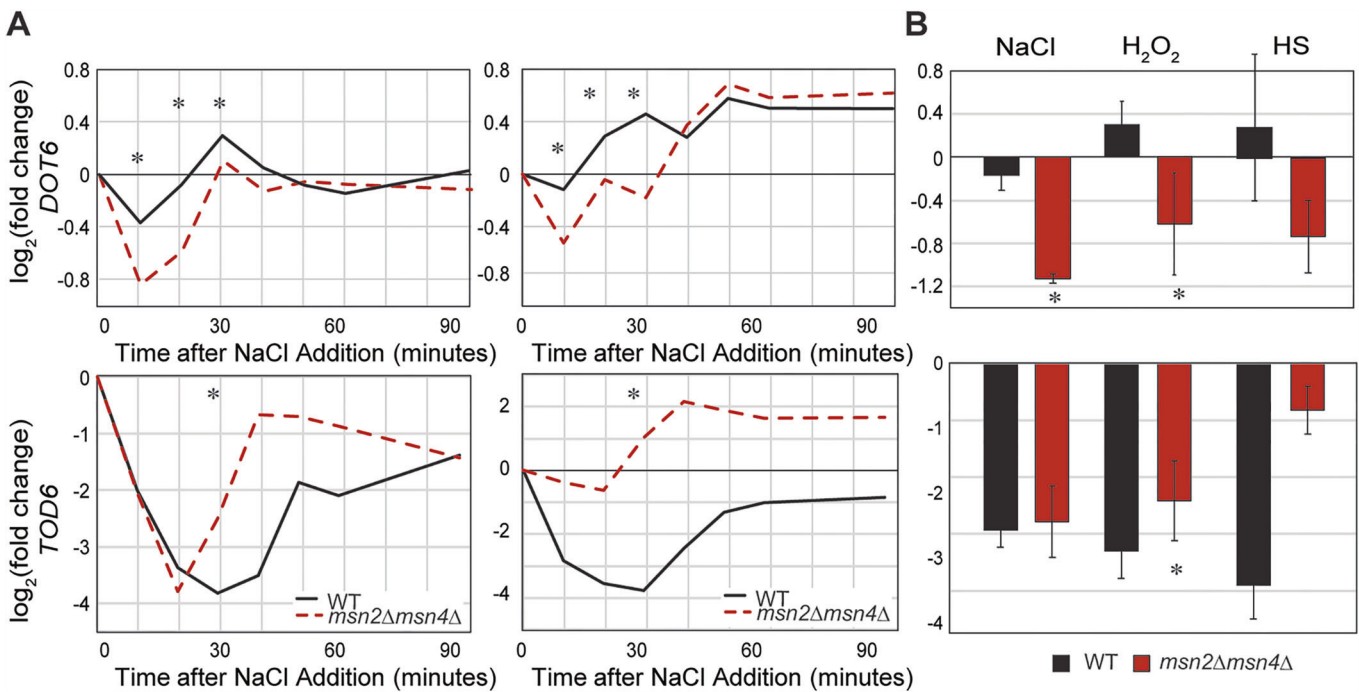

**Figure 6. Msn2/4 are required to maintain *DOT6* mRNA levels after salt stress.**

(A) Relative log₂(fold change) in *DOT6* (top) or *TOD6* (bottom) mRNA abundance at indicated time points compared to unstressed cells in wild-type (black) and *msn2Δmsn4Δ* cells (red) in replicate 1 (left) and 2 (right) RNA-seq NaCl time courses. Asterisk, significant expression difference, FDR < 0.05, *edgeR* analysis. (B) Average and standard deviation (*n* = 3–5) of *DOT6* (top) and *TOD6* (bottom) log₂(fold change) in response to 30 min of 0.7 M NaCl (Chasman et al, 2014), 30 min of 0.4 mM H₂O₂ (Huebert et al, 2012), or 20 min after a 25–37 °C heat shock (Gasch et al, 2000). *P < 0.05, two-tailed *T* test. Exact *P* values are listed in Dataset EV1.

through protein interactions, which will require future dissection to elucidate.

## Modeling suggests evolutionary pressures that maintain the Msn2/4 response

Our results confirm and expand that Msn2/4 contribute to both iESR and rESR expression changes (Chasman et al, 2014; Elfving et al, 2014). The Msn2/4 response comes at a cost to growth rate during a single-stress exposure, since the *msn2Δmsn4Δ* mutant grows faster than wild-type in all single-stress treatments tested (Fig. 1B–D). This raises questions about the environmental pressures required to maintain Msn2/4 over evolutionary time. One pressure would be a yet-uncharacterized condition in which *msn2Δmsn4Δ* cells grow more slowly than wild-type. In this scenario, as long as the average wild-type growth rate weighted by exposure time is greater than that of *msn2Δmsn4Δ* cells, the wild-type will dominate (Abreu et al, 2020). But another possibility explaining the maintenance of the genes is through acquired resistance to a severe secondary stress, where *msn2Δmsn4Δ* cells have a clear defect (Fig. 2B, Berry et al, 2008). An important question is the frequency and severity of otherwise lethal secondary stress events that would be required for evolutionary maintenance of the Msn2/4 response; however, studying this in nature is challenging.

To explore this landscape, we applied a modified Lotka-Volterra competition model, frequently used to characterize competitive dynamics of mixed microbial populations growing in fixed nutrients (Bucci et al, 2016; Davis et al, 2022; Dimas Martins and

Gjini, 2020; Stein et al, 2013). Although we make several simplifying assumptions (including that cells are continuously grown in single-stress conditions), this model can produce ecological bounds under which wild-type cells are more fit than *msn2Δmsn4Δ* cells. We adapted a published piecewise Lotka-Volterra competition model (Hsu and Zhao, 2012) to contrast the relative fitness benefit of *msn2Δmsn4Δ* cells growing in a single-stress condition against the relative fitness defect of *msn2Δmsn4Δ* cells surviving severe secondary stress. This model assumes a fixed carrying capacity, such that cells experience bouts of exponential growth and a stationary phase in the single-stress condition. We used parameters from our experimental results, including that the wild-type grows at ~0.9× the rate of the mutant in single-stress conditions (see Fig. 1D).

The resulting phase map reveals the competitive dynamics of the two strains depending upon frequency (in doubling-time equivalents, *x*-axis) and severity (in percent mutant death, *y*-axis) of severe secondary stress (Fig. 7). Three states emerge, shown with defining parameters for a relative wild-type growth rate of 0.9×. At frequent and/or severe secondary stress treatments, mutant cells either die entirely (state I) or are outcompeted by wild-type (state II), regardless of the relative cell numbers in the culture. State III represents parameter zones in which competition depends on the starting proportion of wild-type and mutant cells. For example, for a starting 1:1 mixed population growing in single-stress conditions, the culture would need to experience 80% *msn2Δmsn4Δ* killing every 50 doubling-time equivalents for the wild-type to outcompete. However, a more realistic scenario is an excess of wild-type cells if the gene losses emerge through new mutation. At a

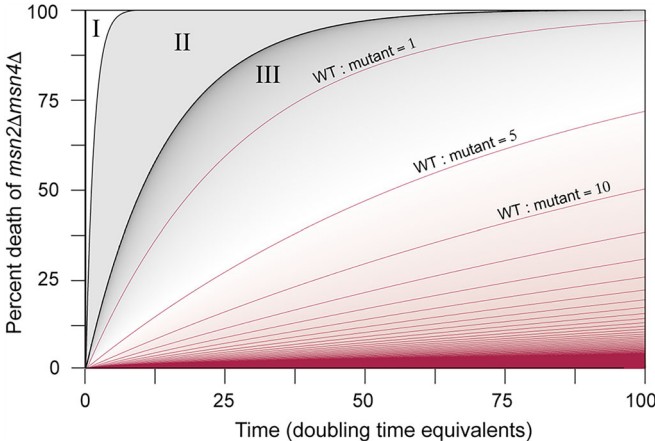

**Figure 7. Modeling competition dynamics suggests environmental pressures that maintain the Msn2/4 response.**

A phase-state map based on the frequency (in doubling-time equivalents, x-axis) and severity (percent mutant killed, y-axis) of secondary stress, for conditions in which wild-type cells grow at 0.9X the *msn2Δmsn4Δ* growth rate, see Methods and main text for details. Competition in state III zones is dependent on the starting populations of wild-type versus mutant cells. Representative contour lines represent parameter space for different starting ratios of wild-type to mutant cells, where area above each line indicates parameters in which wild-type has the competitive advantage for that population makeup. Source data are available online for this figure.

wild-type ratio of 25:1, only 25% of *msn2Δmsn4Δ* cells need to die off every 100 doubling-time equivalents for wild-type cells to maintain the advantage. Periodic growth in "non-stress" conditions in which the mutant has no advantage would further reduce these requirements compared to our model. Given that *S. cerevisiae* experiences fluctuating environments in nature, such as decomposing fruit and insect-based relocation (Goddard and Greig, 2015; Jouhten et al, 2016), it is feasible that infrequent exposure to severe stresses is enough to provide the evolutionary pressure to maintain the Msn2/4 response.

# Discussion

While it was well appreciated that stressed yeast mounts the common ESR response, understanding of the separable roles of iESR induction and rESR repression has remained incomplete. Our results here decouple these responses to quantify the importance of resource reallocation that enables the induced component of the stress response (Fig. 8). Upon acute stress, Msn2/4 transcriptionally induce their targets, most likely through direct binding of upstream STRE elements enriched in the genes' promoters. Msn2/4 also influences rESR repression, likely through both direct effects on *DOT6* and other rESR genes through promoter binding, and indirect effects. Thus, Msn2/4 influence iESR induction and rESR repression, observed in prior studies (Chasman et al, 2014; Elfving et al, 2014). Defective rESR repression (e.g., through *DOT6/TOD6* deletion) in the context of an otherwise normal Msn2/4 response (see Fig. 1) produced a drag on post-stress growth rate, delayed production of defense proteins, and slowed acquisition of stress tolerance. Thus, we argue that the effect of Msn2/4 on rESR genes helps to coordinate resource reallocation needed for its action on induced genes.

The insights uncovered here likely pertain to stress defense systems in other organisms as well. This includes the so-called stringent response in bacteria, where the alarmone metabolite ppGpp suppresses transcription of growth-promoting genes while supporting synthesis of proteins needed for survival (Gourse et al, 2018; Zhu et al, 2019). Similar themes emerge in the mammalian integrated stress response (ISR), where a host of kinases, each responding to different signs of adversity, phosphorylate eIF2α to inhibit global translation but stimulate production of stress-responsive transcription factors that induce downstream defense targets (Costa-Mattioli and Walter, 2020; Dever et al, 2023; Harding et al, 2003; Houston et al, 2020). Results from our study in yeast can therefore reflect generalizable insights into stress responses across species.

The first is that cells alternate between objectives to balance the cost of defense versus growth. Under the conditions studied here, cells are clearly not maximizing post-stress growth rate after a single stress treatment, since they are capable of growing faster in the absence of an Msn2/4 response (Fig. 1B,C). This adds to a growing body of evidence that maximizing growth is not a universal objective in microbes, especially during adversity (Balakrishnan et al, 2021; Basan, 2018; Basan et al, 2020; Dai et al, 2018; Ho et al, 2018; Korem Kohanim et al, 2018; Schuetz et al, 2012; Zhu et al, 2024). It also shows that mere abundance of ribosome-related transcripts does not predict "instantaneous" growth rate, since cells with (albeit transiently) fewer RiBi transcripts grow faster after salt stress (Fig. 2 and Bergen et al, 2022).

Instead of maximizing growth at all costs, cells invest in preparing for impending stress at the first signs of adversity. Activation of Msn2/4 contributes to acquired stress resistance in yeast and other fungi (Berry et al, 2011; Berry and Gasch, 2008; Brown et al, 2014; Gasch, 2007; Liang et al, 2023). Similarly, activation of the bacterial stringent response supports acclimation to suboptimal carbon sources by shortening the lag phase required to produce needed proteins (Balakrishnan et al, 2021; Boutte and Crosson, 2013; Gourse et al, 2018; Zhu and Dai, 2023). Upon a shift away from optimal carbon sources, both yeast and bacteria invest in producing enzymes for alternate sugar utilization, even when the substrate sugars of those enzymes are not present (Balakrishnan et al, 2021; Simpson-Lavy and Kupiec, 2019; Turcotte et al, 2010; Vermeersch et al, 2022). Activating

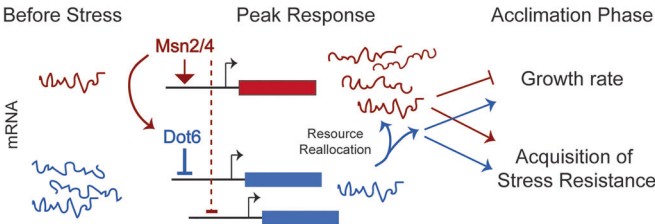

**Figure 8.  Model for Msn2-dependent resource reallocation during stress.**

Msn2/4 activation induces defense genes (red) and *DOT6* by direct regulation, while also influencing stress-dependent repression of ribosome and growth genes (blue), both directly through promoter binding and indirectly by influencing Dot6 activity. Tod6 may contribute to RiBi repression under other stress conditions, but did not have a significant impact in this study. Resource reallocation provided by transient rESR repression enables and accelerates the costly Msn2/4 response, promoting faster acquisition of subsequent stress tolerance, see text for details.

these responses comes with a cost to growth rate, explaining why maximal stress tolerance is not constitutive in these organisms (Balakrishnan et al, 2021; Basan et al, 2020; Zhu et al, 2024; Zhu and Dai, 2023). But it also underscores the importance of anticipatory programs in evolution and reveals a unifying theme for fast-growing microbes like *S. cerevisiae* and *E. coli*: when times are good, cells direct resources to support maximal growth, but in response to early signs of adversity, they redirect focus to invest in the future. An important corollary of this result is that growth rate is not a universal proxy for fitness in nature.

Second, our results delineate the importance of transcriptional and translational suppression for resource reallocation during stress. The isolated role of this suppression has been hard to study in other organisms, because it is often tightly coupled with the production of defense transcripts and proteins. Nonetheless, it has been suggested in *E. coli*. Cells lacking ppGpp grow fine without stress, but have a much longer lag when shifted to suboptimal conditions; conversely, ppGpp over-production, leading to stronger repression of growth-promoting genes, slows growth in the absence of stress but accelerates stress acclimation and promotes stress tolerance (Zhu et al, 2019, 2024; Zhu and Dai, 2023). Furthermore, over-production of unnecessary proteins slows cell growth rate and stress acclimation, supporting the notion that a tax on protein-synthesis capacity is suboptimal (Balakrishnan et al, 2021; Basan et al, 2020). The intimate coupling of stress-induced and -repressed responses across organisms mounting common stress responses underscores that coregulated resource reallocation is a unifying principle.

A remaining question had been how this resource reallocation is regulated in yeast. Our results imply that resource reallocation is built into the ESR program (Fig. 8). Msn2/4 maintain Dot6 protein levels, likely through direct binding of the *DOT6* promoter and potentially other indirect effects (Figs. 4, 6, and EV5). Our evidence suggests they also contribute to repression of other rESR promoters during stress in a manner that may be distinct from induced proteins, given the enrichment of Msn2/4 promoter binding but dearth of upstream binding elements for these gene groups (Fig. 5). Thus, Msn2/4 activity helps to orchestrate the resource reallocation needed for its own response. A remaining mystery is how yeast sense their internal system to set the balance between growth-promoting and stress-defending programs. *S. cerevisiae* does not utilize ppGpp, which in *E. coli* directly senses translational flux at individual ribosomes (Wu et al, 2022). In yeast, PKA and TOR may play a role, since they respond to quality nutrients to promote

growth at the expense of defense (González and Hall, 2017; Kocik and Gasch, 2022). Even in an organism as well-studied as *S. cerevisiae*, these mysteries await further investigation.

# Methods

**Reagents and tools table**

| Reagent/resource | Reference or source | Identifier or catalog number |
| --- | --- | --- |
| **Experimental models** | | |
| *S. cerevisiae* - MATa, his3Δ1, leu2Δ0, ura3Δ0, met15Δ0 | Open Biosystems | AGY4 |
| *S. cerevisiae* - MATα, his3Δ1, leu2Δ0, ura3Δ0, lys2Δ0 | Open Biosystems | AGY5 |
| *S. cerevisiae* - MATa, his3Δ1, leu2Δ0, ura3Δ0, met15Δ0, ctt1::KAN | Open Biosystems | AGY345 |
| *S. cerevisiae* - MATa, his3Δ1, leu2Δ0, ura3Δ0, met15Δ0, dot6::KAN | Open Biosystems | AGY523 |
| *S. cerevisiae* - MATa, his3Δ1, leu2Δ0, ura3Δ0, met15Δ0, tod6::KAN | Open Biosystems | AGY524 |
| *S. cerevisiae* - MATa, his3Δ1, leu2Δ0, ura3Δ0, met15Δ0, dot6::KAN, tod6::HYG | Lee et al, 2011 | AGY594 |
| *S. cerevisiae* - MATa, his3Δ1, leu2Δ0, ura3Δ0, met15Δ0 DOT6-GFP(S65T)-His3MX, MSN2-mCherry-HYGMX | Bergen et al, 2022 | AGY1328 |

| Reagent/resource | Reference or source | Identifier or catalog number |
|---|---|---|
| S. cerevisiae - MATa, his3Δ1, leu2Δ0, ura3Δ0, met15Δ0, TOD6-GFP(S65T)-His3MX | Open Biosystems | AGY1441 |
| S. cerevisiae - MATα, his3Δ1, leu2Δ0, ura3Δ0, met15Δ0, msn2::KAN, msn4::HYG | This study | AGY2046 |
| S. cerevisiae - MATa, his3Δ1, leu2Δ0, ura3Δ0, met15Δ0 DOT6-GFP(S65T)-His3MX, MSN2-mCherry-HYGMX | This study | AGY2228 |
| S. cerevisiae - MATα, his3Δ1, leu2Δ0, ura3Δ0, lys2Δ0, msn2::KAN, msn4::HYG | This study | AGY2229 |
| S. cerevisiae - MATa, his3Δ1, leu2Δ0, ura3Δ0, met15Δ0 DOT6-GFP(S65T)-His3MX, MSN2-mCherry-HYGMX, NHP6A-iRFP | This study | AGY2233 |
| S. cerevisiae - MATa, his3Δ1, leu2Δ0, ura3Δ0, met15Δ0, dot6::KAN tod6::HYG, MSN2-mCherry-HYGMX | This study | AGY2234 |
| S. cerevisiae - MATa, his3Δ1, leu2Δ0, ura3Δ0, met15Δ0, MSN2-mCherry-HYGMX | This study | AGY2237 |
| S. cerevisiae - MATa, his3Δ1, leu2Δ0, ura3Δ0, met15Δ0, TOD6-GFP(S65T)-His3MX, NHP6A-iRFP | This study | AGY2242 |
| S. cerevisiae - MATα, his3Δ1, leu2Δ0, ura3Δ0, met15Δ0, chrV: 333177-333677::URA3 | This study | AGY2244 |
| S. cerevisiae - MATα, his3Δ1, leu2Δ0, ura3Δ0, met15Δ0, chrV: 333177-333677::URA3, DOT6-GFP(S65T)-His3MX | This study | AGY2245 |

| Reagent/resource | Reference or source | Identifier or catalog number |
|---|---|---|
| S. cerevisiae - MATα, his3Δ1, leu2Δ0, ura3Δ0, met15Δ0 TOD6-GFP(S65T)-His3MX, MSN2-mCherry-HYGMX | This study | AGY2247 |
| S. cerevisiae - MATα, his3Δ1, leu2Δ0, ura3Δ0, met15Δ0, msn2::KAN msn4::HYG, DOT6-GFP(S65T)-His3MX | This study | AGY2254 |
| S. cerevisiae - MATα, his3Δ1, leu2Δ0, ura3Δ0, met15Δ0, msn2::KAN msn4::HYG, DOT6-GFP(S65T)-His3MX, NHP6A-iRFP | This study | AGY2255 |
| S. cerevisiae - MATα, his3Δ1, leu2Δ0, ura3Δ0, met15Δ0, MYO2pr-rtTA-TETO7promoter-DOT6 | This study | AGY2256 |
| S. cerevisiae - MATα, his3Δ1, leu2Δ0, ura3Δ0, met15Δ0, MYO2pr-rtTA-TETO7promoter-DOT6 -GFP(S65T)-His3MX | This study | AGY2257 |
| S. cerevisiae - MATα, his3Δ1, leu2Δ0, ura3Δ0, met15Δ0, msn2::KAN msn4::HYG, TOD6-GFP(S65T)-His3MX | This study | AGY2260 |
| S. cerevisiae - MATα, his3Δ1, leu2Δ0, ura3Δ0, met15Δ0, msn2::KAN, msn4::HYG dot6::KAN tod6::HYG | This study | AGY2264 |
| **Oligonucleotides and other sequence-based reagents** | | |
| PCR Primer (msn4::HYG forward) | This study (IDT) | 5'-TATCAGTTCGGCTTTTTTTTCTTTTCTTCTTATTAAAAACGTTCGAGTTTATCATTATCAATACTGCC-3' |
| PCR Primer (msn4::HYG reverse) | This study (IDT) | 5'-GCTTGTCTTGCTTTTATTTGCTTTTGACCTTATTTTTTTCCCGGTAGAGGTGTGGTCAATAAG-3' |
| PCR Primer (URA3-DOT6 forward) | This study (IDT) | 5'-CCACCACCATCGCTACCAACAGCAGGATATCCGGATGCAGTTTGACGCTTTTCTGGGTAGAAGATCGGTCTG-3' |

| Reagent/resource | Reference or source | Identifier or catalog number |
|---|---|---|
| PCR Primer (URA3-DOT6 reverse) | This study (IDT) | 5'-ATGCTGCTCAAATGAATGGAAG CTGAGTTCAAACTGG TTGAAATGGACATC AGTTCAATACAACAGATCACGTG-3' |
| PCR Primer (TET-DOT6 forward) | This study (IDT) | 5'-CCACCACCATCGCT ACCAACAGCAG GATATCCGGATGCAGTTTGACGCTTA ACGCCGTTTCTCGATGCTTATCTG-3' |
| PCR Primer (TET-DOT6 reverse) | This study (IDT) | 5'-ATGCTGCTCAAATGA ATGGAAGCTG AGTTCAAACTGGTTGAAATGGACATG TGCTCAGTATCTCTATCACTGATA-3' |
| **Chemicals, enzymes, and other reagents** | | |
| Yeast nitrogen base without ammonium sulfate, folic acid, or riboflavin | Sigma | Y1251-100G |
| Ammonium sulfate | Sigma | A2939-100G |
| Yeast synthetic drop-out medium without histidine | Sigma | Y1751 |
| Histidine | Fisher | BP382-100 |
| Dextrose | Fisher | D16500 |
| Sodium chloride | Fisher | AC327300010 |
| Hydrochloric acid | Fisher | A144SI-212 |
| Sodium hydroxide | Fisher | S3818100 |
| Absolute ethanol | Fisher | BP28184 |
| Hydrogen peroxide (30% in water) | Fisher | BP2633500 |
| Doxycycline hyclate | Sigma | D9891-5G |
| Zymolyase | Zymo Research | E1004 or E1005 |
| Concanavalin A | MP Biomedical | 150710 |
| Phenol ≥99.0% saturated with buffer pH 4.5 | VWR | 0981-400 ML |
| Chloroform | Fisher | C298-500 |
| RNase-free DNase set | Qiagen | 79254 |
| **Software** | | |
| NIS-Elements | Nikon | n/a |
| MATLAB | The MathWorks, Inc. | Version R2022b |
| R | R Foundation, USA | Version 4.3.1 |
| Gene Cluster 3.0 | Eisen et al, 1998 | n/a |
| Java TreeView | Saldanha, 2004 | Version 1.2.0 |

| Reagent/resource | Reference or source | Identifier or catalog number |
|---|---|---|
| Trimmomatic | Bolger et al, 2014 | Version 0.39 |
| Bowtie 2 | Langmead and Salzberg, 2012 | Version 2.4.4 |
| HTSeq | Anders et al, 2015 | Version 0.6.0 |
| edgeR | Robinson and Oshlack, 2010 | Version 4.3.2 |
| Yeastract | Teixeria et al, 2023 | n/a |
| RSAT | Santana-Garcia et al, 2022 | n/a |
| **Other** | | |
| Spectronic 20D+ spectrophotometer | 2007 Thermo Fisher Scientific Inc | n/a |
| FCS2 chamber | Bioptechs Inc | n/a |
| Nikon Eclipse Ti inverted microscope with perfect focus system | Nikon | n/a |
| ET-EGFP single band filter cube | Chroma Technology Corp | #49002 |
| ET/mCH/TR single band filter cube | Chroma Technology Corp | #96365 |
| Cy 5.5 single band filter cube | Chroma Technology Corp | #49022 |
| RNeasy MinElute Cleanup Kit | Qiagen | #74204 |
| TruSeq Stranded mRNA Kit | Illumina | 20020595 |
| AMPure XP Beads | Fisher Scientific | NC9933872 |
| NovaSeq 6000 | Illumina | n/a |

## Strains and growth conditions

*Saccharomyces cerevisiae* strains of the BY4741 background used in this study are listed in Table 1. All strains were grown in Low Fluorescent Media (LFM) as previously described in Bergen et al, 2022 (0.17% yeast nitrogen base without ammonium sulfate, folic acid, or riboflavin; 0.5% ammonium sulfate; 0.2% complete amino acid supplement, and 2% glucose). Strain AGY2046 (*msn2Δmsn4Δ*) was generated by replacing *MSN4* in the BY4741 *msn2::KANMX* strain (Open Biosystems) with the hygromycin-MX cassette via homologous recombination and validated using diagnostic PCRs.

**Table 1.  Strains used in this study.**

| Strain Name | Description | Source |
|---|---|---|
| AGY4 | *MATa, his3Δ1, leu2Δ0, ura3Δ0, met15Δ0* | Open Biosystems |
| AGY5 | *MATα, his3Δ1, leu2Δ0, ura3Δ0, lys2Δ0* | Open Biosystems |
| AGY345 | *MATa, his3Δ1, leu2Δ0, ura3Δ0, met15Δ0, ctt1::KAN* | Open Biosystems |
| AGY523 | *MATa, his3Δ1, leu2Δ0, ura3Δ0, met15Δ0, dot6::KAN* | Open Biosystems |
| AGY524 | *MATa, his3Δ1, leu2Δ0, ura3Δ0, met15Δ0, tod6::KAN* | Open Biosystems |
| AGY594 | *MATa, his3Δ1, leu2Δ0, ura3Δ0, met15Δ0, dot6::KAN, tod6::HYG* | Lee et al, 2011 |
| AGY1328 | *MATa, his3Δ1, leu2Δ0, ura3Δ0, met15Δ0 DOT6-GFP(S65T)-His3MX, MSN2-mCherry-HYGMX* | Bergen et al, 2022 |
| AGY1441 | *MATa, his3Δ1, leu2Δ0, ura3Δ0, met15Δ0, TOD6-GFP(S65T)-His3MX* | Open Biosystems |
| AGY2046 | *MATa, his3Δ1, leu2Δ0, ura3Δ0, met15Δ0, msn2::KAN, msn4::HYG* | This study |
| AGY2228 | *MATα, his3Δ1, leu2Δ0, ura3Δ0, met15Δ0 DOT6-GFP(S65T)-His3MX, MSN2-mCherry-HYGMX* | This study |
| AGY2229 | *MATα, his3Δ1, leu2Δ0, ura3Δ0, lys2Δ0, msn2::KAN, msn4::HYG* | This study |
| AGY2233 | *MATa, his3Δ1, leu2Δ0, ura3Δ0, met15Δ0 DOT6-GFP(S65T)-His3MX, MSN2-mCherry-HYGMX, NHP6A-iRFP* | This study |
| AGY2234 | *MATa, his3Δ1, leu2Δ0, ura3Δ0, met15Δ0, dot6::KAN tod6::HYG, MSN2-mCherry-HYGMX* | This study |
| AGY2237 | *MATa, his3Δ1, leu2Δ0, ura3Δ0, met15Δ0, MSN2-mCherry-HYGMX* | This study |
| AGY2242 | *MATa, his3Δ1, leu2Δ0, ura3Δ0, met15Δ0, TOD6-GFP(S65T)-His3MX, NHP6A-iRFP* | This study |
| AGY2244 | *MATa, his3Δ1, leu2Δ0, ura3Δ0, met15Δ0, chrV: 333177-333677::URA3* | This study |
| AGY2245 | *MATa, his3Δ1, leu2Δ0, ura3Δ0, met15Δ0, chrV: 333177-333677::URA3, DOT6-GFP(S65T)-His3MX* | This study |
| AGY2247 | *MATa, his3Δ1, leu2Δ0, ura3Δ0, met15Δ0 TOD6-GFP(S65T)-His3MX, MSN2-mCherry-HYGMX* | This study |
| AGY2254 | *MATa, his3Δ1, leu2Δ0, ura3Δ0, met15Δ0, msn2::KAN msn4::HYG, DOT6-GFP(S65T)-His3MX* | This study |
| AGY2255 | *MATa, his3Δ1, leu2Δ0, ura3Δ0, met15Δ0, msn2::KAN msn4::HYG, DOT6-GFP(S65T)-His3MX, NHP6A-iRFP* | This study |
| AGY2256 | *MATa, his3Δ1, leu2Δ0, ura3Δ0, met15Δ0, MYO2pr-rtTA-TETO7promoter-DOT6* | This study |
| AGY2257 | *MATa, his3Δ1, leu2Δ0, ura3Δ0, met15Δ0, MYO2pr-rtTA-TETO7promoter-DOT6 -GFP(S65T)-His3MX* | This study |
| AGY2260 | *MATa, his3Δ1, leu2Δ0, ura3Δ0, met15Δ0, msn2::KAN msn4::HYG, TOD6-GFP(S65T)-His3MX* | This study |
| AGY2264 | *MATa, his3Δ1, leu2Δ0, ura3Δ0, met15Δ0, msn2::KAN, msn4::HYG dot6::KAN tod6::HYG* | This study |

Strain AGY2244 and AGY2245 were generated by replacing 500 bp upstream of *DOT6* start codon in the BY4741 wild-type strain (AGY4) or AGY1319 expressing DOT6-GFP with the *K. lactis URA3* gene via homologous recombination to knock out the endogenous *DOT6* promoter region, then replacing *K. lactis URA3* upstream of the *DOT6* (or DOT6-GFP) start codon with a construct containing the Tet promoter along with the rtTA inducer (MYO2promoter-rtTA-TETO7promoter – gift from Michael Springer Lab) via homologous recombination, selecting for loss of uracil prototrophy. The strain was validated using diagnostic PCR and microscopy to verify doxycycline-responsive Dot6-GFP induction. The remaining strains were generated through genetic crosses as listed below, dissection of haploid spores, and selection of spores with appropriate markers. Gene deletions were verified by diagnostic PCR and fluorescent microscopy when appropriate. Crosses were used to generate AGY2228 (AGY1328 x AGY5), AGY2229 (AGY2046 x AGY5), AGY2233 (AGY1328 x AGY2232), AGY2234 (AGY594 x AGY2228), AGY2237 (AGY1328 x AGY5), AGY2242 (AGY1441 x AGY2232), AGY2254 (AGY1328 x AGY2229), AGY2247 (AGY1441 x AGY2237), AGY2255 (AGY2234 x AGY2232), AGY2260 (AGY1441 x AGY2229), and AGY2264 (AGY594 x AGY2229).

## Liquid growth curves

Liquid cultures for growth rate assessment were inoculated in test tubes from an overnight culture grown ~12 h in LFM and grown for at least 4.5 h to a starting optical density at 600 nm ($OD_{600}$) of ~0.1 before measurements were taken every 15 min. For salt stress, NaCl was added to 0.7 M NaCl. For heat stress, cells were collected by centrifugation and then shifted to 40 °C media. For alkaline stress, cells were collected by centrifugation and transferred to LFM adjusted to pH 7.5. For acid stress, cells were collected by centrifugation and transferred to LFM adjusted to pH 2.75. For ethanol stress, absolute ethanol was added to 3% and 4%. Growth rates were calculated by fitting exponential curves to data from time points spanning 75 min to 225 min after NaCl was added or exponential time points for other stresses. Mutants were grown side-by-side with wild-type cultures, with paired replicates done on separate days, allowing paired statistical analysis.

## Acquired stress resistance experiments

See Fig. 2A for a schematic of this protocol. Cultures were grown in LFM in flasks at 30 °C in a shaking incubator for at least 15 h to a starting

OD$_{600}$ ~0.3–0.4. An aliquot of unstressed cells (0 min) was removed, and then NaCl was added to a final concentration of 0.7 M. At various timepoints following the addition of NaCl (10, 20, 30, 40, 50, 60, 90, and 120 min), an aliquot of culture was retrieved, cells collected by brief centrifugation, and resuspended in fresh LFM without NaCl to an OD$_{600}$ of 0.6. Cells were subsequently threefold diluted into 96-well plates containing LFM or LFM plus one of 11 doses of H$_2$O$_2$ (spanning from 0 to 20 mM final concentration of H$_2$O$_2$). Cells were incubated for 2 h at 30 °C in a shaking incubator, then a 200-fold dilution of each culture was spotted on YPD agar plates (1% yeast extract, 2% peptone, 2% glucose, and 2% agar). Plates were grown ~48 h at 30 °C, then viability at each dose of H$_2$O$_2$ "secondary" stress was scored visually on a four-point scale: 100% (3), 50–100% (2), 10–50% (1), and 0% (0) survival compared to the wild-type cells treated with NaCl but no H$_2$O$_2$. A single H$_2$O$_2$ survival score was calculated for each time point as the sum of scores across the 11 different doses of H$_2$O$_2$. Each mutant was compared to wild-type culture grown side-by-side on each day, with 3 biological replicates for most strains except the *dot6Δtod6Δ*, which was done with 6 replicates for added statistical power.

## Microscopy and image analysis

Time-lapse microscopy was performed using an FCS2 chamber (Bioptechs Inc., Butler, Pennsylvania). Data collection and analyses were conducted as previously described in Bergen et al, 2022 (Bergen et al, 2022), with the following changes. Each mutant was grown to mid-log phase in LFM media in a flask and then mixed within the microfluidic chamber at a 50:50 ratio with the iRFP-tagged wild-type strain. Media flow was switched from LFM to LFM + 0.7 M NaCl after T12, as previously described. The GFP and mCherry signal was recorded at each time point before and after NaCl treatment as previously described. To distinguish wild-type from mutant cells in mixed cultures, Principal component analysis of cells was performed based on the iRFP signal across timepoints T1-T40, using R Statistical Software (R version 4.3.1). This analysis led to a clear dichotomy of cell types that also correlated with the presence of both GFP and mCherry signal, where mutant cells showed no iRFP and loss of either GFP or mCherry signal according to the strain.

Dot6-GFP and Msn2-mCherry phenotypes were also determined as previously described, including fraction of nuclear Dot6-GFP and Msn2-mCherry signal ("nuclear/cytoplasmic ratio", defined as the average signal of the top 5% of pixels divided by the median of all pixels), acute stress peak height, and area under the curve of fraction of nuclear signal across pre-stress or post-stress timepoints (Bergen et al, 2022). Tod6-GFP phenotypes were also calculated in the same way. Acute stress peak height as shown in Figs. 3–4 and EV2 was calculated as the maximum nuclear localization score just after NaCl addition (T13–T20) minus the minimum fraction of nuclear signal just before salt was added (T11–T13). Dot6 and Tod6 abundance was measured based on the median Dot6-GFP or Tod6-GFP signal in each cell; the average signal before NaCl treatment (T1–T12) or after (T20–T36) is shown in Figs. 4C and EV2B, respectively. Cells shown in Fig. 4D were defined as those with a similar levels of Dot6-GFP signal (values between 635 to 650 signal intensity), such that the mutant and wild-type signal were not different (Wilcoxin rank-sum test *P* > 0.05). Cells shown in Fig. EV2C were defined as those with similar levels of Tod6-GFP signal (values between 605 and 615 signal intensity), such that mutant and wild-type signals were not different (Wilcoxon rank-sum test *P* > 0.05).

Cell clustering in Figs. 3–4 was performed based on the population median (i.e., each column) of GFP or mCherry nuclear signal using Gene Cluster 3.0 (Eisen et al, 1998) and visualized using Java TreeView version 1.2.0 (Saldanha, 2004). The fraction of nuclear signal shown on the left was added after clustering for display. Data are available in Dataset EV3.

## RNA sequencing

RNA-seq was performed using total RNA isolated from log-phase cultures before and after NaCl treatment. Cultures were grown in LFM in flasks at 30 °C in a shaking incubator for at least 16 h to a starting OD$_{600}$ of mid-log phase. For TET-inducible experiments, all cells being compared were grown in LFM + 50 μg/mL Doxycycline (Sigma-Aldrich) for 16 h. 5 mL of culture was harvested via centrifugation at 3000 RPM for 3 min, flash frozen in liquid nitrogen, and stored at −80 °C. Total RNA was extracted using hot phenol lysis (Gasch, 2002b) and purified using the RNeasy MinElute Cleanup Kit (QIAGEN, Hilden, Germany) and DNase digestion. RNA-seq libraries were prepared using a TruSeq Stranded Total RNA kit (Illumina), and PCR purified using AMPure XP beads (Beckman Coulter, Indianapolis, IN). Paired-end sequencing was performed on an Illumina NovaSeq 6000 sequencer (Illumina).

RNA-seq reads were processed using Trimmomatic version 0.39 (Bolger et al, 2014) and mapped to the S288c genome using Bowtie 2 version 2.4.4 (Langmead and Salzberg, 2012). Read counts for each gene were calculated using HTSeq version 0.6.0 (Anders et al, 2015). Raw data can be found in the NIH GEO database Accession GSE283327. Differentially expressed genes were identified at each timepoint using a glm model in edgeR version 4.3.2 using TMM normalization (Robinson and Oshlack, 2010) with significance at <0.05 Benjamini and Hochberg false discovery rate (FDR) (Benjamini and Hochberg, 1995). Genes significant in at least two time points were considered for analysis. Cells with an induction (or repression) defect were defined if the gene was induced (or repressed) in a majority of time points in the wild-type cells and showed a smaller log$_2$(fold change) in the corresponding mutant. Data are available in Dataset EV4.

In total, 1306 genes with significant differences in salt-induced expression in the *msn2Δmsn4Δ* strain were partitioned into $k = 10$ clusters by k-means clustering (where k was defined manually) based on the log$_2$(fold change) in expression in wild-type, *msn2Δmsn4Δ*, and *dot6Δtod6Δ* time courses, along with the log$_2$ differences at each timepoint in each mutant versus wild-type (as shown in Fig. 5A,B). Targets of 184 transcription factors with evidence of direct DNA binding were received courtesy of Yeastract (Teixeira et al, 2023) and used to score enrichment in each cluster compared to all yeast genes, using the hypergeometric test and Benjamini-Hochberg correction. Upstream STRE (CCCCT) or GATGAG sequences were scored within 500 bp upstream of genes in each cluster, searching both strands and preventing overlapping hits, using RSAT (Santana-Garcia et al, 2022).

## Modeling

To explore the necessary frequency and severity of secondary stress treatments required to provide wildtype cells with an advantage, we constructed the following piecewise Lotka-Volterra model,

assuming that the yeast grow exponentially with linear competition until saturation, in a culture that can support a fixed number of cells (i.e., which has a fixed carrying capacity). We assumed both the carrying capacities and mutual competition of the two strains to be equivalent, as Msn2/4 are not known to have an effect on these traits. We further simplified by modeling only a single-stress growth condition in which the wild-type grows more slowly and where wild-type death during secondary stress is discounted, since it is much less than that of the mutant. The first assumption makes our model more stringent than is likely in nature. These assumptions yield the following model

$$\frac{dN_{WT}}{dt} = \begin{cases} r_{WT}N_{WT}(1 - \frac{N_{WT}+N_{Null}}{K}) & m\omega + (1-\phi)\omega \le t \le (m+1)\omega \\ 0 & m\omega \le t \le m\omega + (1-\phi)\omega \end{cases}$$

$$\frac{dN_{Null}}{dt} = \begin{cases} r_{Null}N_{Null}(1 - \frac{N_{WT}+N_{Null}}{K}) & m\omega + (1-\phi)\omega \le t \le (m+1)\omega \\ -DN_{Null} & m\omega \le t \le m\omega + (1-\phi)\omega \end{cases}$$

where $N_{WT}$ is the number of wildtype cells and $N_{Null}$ is the number of $msn2\Delta msn4\Delta$ cells, with $r_{WT}$ and $r_{Null}$ as their respective growth rates. K is the carrying capacity, $D$ is the death rate of $msn2\Delta msn4\Delta$ cells in secondary stress conditions, $\omega$ is the period of secondary stress cycles with $\omega(1-\phi)$ the duration of the resulting severe stress.

This in turn enables us to use a modified version of the results in Hsu and Zhao, 2012, deriving the following possible states depending on the frequency and severity of secondary stresses:

State I: The $msn2\Delta msn4\Delta$ death per secondary-stress exposure is greater than recovered by their growth between secondary stress events, such that all $msn2\Delta msn4\Delta$ cells die regardless of competition.

State II: Sufficient $msn2\Delta msn4\Delta$ growth to compensate for death occurs between secondary stress events, however, they are outcompeted by wild-type cells regardless of relative initial proportions of the two strains in the population.

State III: Either the $msn2\Delta msn4\Delta$ cells or the wildtype cells can outcompete the other, depending on relative initial proportions of the two strains.

We can then reparametrize the above equations in terms of the relative fitness $\alpha = \frac{r_{WT}}{r_{Null}}$, time between secondary stresses normalized by the uninhibited doubling time $\beta = \frac{\phi}{ln(2)}$, and the mutant mortality following each secondary stress exposure $\frac{Death_{Null}}{N_{Null}} = 1 - e^{-D(1-\phi)}$. For $r_{Null} = 1$, the states are defined by the following formulae:

State I: $msn2\Delta msn4\Delta$ cells are nonviable if

$$\frac{Death_{Null}}{N_{Null}} > 1 - e^{-\beta ln(2)}$$

State II: Wild-type cells always outcompete $msn2\Delta msn4\Delta$ cells if

$$\frac{Death_{Null}}{N_{Null}} > 1 - e^{(\alpha-1)\beta \ln(2)}$$

Otherwise, wild-type cells outcompete $msn2\Delta msn4\Delta$ cells so long as

$$\frac{N_{WT}}{N_{Null}} > \frac{(1-\frac{1}{\alpha})\beta \, ln(2)}{ln\left(1 - \frac{Death_{Null}}{N_{Null}}\right)} - \frac{1}{\alpha}$$

## Data availability

The RNA-seq data from this publication have been deposited to the NIH GEO database https://www.ncbi.nlm.nih.gov/geo/] and assigned the identifier #GSE283327.

The source data of this paper are collected in the following database record: biostudies:S-SCDT-10_1038-S44318-026-00727-x.

## Peer review information

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

## Acknowledgements

We thank the McClean lab for reagents and microfluidics support, the M. Springer Lab for the plasmid used to engineer the TET promoter strains, James Hose for help with RNA-sequencing, Michael Place for computational assistance, and members of the Gasch Lab for constructive feedback. We thank the Wisconsin-Madison University Biotechnology Center DNA Sequencing Core Facility (RRID: SCR_017759) for sequencing. This work was supported by NIH grant R01GM14975 to APG, NIH R35GM128873, and R01AI154940 to MNM. RAK and EGC were supported by an NHGRI training grant to the Genomic Sciences Training Program T32HG002760.

## Author contributions

**Rachel A Kocik**: Conceptualization; Data curation; Formal analysis; Investigation; Methodology; Writing—original draft; Writing—review and editing. **Eli G Cytrynbaum**: Formal analysis; Investigation; Methodology; Writing—original draft; Writing—review and editing. **Jamie M Ahrens**: Formal analysis; Investigation. **Megan N McClean**: Supervision; Methodology. **Audrey P Gasch**: Conceptualization; Formal analysis; Supervision; Funding acquisition; Investigation; Methodology; Writing—original draft; Project administration; Writing—review and editing.

Source data underlying figure panels in this paper may have individual authorship assigned. Where available, figure panel/source data authorship is listed in the following database record: biostudies:S-SCDT-10_1038-S44318-026-00727-x.

## Disclosure and competing interests statement

The authors declare no competing interests.

# Expanded View Figures

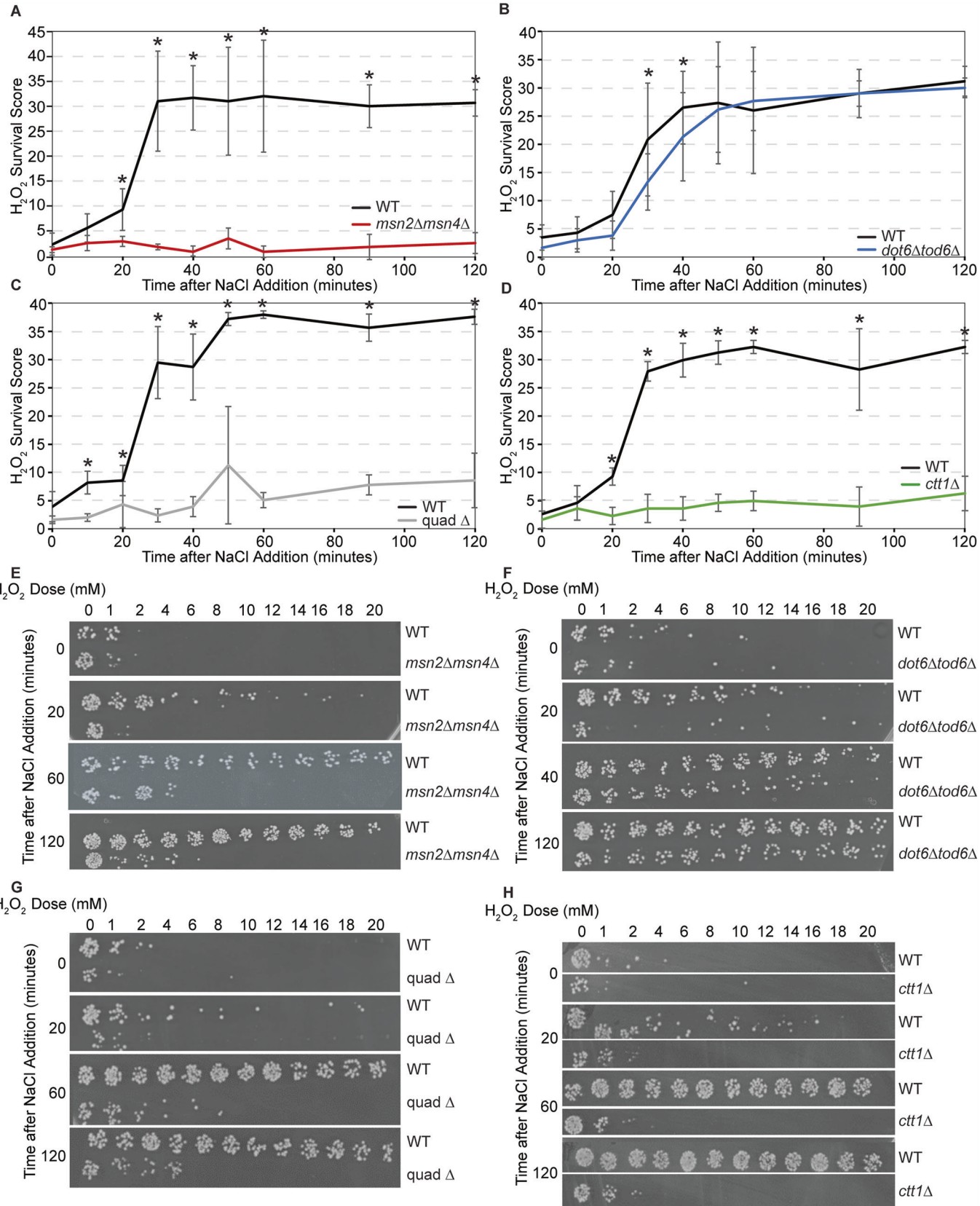

**Figure EV1.   Msn2/4 and Dot6/Tod6 responses are important for acquired stress resistance.**

Related to Fig. 2. (**A–D**) The average change in $H_2O_2$ survival scores for wild-type (black), *msn2Δmsn4Δ* (red), *dot6Δtod6Δ* (blue), quad Δ (gray), and *ctt1Δ* (green) cells $+/- 1$ standard deviation, as described in Fig. 2 ($n = 3$ except for (**C**) where $n = 6$). Colored lines are as shown in Fig. 2, along with the paired wild-type culture done side-by-side with each mutant. (**E–H**) Representative images of cell viability across doses of $H_2O_2$ and time used to calculate $H_2O_2$ survival scores. Paired *t* test, exact *P* values are available in Dataset EV1. Source data are available online for this figure.

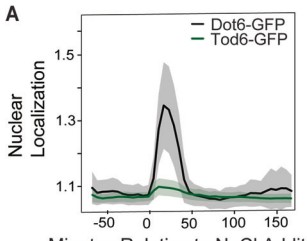

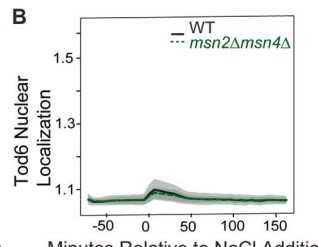

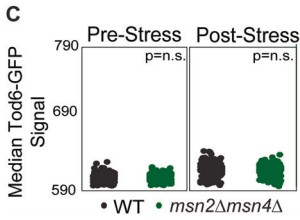

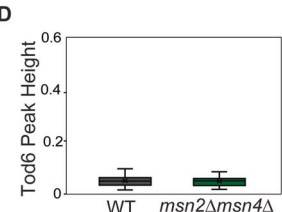

**Figure EV2.   Loss of *MSN2/4* leads does not affect Tod6 abundance and nuclear localization.**

*Y*-axes are set to scales comparable to Fig. 3B–D for comparison with Dot6. (**A**) Dot6-GFP in an iRFP-marked strain was mixed with cells expressing Tod6-GFP, both expressed from the native genomic loci, and nuclear translocation of each factor was scored before and after NaCl treatment in the same device ($n = 132$ cells: WT, 56, *msn2Δmsn4Δ*, 76). The results confirm that Dot6-GFP shows strong nuclear relocalization in response to NaCl, whereas Tod6-GFP shows only a weak change under the conditions used here. (**B**) The population average of Tod6-GFP nuclear/cytoplasmic ratio in wild-type (black line) and *msn2Δmsn4Δ* (dashed green line) cells $+/-$ one standard deviation (shading). The two strains are statistically indistinguishable ($P > 0.1$) (**C**) Distribution of median Tod6-GFP signal scored before (0–72 min) or after (120–216 min) NaCl treatment (see "Methods" for details) for WT and *msn2Δmsn4Δ* cells; $P = 0.138$ (left panel), 0.817 (right panel), Wilcoxon rank-sum test; n.s. = not significant. (**D**) Distribution of Tod6-GFP acute-stress peak heights across 132 cells, see "Methods". $P = 0.896$, Wilcoxon rank-sum test. Boxplots show median (line) and 0.25 and 0.75 quartiles (box), with whiskers extending from minimum to maximum excluding outliers (circles) that are <0.25 quartile–1.5× interquartile range or >0.75 quartile–1.5× interquartile range. This figure shows that cells lacking *MSN2/MSN4* show no difference in Tod6-GFP levels or activation compared to wild-type. Source data are available online for this figure.

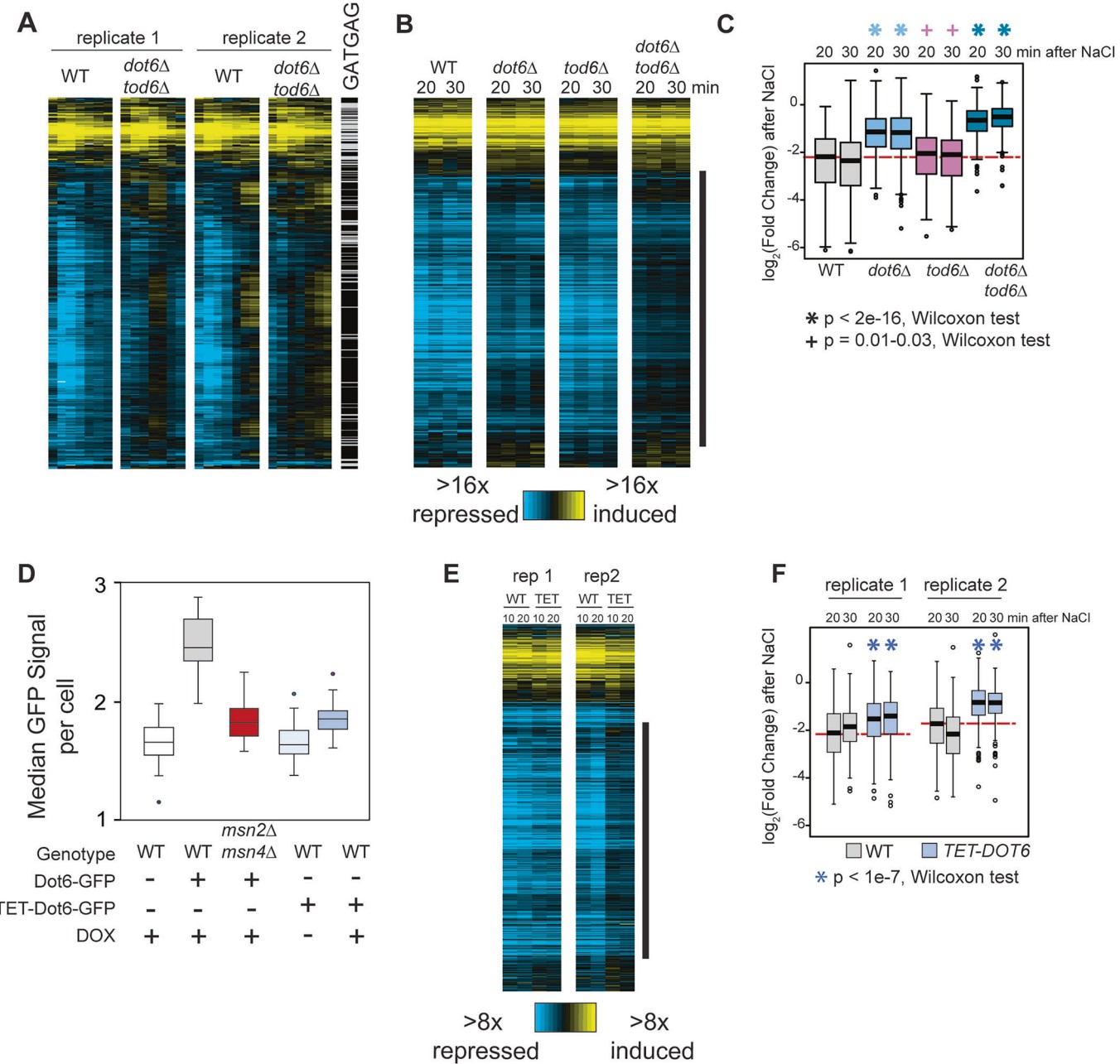

**Figure EV3.  Dot6 contributes to gene repression more than Tod6 during NaCl stress.**

(A) Log₂(fold change) of 489 genes significantly affected by *dot6Δtod6Δ* deletion in the time course (10–90 min), as shown in Fig. 5A, see Methods for details. Presence of GATGAG sequences within 500 bp upstream of each gene is indicated with a black line next to each gene. (B) Expression of the same genes in single-gene deletion of *DOT6* or *TOD6* at the peak of the expression response (20 and 30 min). (C) Boxplots of replicate-averaged data from the highlighted cluster from (B) indicated with the black bar (379 genes). The median repression of wild-type at 20 min is indicated with a red-dashed line. Note the *dot6Δ* mutant has a partial repression defect that is highly statistically significant compared to the wild-type ($P < 2e{-}16$, Wilcoxon rank-sum test), whereas the *tod6Δ* mutant shows only subtle repression differences compared to the wild-type with marginal significance ($P = 0.01{-}0.03$, Wilcoxon rank-sum test). (D) The distribution of median Dot6-GFP levels in individual cells of indicated strains ($n = 50$ each strain) exposed to 50 µg/mL doxycycline. This dose produces Dot6-GFP protein levels in the *TET-DOT6* strain comparable to *msn2Δmsn4Δ* cells. (E) As shown in (A) but for wild-type cells paired with TET-inducible *DOT6* both grown side-by-side in 50 µg/mL doxycycline before and after NaCl treatment. (F) Box-plot of data from (E) for 379 genes. All four *TET-DOT6* samples show weaker repression compared to their paired wild-type sample ($P < 1e{-}7$, Wilcoxon rank-sum test). Boxplots show median (line) and 0.25 and 0.75 quartiles (box), with whiskers extending from minimum to maximum excluding outliers (circles) that are <0.25 quartile–1.5× interquartile range or >0.75 quartile–1.5× interquartile range. Exact *P* values provided in Dataset EV1. Source data are available online for this figure.

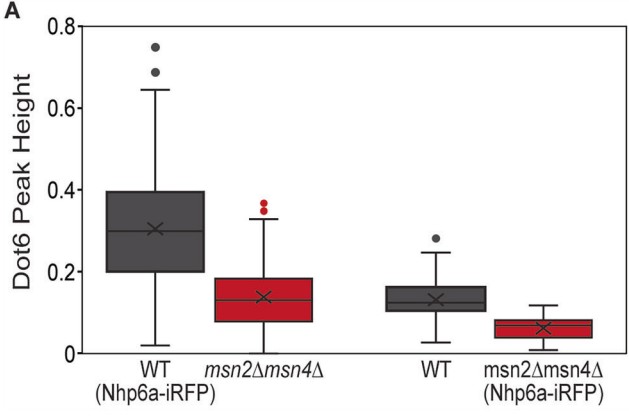

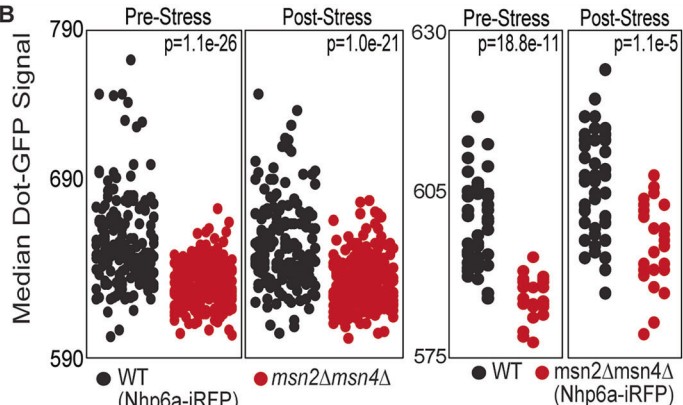

**Figure EV4.  Msn2/4 effects on Dot6 activity are not an artifact of iRFP.**

One consideration was if iRFP expressed in one strain affected GFP signal within the same strain. To ensure that our results in Fig. 4 were not due to unanticipated effects of iRFP, we generated a new set of strains in which the *msn2Δmsn4Δ* cells, rather than the wild-type, carried the distinguishing iRFP signal. We found that the trends discussed in the main text were not affected by which strain carried the iRFP marker. (A) Distribution of Dot6 acute stress peak height across wild-type and *msn2Δmsn4Δ* cells when the wild-type carried expressed Nhp6a-iRFP (left, WT $n = 173$, *msn2Δmsn4Δ* $n = 270$) or when the *msn2Δmsn4Δ* strain expressed Nhp6a-iRFP (right, WT $n = 49$, *msn2Δmsn4Δ* $n = 29$). Despite some differences in signal for experiments done with different laser power, *msn2Δmsn4Δ* cells showed weaker Dot6-GFP nuclear translocation signal in both sets of experiments ($P = 8.8e-11$, left, $P = 1.1e-5$, right, Wilcoxon rank-sum test). (B) Distribution of median Dot6-GFP signal within the cells, scored before (0–72 min) or after (120–216 min) NaCl treatment for wild-type cells and *msn2Δmsn4Δ* cells as described in (A). p, Wilcoxon rank-sum test. Boxplots show median (line) and 0.25 and 0.75 quartiles (box), with whiskers extending from minimum to maximum excluding outliers (circles) that are <0.25 quartile–1.5× interquartile range or >0.75 quartile–1.5× interquartile range. Source data are available online for this figure.

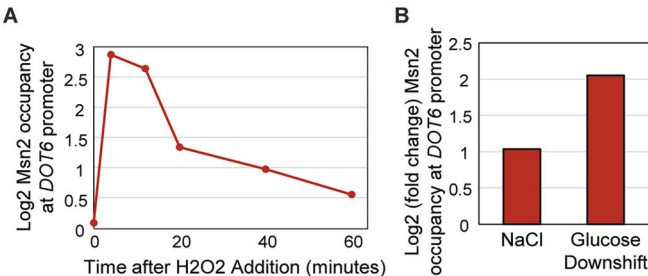

**Figure EV5.  Msn2 binds *DOT6* promoter under various stresses.**

(A) Log$_2$ enrichment of Msn2 occupancy relative to the whole-cell extract at the *DOT6* promoter in response to 0.4 mM H$_2$O$_2$ (from Huebert et al, 2012). (B) Log$_2$ (fold change) of Msn2 occupancy at the *DOT6* promoter (ranging from 0 to −1000 bp from Ni et al and +/− 250 bp surrounding the Msn2 STRE element in the *DOT6* promoter) in response to 30 min of 0.6 M NaCl (left, from Ni et al, 2009) or 20 min after shift from glucose to glycerol (right, from Elfving et al, 2014) compared to the corresponding measurement in unstressed cells.

