## [Peer Review File · The EMBO Journal]

The importance of regulated resource reallocation during dynamic environmental shifts in yeast

Rachel Kocik, Eli Cytrynbaum, Jamie Ahrens, Megan McClean, and Audrey Gasch

Corresponding author(s): Audrey Gasch (agasch@wisc.edu)

Review Timeline:

Submission Date:	2nd Dec 24
Pre-decision Consultation:	20th Jan 25
Authors' Tentative Response:	31st Jan 25
Editorial Decision:	10th Feb 25
Revision Received:	15th Dec 25
Editorial Decision:	2nd Feb 26
Revision Received:	6th Feb 26
Accepted:	11th Feb 26

Editor: Hartmut Vodermaier

Transaction Report:

Dear Audrey,

Thank you for submitting your manuscript on resource reallocation during yeast stress responses to The EMBO Journal. I apologize for the delay in getting back to you with a decision, as it had been difficult to assign appropriate reviewers in the busy period around the end of the year. We have now received the below-copied reports from three expert reviewers. As you will see, while all referees acknowledge the interest and potential importance of this work, they nevertheless retain significant reservations at this stage. These include (but are not limited to) unjustified novelty statements, insufficient descriptions of experiments or conclusions, missing direct evidence for Msn2/4 binding and function at the Dot6 promoter, as well as the need for validation in other physiological stress-settings, and for better supporting important aspects of the proposed model.

I will not go through all referee points in detail here, but it is apparent that to become a stronger candidate for EMBO Journal publication, the study would not only require decisive clarification of specific concerns, but also certain extension. Since it is not clear if and how this might be achieved during a single, regular round of major revision, I would at this point invite you to carefully consider the referee reports and to prepare a tentative point-by-point response, detailing how you would envision addressing the key concerns of the referees in case you should be given the opportunity to revise this work for The EMBO Journal. Based on this revision plan (which I may share and discuss with some of the referees) and possible follow-up discussion, we could then determine whether it would be warranted and promising to invite a major revision for The EMBO Journal. It would be great if you could get back to me with such a revision plan by early next week.

Looking forward to hearing from you,

Best regards,

Hartmut

Referee #1 (Report for Author)

Across species, cellular stress responses are often associated with the modulation of transcriptional and translational capacity, which appears to be important for the reallocation of cellular energy and resources from cell growth/proliferation to selective responses that confer stress tolerance. However, the molecular details of this reallocation are not fully understood. Kocik and Gash addressed this question utilizing the yeast environmental stress response (ESR) and high salt stress as a model system. They identified the stress-responsive transcription factor Msn2/Msn4 as a key coordinator of the ESR, regulating the induction of defense genes and the repression of growth-related genes in response to stress conditions. The Msn2/4-dependent response appears to slow cell growth rate under salt stress conditions and is required for the acquisition of stress tolerance to secondary hydrogen peroxide insult after the salt stress. The authors also found that Msn2/4 is an upstream regulator of the Dot6/Tod6-dependent transcriptional repression of growth-promoting genes through Dot6 expression and nuclear translocation. The findings here seem to be an important addition to the model of the ESR that the authors have established through their previous work. Overall, the single cell analysis and RNA-seq data with yeast genetics are solid. However, the proposed mechanisms for regulation of rESR genes by Msn2/4 are less convincing. To further support their conclusion, concerns listed below should be addressed.

Major points.

1) Since it is hard to interpret the difference between the strains in the growth curve in the acute stress response phase (30-60 min after NaCl) in Figure 1B, the authors should show the bar graphs with statistical analysis for this early phase as shown in Figure 1C for the acclimation phase. Also, there seems to be no clear description of why the quad mutant exhibited the phenocopy of the Msn2/4 mutant only in the late phase but not in the early phase.

2) Figure 4, it would be unclear to readers not familiar with the yeast gene expression system (like this reviewer) whether exogenously expressed Dot6-GFP is under the control of the endogenous Dot6 gene promoter or some other promoter which is co-introduced with Dot6-GFP. If the latter is the case, promoter specific transcriptional regulation of Dot6-GFP is unlikely to be the cause of the reduction of Dot6-GFP expression in the MSN2/4 mutant, since the reporter expression should be independent of endogenous Dot6 gene expression. Is it possible that Msn2/4 regulates Dot6-GFP protein levels by regulating the protein stability of Dot6-GFP or the translation rate of Dot6-GFP?

3) Related to the above question, the Msn2/4 mutant exhibited weaker Dot6 nuclear translocation (Figure 4D). Given the subcellular localization of Dot6 is regulated by phosphorylation, is it possible that the MSN2/4 affects phosphorylation of Dot6, e.g.,

through the activity of PKA or mTOR signaling?

4) Figure 5, it is unclear whether MSN2 and MSN4 binding to each rESR gene promoter is experimentally confirmed or just speculated from the transcription factor binding site. It seems that the authors used the YEASTRACT+ database as described in Methods, but a more detailed explanation of how the MSN2/4 binding is determined should be added in the main text.

5) Overall, experimental evidence for direct binding of MSN2/4 to the Dot6 promoter and the effect of MSN2/4 on Dot6 mRNA expression is weak. The authors only showed the data from previous works with different conditions (Fig. EV3). Experiments to show the binding of MSN2 to the Dot6 promoter, such as Chromatin-IP in the current conditions, would be needed. Also, changes in Dot6 mRNA levels do not appear to be significant (Fig. 6).

Immunoblotting of endogenous Dot6 would support the argument that the reduction in Dot6 mRNA levels in the MSN2/4 mutant does reduce functional Dot6 protein in the cell.

6) MSN2/4 mutant showed reduced Dot6 expression/activity (Fig. 4) and de-repression of rESR genes that partially overlap with the Dot6/Tod6 target. These observations seem to contradict the increased post stress growth rate of Msn2/4 mutant (Fig. 2). Does this mean that the cost of iESR induction outweighs the saved cost of rESR repression? This point could be discussed further.

Minor points.

1) In HIGHLIGHTS, the authors make the point "Cells lacking repressors of growth promoting genes have a slower post-stress growth rate". As the authors mention in the text, this is the finding of their previous paper (Bergen et al. eLife 2022) and should not be highlighted in this work.

2) Figure 7, according to the authors' conclusion, the dashed line from Msn2/4 to the promoter for ribosome and growth gene should be a negative regulation ("T shape" allow).

3) Page 21, line 18, there may be typo (... , slows growth | the absence....).

Referee #2 (Report for Author)

Msn2 and Dot6/Tod6 have been the subject of numerous studies demonstrating their importance in stress adaptation and recovery from quiescence. In this study, the authors demonstrate that induction of genes targeted by Msn2 (iESR) but also the inhibition of growth promoting genes (rESR) by Dot6/Tod6 are key points that determine both the future capacity to grow and to resist to future stress. In support of this conclusion, the authors show that deletion of Msn2 can increase growth rate during recovery at the expense of future stress resistance, and consequently Msn2-deficient cells grow significantly faster than wild-type cells during salt acclimation, but have defects in stress resistance (peroxide

resistance). The authors propose that Msn2 is not only required for the expression of key stress-related genes involved in stress adaptation, but also for the repression of growth-promoting genes through direct regulation of Dot6 levels and through indirect/direct action of Msn2.

The observation that Msn2 deleted cells have higher growth rate during recovery after stress, in contrast to Dot6/Tod6 mutants, and a defect in peroxide resistance represents the major interest of this study presented in Figure 1, 2. Furthermore, the fact that both Msn2 and Dot6/Tod6 deletion have no growth phenotype in exponential growth phase avoids misinterpretation bias and helps to support their conclusions.

The other parts of this work (Figures 3, 4, 5, 6) add no or only a modest conceptual advance without important mechanistic insights and some conclusions are less well supported by the data presented (mainly the role of Msn2 in repressing rESR genes). Importantly, some data are redundant with previously published studies, e.g. the authors already published in a previous study that deletion of Dot6/Tod6 reduces growth rate after stress (although they presented this data as new data in HIGHLIGHTS "Cells lacking repressors of growth-promoting genes have a slower post-stress growth rate"). Moreover, it has already been suggested that Msn2 has an (indirect) effect on growth-promoting genes, for reasons that remain unclear and were not really clarified in this study. As described in more detail below, new experiments testing mutants under different growth/stress conditions need to be included in the first part of this study to improve the interest of the manuscript, as well as a more cautious interpretation of the genome-wide data, which is mainly based on Venn diagram analysis and the criteria chosen by the authors to define Msn2/4 target genes. The model also needs to be modified, as the data in this manuscript never showed that Msn2 regulates the DOT6 paralog, TOD6 (see more details in major concern 4).

Major concerns:

1/ To increase the impact of this study for the wider readership of The EMBO Journal, and because the author uses the general term "yeast stress response" in the title, it would be important to test not only salt stress but also growth recovery after other physiological stress conditions (pH, heat shock, starvation) with the experimental set-up used in Figure 1-2. It would also be important to test single Dot6 and Tod6 deleted cells as well as cells mutated for the Msn2 binding site at the Dot6 promoter for the reasons explained below in points 3-4. It would also be interesting to test other mutant strains such as the Msn26A allele described in Elfving et al, 2014 or deletion of STB3 or XBP1, since these two factors are involved as Dot6/Tod6 in recovery from quiescence and in repression of growth or cell cycle transcriptional program by promoting recruitment of HDAC Rpd3 at specific set of

promoters (PMID: 26300265). Interestingly, a previous study has already shown that Msn2 also regulates and binds to the promoter of the repressor XBP1 (Elfbing et al, 2014). These simple experiments will be very useful to extend and support the conclusion drawn in this study regarding the interplay between Msn2 and Dot6/Tod6 during the yeast stress response.

2/ In Figure 2, the effect of the delay in the establishment of peroxide tolerance appears weak and curiously not significant at 20 minutes. Moreover, in the extended view of Figure 1 (panel C), it appears that maximum tolerance is not established in some wild type until 60 minutes, similar to what was described in some older studies and for the double mutant in the main figures 2. Would the authors like to comment on this point? Furthermore, it's clear that the figure in the extended view of Figure 1 (E,F,G,H) does not follow the EMBO guidelines for figures, as it seems that a large number of drop tests from different plates have been fused (mainly in panel H). These data are important in this study and the authors need to show the raw data of the different replicates for these drop test in Expanded View Figure 1-H and also the raw data that allow to make the figure shown in Figure 2.

3/ At the beginning of the paragraph on page 15, the authors say "A remaining question is if the weaker Dot6 activation in the *msn2Δmsn4Δ* mutant leads to weaker repression of Dot6 target genes. To directly address this question, we followed dynamic changes to the transcriptome before and in ten-minute increments after salt stress in the different strains. We first identified 489 genes whose response to salt treatment was altered in the *dot6Δtod6Δ* strain compared to wild-type (FDR < 0.05) in at least two timepoints (see Methods)."

The data presented in this paper only show that Msn2 is a DOT6 regulator and not TOD6. Therefore, to study the importance of Msn2 in regulating Dot6 levels, the authors can't only use the double mutant Dot6/Tod6 for the experiments in Figure 1, 2 and for the transcriptomic analysis in Figure 5, but they need to also test also single DOT6 mutant and a strain mutated for the binding site of Msn2 at the DOT6 promoter. These experiments are essential for understanding the importance of Msn2 on Dot6 level regulation.

4/ Following on from the previous comments, it does not seem correct to show in the final model (Figure 6) that Msn2 regulates both Dot6 and Tod6 for two main reasons. Firstly, there are no data in this study showing that Msn2 regulates Tod6 and secondly, the authors ignore that the major transcriptional regulator of growth promoting genes, Sfp1, has already been shown to directly regulate Tod6 expression (PMID: 30804227). The latter point is particularly interesting as it suggests that the two stress sensitive TFs, Sfp1 and Msn2, regulate rESR and iESR respectively, and the two paralogs Tod6 and Dot6. Such regulation

would maintain a pool of Dot6 and Tod6 in a wide range of growth conditions. In addition, it might be interesting to plot the mRNA levels of Dot6 and Tod6 on the same graph in Figure 7.

5/ In Figure 4, the authors show that Msn2 has an effect on Dot6 levels and localisation, but given that the two paralogs are thought to have similar function, it seems important to measure Tod6 localisation using a similar experimental setup. In addition, Msn2 is unlikely to affect the level of Tod6, unlike Dot6, and for this reason it will be easier to examine changes in Tod6 localisation by microscopy than Dot6 in Msn2-deficient cells.

6/ Finally, I have probably the most important concerns with the analysis and conclusions in Figure 5-6.

My first concern relates to analysis of Msn2 targets. The authors use YeasTRACT, which is a really useful tool, but some large-scale data have to be taken with cautious, as do many databases that pool genome-wide data from large-scale studies of varying quality. Indeed, according to Yeastract, Msn2/4 directly bind to more than 2100 of the 5500 RNAPII promoters, but it seems very unlikely that Msn2/4 bind to more than a third of the coding gene promoters. Similarly, by using a very poor stringent cut-off to identify promoters with Msn2 binding site (they select all genes containing a CCCCT STRE 1000bp upstream TSS), they identified more than 2000 genes containing this motif. Furthermore, a recent study shows that the presence of the STRE motif is a poor predictor of Msn2 binding in vivo (PMID: 32553192, PMID: 38109289), as the disordered region associated with Msn2 mainly defines its binding to promoters containing an STRE motif in living cells, suggesting that the presence of the STRE motif does not accurately define Msn2 binding. In this sense, in a study aimed at defining the target of Msn2, it was reported that "while approximately 11,500 STRE sequences are present in yeast genome, with 3160 lying in the promoters of almost 2000 genes, Msn2 fails to bind most of these sites." (Elfving et al, 2014, PMID: 24598258). In the latter study, Elfving and colleagues defined a list of 268 targets (including Dot6) using various analyses including RNAPII and Msn2 CHIP-seq in different mutants and nucleosome profiling, which is likely to be more appropriate than the list used in this manuscript.

Nevertheless, the authors use these poorly defined categories to analyze their transcriptomic data, mainly by defining subcategories using a Venn diagram, and the conclusion appears confusing. From my point of view, this Venn diagram analysis needs to be repeated with a more precise definition of the promoter and the number of STRE motifs per promoter, knowing that most studies that carefully assess Msn2/4 binding suggest that Msn2/4 binds to roughly 300 RNAPII promoters.

Furthermore, if the authors want to show a direct effect on transcription of Msn2, they need

to measure RNA synthesis or RNAPII binding and not mRNA accumulation.

7/ I also have major concerns about redundant conclusions between the Kocik et al. manuscript and other older studies.

-The fact that the absence of Dot6/Tod6 leads to slower acclimation to salt has already been shown (Bergen et al. 2022, PMID: 36350693) and can't be presented as new data in HIGHLIGHT.

-Similarly, it has already been shown in previous studies that deletion of DOT6/TOD6 prevent proper repression of rESR has been shown in numerous studies (PMID: 31395866, PMID: 21730963, PMID: 35310337, among others) and may not be present as a new result: "These results indicate a primary role for Dot6/Tod6 in repression of their RiBi target genes.", this sentence must be accompanied by references to other studies.

- Finally, Elfving and colleagues (PMID: 24598258), in an elegant study aimed at defining the role of Msn2/4 during stress, already show that "Msn2 promotes both transcriptional activation and transcriptional repression" and propose "We do find that many of the genes repressed upon activation of Msn2 are highly enriched for those involved in ribosome biogenesis. However, few of these genes are bound by Msn2, at least under conditions of nutrient downshift. Rather, we observed that Msn2 activates transcription of DOT1, which encodes a repressor of ribosome biogenesis genes (see Supplementary Tables S1 and S2). Moreover, we find that Msn2 binds to and activates transcription of XBP1, which encodes a repressor of a number of genes required for cell cycle progression. Accordingly, Msn2, while a primary purveyor of the ESR, may indirectly repress the growth-associated genes encompassed in the ESR."

This conclusion is very similar to the one made in the present manuscript (but Elfving and colleagues made a mistake in the name of DOT1 instead of DOT6 (in agreement with this point, DOT1 is not present in their Supplementary Table S1 showing the list of promoters bound by Msn2, in contrast to DOT6).

In conclusion, the data presented in Figures 5 and 6 mainly use old data and the conclusions are redundant with older studies.

If this paper is accepted for revision, the authors must at least improve the quality of their analysis, try to validate the binding of Msn2 at the Dot6 promoter by ChIP of Msn2 in a strain mutated for the Msn2 motif at the DOT6 promoter, and more carefully report previous observations already made by other laboratories.

Minor comments :

8/ The authors propose that the timing of the delay correlates with delayed Ctt1 protein production in the dot6 Δ tod6 Δ mutant. In a previous study, the authors showed that increasing Ctt1 mRNA levels using a Gal-estradiol-inducible promoter is sufficient to establish peroxide tolerance (PMID: 22851651), but that the absence of Dot6/Tod6 also

increases Ctt1 mRNA levels, but not sufficiently to increase total Ctt1 protein levels (Bergen et al. 2022, PMID: 36350693). To confirm their suggestion that the delay in the acquisition of peroxide tolerance is dependent on Ctt1, it may be interesting to use this inducible Gal system to artificially increase Ctt1 mRNA and protein levels. It is also possible that this delay is independent of Ctt1 accumulation.

9/ Perhaps it's my fault and these data are available, but I can't find an Excel table reporting the source data of Figure 5, allowing me to see which gene of all RNAPII-transcribed genes was associated as an Msn2 target, STRE motif, GATGAG motif according to authors.

10/ The figure 4 and one part of associated text have minor interest and could be remove in supplemental Figure.

11/ Probably beyond the scope of this study, but it would be interesting for the authors to discuss stufy from the Barkai lab and/or to use the genetic tool developed in Kafri et al, 2016 to study growth rate recovery during adaptation and acquisition of peroxide resistance. Kafri and colleagues use different strains expressing one to ~20 copies of mCherry under the control of a strong promoter to challenge the physiological effect of overexpression of an inert protein or unstable mRNA (cells forced to transcribe unstable DAmP transcripts). According to the model and the result in Figure 2, it would be expected that such overexpression would also delay the acquisition of resistance. Furthermore, these constructions will also help to understand whether the effect is related to transcription and/or translation. Perhaps the author can at least discuss this point.

Referee #3 (Report for Author)

This study investigates how yeast cells relocate transcriptional and translational resources among gene groups during environmental stress. The authors explore the interplay between the transcription factors Msn2/Msn4 and Dot6/Tod6, concluding that Msn2/Msn4 are the primary regulators in resource redirection. Beyond activating stress-response genes, Msn2/Msn4 play a pivotal role in repressing growth-related genes via three mechanisms: 1) Regulating the transcription of DOT6/TOD6. 2) Promoting the nuclear localization of Dot6/Tod6. 3) Regulating the transcription of a subset of rESR genes. While the proposed modeling is intriguing, the supporting evidence is insufficient, and many underlying mechanisms remain unclear. Additional experiments are necessary to solidify these conclusions.

Specific Comments:

1) In Figure 6, the authors show that *msn2Dmsn4D* mutants have lower DOT6 mRNA levels compared to wild-type cells. However, time-course data indicate that Msn2/Msn4 are only minor contributors, as both mutant and wild-type cells upregulate DOT6 mRNA to comparable levels post-stress (60 min), albeit with a slight delay in the mutants. The role of Msn2/Msn4 in regulating DOT6 expression is overstated.

2) In Figure 4, the reduction of Dot6 protein levels in *msn2Dmsn4D* mutants before stress is insufficiently explained. The authors attribute this to the absence of stochastic bursts of ESR induction, but single-cell data show consistently low Dot6 abundance across nearly all mutant cells before stress (Figure 4A). This observation is difficult to reconcile with stochastic bursts. A potential experiment to address this would involve mutating the Msn2/Msn4 binding elements in the DOT6 promoter and assessing whether Dot6 protein levels remain low under no-stress conditions.

3) The presentation of results in Figure 5 is unclear. For example, the sentence "Two thirds (68%) of these genes contain upstream Dot6/Tod6 binding elements...and 249 genes showed a repression defect..." raises questions: How many genes overlap between these two groups? How many of these genes also exhibit Msn2/Msn4 binding? How many Msn2/Msn4 binding genes lack upstream STRE elements? The data and descriptions should be reorganized to allow clear comparisons of specific features across gene groups.

4) Numerous paragraphs in the Results section end with "see Discussion," but it is challenging to locate where these data are discussed. It would be more effective to integrate relevant discussions directly within the results paragraphs.

Minor Comments:

1) Please add page numbers and line numbers to the manuscript for easier reference.

2) Do the authors have any hypotheses regarding how Msn2/Msn4 binding represses rESR genes? Including such speculation would strengthen the interpretation of results.

3) In the sentence "conversely, ppGpp over-production, leading to stronger... slow growth I the absence of stress...", it should be "in the absence of stress".

The original comments from the reviewers are listed in black text with our response in italicized blue text.

Referee #1 (Report for Author)

Across species, cellular stress responses are often associated with the modulation of transcriptional and translational capacity, which appears to be important for the reallocation of cellular energy and resources from cell growth/proliferation to selective responses that confer stress tolerance. However, the molecular details of this reallocation are not fully understood. Kocik and Gash addressed this question utilizing the yeast environmental stress response (ESR) and high salt stress as a model system. They identified the stress-responsive transcription factor Msn2/Msn4 as a key coordinator of the ESR, regulating the induction of defense genes and the repression of growth-related genes in response to stress conditions. The Msn2/4-dependent response appears to slow cell growth rate under salt stress conditions and is required for the acquisition of stress tolerance to secondary hydrogen peroxide insult after the salt stress. The authors also found that Msn2/4 is an upstream regulator of the Dot6/Tod6-dependent transcriptional repression of growth-promoting genes through Dot6 expression and nuclear translocation. The findings here seem to be an important addition to the model of the ESR that the authors have established through their previous work. Overall, the single cell analysis and RNA-seq data with yeast genetics are solid. However, the proposed mechanisms for regulation of rESR genes by Msn2/4 are less convincing. To further support their conclusion, concerns listed below should be addressed.

We appreciate this reviewer's assessment that this work is an "important addition" to the field and that the work is solid. In the response below, we have clarified some points of confusion and outline experiments that we will add to the manuscript to address their concerns.

Major points.

1) Since it is hard to interpret the difference between the strains in the growth curve in the acute stress response phase (30-60 min after NaCl) in Figure 1B, the authors should show the bar graphs with statistical analysis for this early phase as shown in Figure 1C for the acclimation phase. Also, there seems to be no clear description of why the quad mutant exhibited the phenocopy of the Msn2/4 mutant only in the late phase but not in the early phase.

We will add the growth rates calculated in the immediate phase after NaCl addition (and other stresses as outlined below) to Fig 1. The quad mutant growth rate is somewhat variable immediately after NaCl, and in the single example shown in Fig 1A the strain seems to grow slightly slower immediately after NaCl addition. We will provide a deeper analysis in the immediate after effects of each stress transition. It is possible that the quad mutant, which lacks Dot6/Tod6-dependent resource allocation, retains a burden during the highly active phase immediately after NaCl stress when other transcription factors are also at play. But the important result is that loss of the costly Msn2/4 response in the quad mutant complements loss of the Dot6/Tod6 response, at least after cells have resumed growth.

2) Figure 4, it would be unclear to readers not familiar with the yeast gene expression system (like this reviewer) whether exogenously expressed Dot6-GFP is under the control of the endogenous Dot6 gene promoter or some other promoter which is co-introduced with Dot6-GFP. If the latter is the case, promoter specific transcriptional regulation of Dot6-GFP is unlikely to be the cause of the reduction of Dot6-GFP expression in the MSN2/4 mutant, since the reporter expression should be independent of endogenous Dot6 gene expression. Is it possible

that Msn2/4 regulates Dot6-GFP protein levels by regulating the protein stability of Dot6-GFP or the translation rate of Dot6-GFP?

We apologize for the confusion: the DOT6-GFP coding sequence was integrated into the yeast genome and is expressed from the native DOT6 promoter. While it is possible that Msn2/4 affect other aspects of the stress response that could influence Dot6 protein turnover, the most parsimonious model is one of transcriptional regulation: Msn2/4 are well known transcriptional activators that bind the DOT6 promoter during NaCl and other stresses (EV3 and included citations) and are required to counteract DOT6 mRNA repression during NaCl and other stresses (Fig 6). Experiments to mutate the MSN2/4 binding site in the DOT6 promoter are underway and will test the consequences of losing the Msn2/4 binding site upstream of DOT6 on Dot6 protein abundance, growth and transcriptional repression.

3) Related to the above question, the Msn2/4 mutant exhibited weaker Dot6 nuclear translocation (Figure 4D). Given the subcellular localization of Dot6 is regulated by phosphorylation, is it possible that the MSN2/4 affects phosphorylation of Dot6, e.g., through the activity of PKA or mTOR signaling?

This is an important and interesting topic that we can expand on in a revised Discussion. Indeed, Msn2/4 induce the expression of other stress-responsive regulators, including both positive and negative regulators of PKA signaling (Gasch et al. 2000, Segal et al. 2003, and likely others). Furthermore, our past work shows that Msn2 physically interact with the stress-activated phosphodiesterase Pde2 that regulates PKA during stress (MacGilvray et al. 2018). We are investigating effects of PDE2 deletion on Msn2 and Dot6 nuclear localization through other work that we hope to save for a different manuscript; notably, deletion of PDE2 does not affect Dot6 protein levels. Nonetheless, we can expand on this possibility to address this point and points raised by Reviewer #2 below.

4) Figure 5, it is unclear whether MSN2 and MSN4 binding to each rESR gene promoter is experimentally confirmed or just speculated from the transcription factor binding site. It seems that the authors used the YEASTRACT+ database as described in Methods, but a more detailed explanation of how the MSN2/4 binding is determined should be added in the main text.

We apologize for the confusion. The heat maps in the figure shows genes with statistically significant expression defects in each mutant, along with a graphical representation of genes that have confirmed Msn2 and/or Msn4 upstream binding (“Msn2” or “Msn4” columns in the figure), as well as genes that harbor upstream binding sites (annotated as “STRE” for the Msn2/4 binding site and ‘GATGAG’ for Dot6/Tod6). The STRE elements were not used to define any groups; rather, we cite in the text that the group of genes that we defined whose induction requires Msn2/4 is enriched with statistical significance for genes with upstream elements. In contrast, the group genes whose repression requires Msn2/4 are also enriched for genes with confirmed Msn2/4 binding – but this group is statistically significantly under-enriched for genes harboring the STRE element. Elucidating how upstream binding of Msn2/4 to rESR promoters without STRE elements, outside of those regulated by Dot6, will require future work and is beyond the scope of this manuscript; however, it suggests that Msn2/4 are playing multifaceted roles to coordinate rESR repression. We will include these points in an expanded Discussion.

Given that both this reviewer and Reviewer 2 were confused by the expression and binding data as presented, we are happy to provide a deeper analysis of which genes depend on Msn2/4 for expression, which have upstream Msn2 versus Msn4 binding and for which stresses, and which

genes have or do not have STRE binding elements within 500 bp upstream of the gene start site.

5) Overall, experimental evidence for direct binding of MSN2/4 to the Dot6 promoter and the effect of MSN2/4 on Dot6 mRNA expression is weak. The authors only showed the data from previous works with different conditions (Fig. EV3). Experiments to show the binding of MSN2 to the Dot6 promoter, such as Chromatin-IP in the current conditions, would be needed. Also, changes in Dot6 mRNA levels do not appear to be significant (Fig. 6). Immunoblotting of endogenous Dot6 would support the argument that the reduction in Dot6 mRNA levels in the MSN2/4 mutant does reduce functional Dot6 protein in the cell.

We show quantitative data from ours and other studies, including a study by (Ni et al. 2009) that ChIP'd Msn2 during NaCl treatment with very similar conditions used here. Since the vast majority of Msn2/4 binding sites are consistent across stresses (Elfving et al. 2014 and others), and since one of these studies has ChIP'd Msn2 during NaCl as studied here, we do not believe that adding more ChIP-seq data would add to the paper.

The lower DOT6 mRNA in the msn2/4Δ mutant responding to NaCl studied here (Fig 6) is indeed statistically significant and was identified because it was on our list of affected genes; we will add asterisks to the figure to avoid confusion.

Unfortunately, we do not have antibodies against Dot6 protein so we rely on quantifying epitope-tagged Dot6 expressed from the native genomic locus driven by the natural DOT6 promoter. Cells lacking MSN2/4 have significantly lower DOT6 mRNA and Dot6 protein after NaCl treatment. We do agree that an important experiment is to test how altered Dot6 abundance affects growth and repression after stress, and these experiments are underway to test mutant strains i) lacking the Msn2/4 binding site upstream of DOT6 (or DOT6-GFP) or ii) for which DOT6 transcription can be controlled with a titratable Tet promoter. These experiments will directly test the effect of Msn2/4 regulation and DOT6 expression and stress phenotypes.

6) MSN2/4 mutant showed reduced Dot6 expression/activity (Fig. 4) and de-repression of rESR genes that partially overlap with the Dot6/Tod6 target. These observations seem to contradict the increased post stress growth rate of Msn2/4 mutant (Fig. 2). Does this mean that the cost of iESR induction outweighs the saved cost of rESR repression? This point could be discussed further.

Yes, this is the point and we are happy to expand on it in the Discussion: cells need the Dot6/Tod6 response to accommodate the Msn2/4 transcriptional changes (and perhaps others in the immediate response) during NaCl. Loss of Msn2/4 alleviates the need for that repression, indicated by the faster post-stress growth rate of the msn2/4Δ strain and the quad strain, during NaCl as originally shown and now also after heat shock. The msn2/4Δ strain grows faster than the wild-type, even though it does not fully repress rESR genes – this is consistent with our model that the main function of rESR repression is resource reallocation: if those resources are no longer needed, there may be little cost to the milder repression of these genes. We will expand on this point in the Discussion.

Minor points.

1) In HIGHLIGHTS, the authors make the point "Cells lacking repressors of growth promoting genes have a slower post-stress growth rate". As the authors mention in the text, this is the finding of their previous paper (Bergen et al. eLife 2022) and should not be highlighted in this work.

We apologize that this was not meant to be presented as a new result, but rather a critical part of our integrated model. We will remove it from the highlights and find a different way to highlight our model.

2) Figure 7, according to the authors' conclusion, the dashed line from Msn2/4 to the promoter for ribosome and growth gene should be a negative regulation ("T shape" allow).

We will adjust the figure to reflect that Msn2/4 may contribute to the repression of these genes.

3) Page 21, line 18, there may be typo (..., slows growth | the absence....).

Thank you, we will correct the typo.

Referee #2 (Report for Author)

Msn2 and Dot6/Tod6 have been the subject of numerous studies demonstrating their importance in stress adaptation and recovery from quiescence. In this study, the authors demonstrate that induction of genes targeted by Msn2 (iESR) but also the inhibition of growth promoting genes (rESR) by Dot6/Tod6 are key points that determine both the future capacity to grow and to resist to future stress. In support of this conclusion, the authors show that deletion of Msn2 can increase growth rate during recovery at the expense of future stress resistance, and consequently Msn2-deficient cells grow significantly faster than wild-type cells during salt acclimation, but have defects in stress resistance (peroxide resistance). The authors propose that Msn2 is not only required for the expression of key stress-related genes involved in stress adaptation, but also for the repression of growth-promoting genes through direct regulation of Dot6 levels and through indirect/direct action of Msn2.

The observation that Msn2 deleted cells have higher growth rate during recovery after stress, in contrast to Dot6/Tod6 mutants, and a defect in peroxide resistance represents the major interest of this study presented in Figure 1, 2. Furthermore, the fact that both Msn2 and Dot6/Tod6 deletion have no growth phenotype in exponential growth phase avoids misinterpretation bias and helps to support their conclusions.

We are glad this reviewer appreciated the novelty and importance of these results. As outlined below, what is equally important are the remaining results of the paper that we incorporate into an integrated model of coordinated resource reallocation.

The other parts of this work (Figures 3, 4, 5, 6) add no or only a modest conceptual advance without important mechanistic insights and some conclusions are less well supported by the data presented (mainly the role of Msn2 in repressing rESR genes). Importantly, some data are redundant with previously published studies, e.g. the authors already published in a previous study that deletion of Dot6/Tod6 reduces growth rate after stress (although they presented this data as new data in HIGHLIGHTS "Cells lacking repressors of growth-promoting genes have a slower post-stress growth rate"). Moreover, it has already been suggested that Msn2 has an (indirect) effect on growth-promoting genes, for reasons that remain unclear and were not really clarified in this study. As described in more detail below, new experiments testing mutants under different growth/stress conditions need to be included in the first part of this study to improve the interest of the manuscript, as well as a more cautious interpretation of the genome-wide data,

which is mainly based on Venn diagram analysis and the criteria chosen by the authors to define Msn2/4 target genes. The model also needs to be modified, as the data in this manuscript never showed that Msn2 regulates the DOT6 paralog, TOD6 (see more details in major concern 4).

We thank the reviewer for their careful reading of the manuscript, and we provide detailed responses below. We agree that additional experiments, specifically testing mutant growth effects in other stresses and testing how deletion of the MSN2/4 binding site and altered DOT6 expression affect growth and repression, will benefit the manuscript. We did not intend for the Highlight of dot6tod6Δ growth effect to represent it as a new result, but rather a critical part of our model – we will remove that as a Highlight and instead focus on the new aspects of our model. Responses to the additional points are presented below.

We very much disagree with this reviewer that the remainder of our manuscript is only a modest advance. While several papers, including Elfving et al. 2014 and our own work (see Chasman et al. 2014) noted that msn2/4Δ cells show altered repression of rESR genes, there had been no evidence that this was anything beyond an indirect response to a perturbed cellular state. We provide new data including mutant growth rates, single-cell microscopy, and transcriptome responses, integrated into a novel model of resource allocation: cells mount a costly Msn2/4 response, which is balanced by the Dot6/Tod6 response, to support acquisition of subsequent stress tolerance. The evolutionary rationale for this model can be explained by acquired stress tolerance: cells accept (and manage, via Dot6/Tod6 activity) a cost for the Msn2/4 response during the mild stress treatment so as to accelerate acquisition of subsequent stress tolerance. We believe that these insights and our novel model will be of significant interest to the stress biology field.

Reviewer 2 cites a NAR study by Elfving et al. as evidence that these points were already known. However, we strongly disagree. Elfving et al. 2014 used Msn2 ChIP-seq and nucleosome occupancy analysis to study regulation of a shift from glucose to glycerol. They identified 192 promoters bound by Msn2. While most of these genes are induced by glucose depletion, the authors did report a handful of repressed genes involved glucose catabolism whose promoters are bound by Msn2 after glucose shift, leading to the conclusion that these factors can mediate gene induction and repression via STRE elements. They also measured gene expression upon very high Msn2 over-expression, which is known to be stressful for cells (and almost certainly activates rESR repression indirectly as part of that stress). They do cite that some genes involved in ribosome biogenesis are repressed in response to artificial Msn2 over-expression and that these genes do not contain upstream STRE element, stating “This suggests that much of the repression is an indirect effect of Msn2 induction.” The remainder of that paper focuses on specific promoters with different nucleosome dynamics. There is one sentence in the Discussion that mentions that Msn2 activates expression of DOT1, a gene involved in telomeric silencing. Reviewer 2 argues that the authors ‘meant’ to say DOT6, but there is no citation, no discussion, no model, no figure and only a reference to a data file. That manuscript furthermore has no mention of resource allocation. Although it is a fine paper, we disagree that it supplants the new experiments, results, and interpretation that our manuscript presents. As outlined below, we do agree that additional experiments will bolster the paper.

Major concerns:

1/ To increase the impact of this study for the wider readership of The EMBO Journal, and because the author uses the general term "yeast stress response" in the title, it would be 1) important to test not only salt stress but also growth recovery after other physiological stress

conditions (pH, heat shock, starvation) with the experimental set-up used in Figure 1-2. It would also be important to test 2) single Dot6 and Tod6 deleted cells as well as 3) cells mutated for the Msn2 binding site at the Dot6 promoter for the reasons explained below in points 3-4. It would also be interesting to 4) test other mutant strains such as the Msn26A allele described in Elfving et al, 2014 or deletion of STB3 or XBP1, since these two factors are involved as Dot6/Dot6 in recovery from quiescence and in repression of growth or cell cycle transcriptional program by promoting recruitment of HDAC Rpd3 at specific set of promoters (PMID: 26300265). Interestingly, a previous study has already shown that Msn2 also regulates and binds to the promoter of the repressor XBP1 (Elfving et al, 2014). These simple experiments will be very useful to extend and support the conclusion drawn in this study regarding the interplay between Msn2 and Dot6/Tod6 during the yeast stress response.

We thank the reviewer for these suggestions and agree with many of them. 1) We are already testing growth properties in other stresses; we have recapitulated the faster growth of msn2/4Δ and slower growth of dot6/tod6Δ after heat shock and are now testing other stresses. 2) We already have single Dot6 and Tod6 mutants and GFP fusions and can evaluate the contribution of each gene. 3) We agree that testing the role of the Msn2/4 binding site upstream of DOT6 is important, as is testing the consequences of augmenting DOT6 transcription. To that end, we are generating a strain in which the genomic copy of DOT6 (or DOT6-GFP) are downstream of a DOT6 promoter that lacks the single STRE element. We are also generating a strain in which DOT6 is controlled by a Tet inducible promoter to test the effects of different levels of DOT6 expression. 4) The proposal to test other gene deletions and mutations is an interesting one, but on that we feel is beyond the scope of this work. However, we will add an extended section to the Discussion presenting how Msn2/4 could regulate other genes important for ESR signaling, the role of STB3 and XBP1, and other regulators mentioned in Elfving et al.

2/ In Figure 2, the effect of the delay in the establishment of peroxide tolerance appears weak and curiously not significant at 20 minutes. Moreover, in the extended view of Figure 1 (panel C), it appears that maximum tolerance is not established in some wild type until 60 minutes, similar to what was described in some older studies and for the double mutant in the main figures 2. Would the authors like to comment on this point? Furthermore, it's clear that the figure in the extended view of Figure 1 (E,F,G,H) does not follow the EMBO guidelines for figures, as it seems that a large number of drop tests from different plates have been fused (mainly in panel H). These data are important in this study and the authors need to show the raw data of the different replicates for these drop test in Expanded View Figure 1-H and also the raw data that allow to make the figure shown in Figure 2.

The assay we used for acquired stress resistance is somewhat noisy with day-to-day variation, and the earliest measured time point at 20 min has a small effect in the wild-type strain, and put together we did not have statistical power to overcome. This can be seen in the extended figure view. The defects at later time points are clearly reproducible and statistically significant, showing that the dot6/tod6Δ cells have a delay in acquired peroxide tolerance but eventually reach wild-type levels. The supplemental image was meant to show representative data, which requires many individual plates over different time points. We are happy to provide all of the plate images, and we will add intervening white space to the representative figure where images across different plates are displayed. Notably, each mutant is compared to a side-by-side treated and plated wild-type strain, maximizing our statistical power with paired analyses.

3/ At the beginning of the paragraph on page 15, the authors say "A remaining question is if the weaker Dot6 activation in the msn2Δmsn4Δ mutant leads to weaker repression of Dot6 target genes. To directly address this question, we followed dynamic

changes to the transcriptome before and in ten-minute increments after salt stress in the different strains. We first identified 489 genes whose response to salt treatment was altered in the dot6 Δ tod6 Δ strain compared to wild-type (FDR < 0.05) in at least two timepoints (see Methods)."

The data presented in this paper only show that Msn2 is a DOT6 regulator and not TOD6. Therefore, to study the importance of Msn2 in regulating Dot6 levels, the authors can't only use the double mutant Dot6/Tod6 for the experiments in Figure 1, 2 and for the transcriptomic analysis in Figure 5, but they need to also test also single DOT6 mutant and a strain mutated for the binding site of Msn2 at the DOT6 promoter. These experiments are essential for understanding the importance of Msn2 on Dot6 level regulation.

Thank you to the reviewer for pointing this out: the goal of this section of the paper was to define genes regulated by Dot6/Tod6 so that we could then investigate their expression when MSN2/4 are deleted. This section of the paper shows that loss of MSN2/4 leads to weaker repression of many genes that are dependent on DOT6/TOD6, even when DOT6 and TOD6 are present in the cell. We agree with this and the other reviewers that testing the impact of Msn2 binding of the DOT6 promoter is important. As outlined above, those experiments are underway.

4/ Following on from the previous comments, it does not seem correct to show in the final model (Figure 6) that Msn2 regulates both Dot6 and Tod6 for two main reasons. Firstly, there are no data in this study showing that Msn2 regulates Tod6 and secondly, the authors ignore that the major transcriptional regulator of growth promoting genes, Sfp1, has already been shown to directly regulate Tod6 expression (PMID: 30804227). The latter point is particularly interesting as it suggests that the two stress sensitive TFs, Sfp1 and Msn2, regulate rESR and iESR respectively, and the two paralogs Tod6 and Dot6. Such regulation would maintain a pool of Dot6 and Tod6 in a wide range of growth conditions. In addition, it might be interesting to plot the mRNA levels of Dot6 and Tod6 on the same graph in Figure 7.

We apologize for the confusion: we are not arguing that Msn2 regulates TOD6 and did not intend to represent that in the figure; we will update the figure to clarify that we provide evidence in the paper that Msn2/4 affect Dot6 levels. We can add TOD6 mRNA levels to Figure 6, and we will provide more detail on Tod6 regulation as outlined below.

5/ In Figure 4, the authors show that Msn2 has an effect on Dot6 levels and localisation, but given that the two paralogs are thought to have similar function, it seems important to measure Tod6 localisation using a similar experimental setup. In addition, Msn2 is unlikely to affect the level of Tod6, unlike Dot6, and for this reason it will be easier to examine changes in Tod6 localisation by microscopy than Dot6 in Msn2-deficient cells.

We already have single gene deletions and Tod6-GFP fusion strains and can certainly quantify their effects on growth rate and abundance in msn2/4 Δ cells. Given that our past work showed that the Dot6 peak height correlates with post-stress growth rate (Bergen et al.) and that MSN2/4 deletion affects Dot6 protein levels (this study), we aim to focus experiments on the effect of reduced Dot6 protein on these phenotypes through other experiments outlined above.

6/ Finally, I have probably the most important concerns with the analysis and conclusions in Figure 5-6.

My first concern relates to analysis of Msn2 targets. The authors use YeasTRACT, which is a

really useful tool, but some large-scale data have to be taken with cautious, as do many databases that pool genome-wide data from large-scale studies of varying quality. Indeed, according to Yeasttract, Msn2/4 directly bind to more than 2100 of the 5500 RNAPII promoters, but it seems very unlikely that Msn2/4 bind to more than a third of the coding gene promoters.

We used agglomerated ChIP-seq data provided in YEASTRACT to display in the figures genes with measured upstream Msn2/4 binding, but we are happy to provide a more detailed analysis of individual ChIP-seq datasets, including from our own lab. We point out that aside of for display, we used this information for statistical analysis: the group of genes that we defined whose induction requires Msn2/4 is enriched with statistical significance for genes with upstream Msn2/4 binding according to YEASTRACT; this group is also heavily enriched for genes with upstream Msn2/4 binding elements. This fits the expected model that Msn2/4 bind STRE elements to mediate gene induction. In contrast, the group of genes whose repression is dependent on Msn2/4 are also enriched with significance for genes with upstream Msn2/4 binding, compared to all other genes with upstream binding in YEASTRACT – but this gene group is statistically significantly under-enriched for genes with upstream STRE elements. We can certainly add a more detailed analysis of upstream Msn2/4 binding specific to NaCl and also seen in other stress datasets, taken directly from those studies instead of from the YEASTRACT database.

Similarly, by using a very poor stringent cut-off to identify promoters with Msn2 binding site (they select all genes containing a CCCCT STRE 1000bp upstream TSS), they identified more than 2000 genes containing this motif. Furthermore, a recent study shows that the presence of the STRE motif is a poor predictor of Msn2 binding in vivo (PMID: 32553192, PMID: 38109289), as the disordered region associated with Msn2 mainly defines its binding to promoters containing an STRE motif in living cells, suggesting that the presence of the STRE motif does not accurately define Msn2 binding. In this sense, in a study aimed at defining the target of Msn2, it was reported that "while approximately 11,500 STRE sequences are present in yeast genome, with 3160 lying in the promoters of almost 2000 genes, Msn2 fails to bind most of these sites." (Elfvig et al, 2014, PMID: 24598258). In the latter study, Elfvig and colleagues defined a list of 268 targets (including Dot6) using various analyses including RNAPII and Msn2 ChIP-seq in different mutants and nucleosome profiling, which is likely to be more appropriate than the list used in this manuscript.

The reviewer is correct that most genomic sequences matching transcription factor binding sites are not functional sites that are bound in cells. Aside of for DOT6, we did not focus on individual STRE elements or use them to define gene groups. As outlined above, we use statistical tests to show that different gene groups that we defined statistically are enriched or under-enriched with significance over other genes in the genome using the same criteria, thus naturally controlling for the high frequency of this short site in the genome. We are happy to repeat the analysis focusing on STRE elements within 500 bp or 200 bp upstream of the gene start sites.

Nevertheless, the authors use these poorly defined categories to analyze their transcriptomic data, mainly by defining subcategories using a Venn diagram, and the conclusion appears confusing. From my point of view, this Venn diagram analysis needs to be repeated with a more precise definition of the promoter and the number of STRE motifs per promoter, knowing that most studies that carefully assess Msn2/4 binding suggest that Msn2/4 binds to roughly 300 RNAPII promoters.

The reviewer is incorrect on how we did our analysis, and we apologize if there was confusion in the presentation. We defined genes whose expression is statistically significantly defective in

*each mutant (using glm models in edgeR with a false discovery rate of 0.05). We then analyzed these gene sets using hierarchical clustering analysis, shown in two types of heat maps in Fig 5A and 5B, and we aligned with that data information on genes with Msn2/4 bound or with upstream elements (the latter of which was used for display and in scoring statistical enrichment, as discussed above, but not in defining any gene groups). The Venn diagrams simply show the overlap in gene sets defined statistically in *msn2/4*Δ and *dot6/tod6*Δ cells; detailed analysis and statistics are presented in the main text. We can certainly provide additional analysis on the hierarchically clustered gene groups shown in the figure heat maps, with sub-cluster analysis of binding patterns.*

Furthermore, if the authors want to show a direct effect on transcription of Msn2, they need to measure RNA synthesis or RNAPII binding and not mRNA accumulation.

*Ultimately, *msn2/4*Δ cells have reduced DOT6 mRNA abundance and Dot6 protein. We will test the role of the upstream Msn2/4 binding element on these phenotypes as well as the effect of altered Dot6 abundance to test functional consequences of altered Dot6 regulation.*

7/ I also have major concerns about redundant conclusions between the Kocik et al. manuscript and other older studies.

-The fact that the absence of Dot6/Tod6 leads to slower acclimation to salt has already been shown (Bergen et al. 2022, PMID: 36350693) and can't be presented as new data in HIGHLIGHT.

As stated above, we will remove this from the Highlights as it was not intended to present new data but rather an important component of our model.

-Similarly, it has already been shown in previous studies that deletion of DOT6/TOD6 prevent proper repression of rESR has been shown in numerous studies (PMID: 31395866, PMID: 21730963, PMID: 35310337, among others) and may not be present as a new result: "These results indicate a primary role for Dot6/Tod6 in repression of their RiBi target genes.", this sentence must be accompanied by references to other studies.

We are happy to change the text to state "these data confirm that Dot6/Tod6 are required for full repression of these genes, as we (Ho et al. 2018) and others [as mentioned by this reviewer] have observed previously." The identification of these genes was not meant to be a new result but rather a mechanism to identify genes dependent on Dot6 so that we could investigate their expression when MSN2/4 are deleted.

- Finally, Elfving and colleagues (PMID: 24598258), in an elegant study aimed at defining the role of Msn2/4 during stress, already show that "Msn2 promotes both transcriptional activation and transcriptional repression" and propose "We do find that many of the genes repressed upon activation of Msn2 are highly enriched for those involved in ribosome biogenesis. However, few of these genes are bound by Msn2, at least under conditions of nutrient downshift. Rather, we observed that Msn2 activates transcription of DOT1, which encodes a repressor of ribosome biogenesis genes (see Supplementary Tables S1 and S2). Moreover, we find that Msn2 binds to and activates transcription of XBP1, which encodes a repressor of a number of genes required for cell cycle progression. Accordingly, Msn2, while a primary purveyor of the ESR, may indirectly repress the growth-associated genes encompassed in the ESR."

Respectfully, the first quote refers to a handful of the 192 genes identified by Elfving et al. that are repressed by glucose depletion and show upstream Msn2 binding and harbor STRE elements. This is not at issue in our study. As outlined above, over-expression of Msn2 is stressful and almost certainly activates the remaining components of the ESR; that over-expression of MSN2 is stressful and indirectly regulates rESR genes does not supplant our results. Finally, as outlined above, DOT1 is a regulator of telomere genes, and cell-cycle genes are not a part of our model.

This conclusion is very similar to the one made in the present manuscript (but Elfving and colleagues made a mistake in the name of DOT1 instead of DOT6 (in agreement with this point, DOT1 is not present in their Supplementary Table S1 showing the list of promoters bound by Msn2, in contrast to DOT6).

With all due respect, the authors did not mention DOT6 and we can find no citations in the paper that study DOT6. Furthermore, the work by Elfving et al. mentions nothing about resource allocation or the function of the rESR. Thus, we disagree that our model in this manuscript has been previously presented by Elfving et al.

In conclusion, the data presented in Figures 5 and 6 mainly use old data and the conclusions are redundant with older studies.

This is not correct: Fig 5 is all new transcriptomic data generated as part of this manuscript; 6A is new data and 6B is from previously published datasets that have not been shown quantitatively in this way.

If this paper is accepted for revision, the authors must at least improve the quality of their analysis, try to validate the binding of Msn2 at the Dot6 promoter by ChIP of Msn2 in a strain mutated for the Msn2 motif at the DOT6 promoter, and more carefully report previous observations already made by other laboratories.

We believe that the experiments proposed above and methods refinement will meet this Reviewer's request. We have clearly cited other data used in the manuscript, including that from our prior studies; we can add clearer statements that Elfving et al. and Chasman et al. manuscripts had previously noted that Msn2/4 may affect the repression of rESR genes but the mechanism or rationale had not been previously known.

Minor comments :

8/ The authors propose that the timing of the delay correlates with delayed Ctt1 protein production in the dot6Δtod6Δ mutant. In a previous study, the authors showed that increasing Ctt1 mRNA levels using a Gal-estradiol-inducible promoter is sufficient to establish peroxide tolerance (PMID: 22851651), but that the absence of Dot6/Tod6 also increases Ctt1 mRNA levels, but not sufficiently to increase total Ctt1 protein levels (Bergen et al. 2022, PMID: 36350693). To confirm their suggestion that the delay in the acquisition of peroxide tolerance is dependent on Ctt1, it may be interesting to use this inducible Gal system to artificially increase Ctt1 mRNA and protein levels. It is also possible that this delay is independent of Ctt1 accumulation.

This is an interesting idea but beyond the scope of this manuscript, since as the reviewer points out we have already shown that artificial induction of CTT1 leads to a coordinate increase in acquired peroxide tolerance.

9/ Perhaps it's my fault and these data are available, but I can't find an Excel table reporting the source data of Figure 5, allowing me to see which gene of all RNAPII-transcribed genes was associated as an Msn2 target, STRE motif, GATGAG motif according to authors.

We are happy to add all of the details shown in Figure 5 to the RNA-seq data file provided.

10/ The figure 4 and one part of associated text have minor interest and could be remove in supplemental Figure.

We disagree and consider the single-cell microscopy in Fig 4 to be among the most important results in the paper. We will retain this figure to show the impact of Msn2/4 on Dot6 abundance and activity.

11/ Probably beyond the scope of this study, but it would be interesting for the authors to discuss stufy from the Barkai lab and/or to use the genetic tool developed in Kafri et al, 2016 to study growth rate recovery during adaptation and acquisition of peroxide resistance. Kafri and colleagues use different strains expressing one to ~20 copies of mCherry under the control of a strong promoter to challenge the physiological effect of overexpression of an inert protein or unstable mRNA (cells forced to transcribe unstable DAMP transcripts). According to the model and the result in Figure 2, it would be expected that such overexpression would also delay the acquisition of resistance. Furthermore, these constructions will also help to understand whether the effect is related to transcription and/or translation. Perhaps the author can at least discuss this point.

This is an interesting point and something we have considered. At this point, we will focus on other experimental additions but can add a point about this in the Discussion.

Referee #3 (Report for Author)

This study investigates how yeast cells relocate transcriptional and translational resources among gene groups during environmental stress. The authors explore the interplay between the transcription factors Msn2/Msn4 and Dot6/Tod6, concluding that Msn2/Msn4 are the primary regulators in resource redirection. Beyond activating stress-response genes, Msn2/Msn4 play a pivotal role in repressing growth-related genes via three mechanisms: 1) Regulating the transcription of DOT6/TOD6. 2) Promoting the nuclear localization of Dot6/Tod6. 3) Regulating the transcription of a subset of rESR genes. While the proposed modeling is intriguing, the supporting evidence is insufficient, and many underlying mechanisms remain unclear. Additional experiments are necessary to solidify these conclusions.

We are glad the reviewer found the model intriguing and hope that added experiments will provide additional clarity on the model.

Specific Comments:

1) In Figure 6, the authors show that *msn2Dmsn4D* mutants have lower DOT6 mRNA levels compared to wild-type cells. However, time-course data indicate that Msn2/Msn4 are only minor contributors, as both mutant and wild-type cells upregulate DOT6 mRNA to comparable levels post-stress (60 min), albeit with a slight delay in the mutants. The role of Msn2/Msn4 in regulating DOT6 expression is overstated.

Both replicates show that $msn2/4\Delta$ cells show reduction of DOT6 mRNA after NaCl treatment, during the highly dynamic phase in which we can readily capture expression differences, and other datasets (Fig 6B) capture the same response. Importantly, Dot6 protein is lower in $msn2/msn4\Delta$ cells – we propose that burst of Msn2/4 activity before and after stress (see Bergen et al.) maintain a pool of Dot6 protein. The main point of this figure is that there is statistically significant lower DOT6 mRNA in the $msn2/4\Delta$ mutant, matching the lower protein amounts we report. Understanding how dynamic mRNA changes affect dynamic protein changes would require mathematical modeling which is beyond the scope of this study.

2) In Figure 4, the reduction of Dot6 protein levels in $msn2\Delta msn4\Delta$ mutants before stress is insufficiently explained. The authors attribute this to the absence of stochastic bursts of ESR induction, but single-cell data show consistently low Dot6 abundance across nearly all mutant cells before stress (Figure 4A). This observation is difficult to reconcile with stochastic bursts. A potential experiment to address this would involve mutating the Msn2/Msn4 binding elements in the DOT6 promoter and assessing whether Dot6 protein levels remain low under no-stress conditions.

As outlined above, this experiment is underway.

3) The presentation of results in Figure 5 is unclear. For example, the sentence "Two thirds (68%) of these genes contain upstream Dot6/Tod6 binding elements...and 249 genes showed a repression defect..." raises questions: How many genes overlap between these two groups? How many of these genes also exhibit Msn2/Msn4 binding? How many Msn2/Msn4 binding genes lack upstream STRE elements? The data and descriptions should be reorganized to allow clear comparisons of specific features across gene groups.

In accordance with this comment and comments of the other reviewers, we will present a more detailed analysis of the expression data to provide higher resolution on different gene groups and their regulation.

4) Numerous paragraphs in the Results section end with "see Discussion," but it is challenging to locate where these data are discussed. It would be more effective to integrate relevant discussions directly within the results paragraphs.

We will update the text to make it clearer which sections we refer to, and to allude to the point in the Results section.

Minor Comments:

1) Please add page numbers and line numbers to the manuscript for easier reference.

Apologies, we will update this going forward.

2) Do the authors have any hypotheses regarding how Msn2/Msn4 binding represses rESR genes? Including such speculation would strengthen the interpretation of results.

We are happy to add some discussion on this point, although ultimately elucidating this will require future experimentation.

3) In the sentence "conversely, ppGpp over-production, leading to stronger... slow growth in the absence of stress...", it should be "in the absence of stress".

Thank you, we will correct the error.

Dr Audrey P. Gasch
University of Wisconsin-Madison
Laboratory of Genetics
425-g Henry Mall
Madison, WI 53706

10th Feb 2025

Re: EMBOJ-2024-119732
Regulated resource reallocation is transcriptionally hard wired into the yeast stress response

Dear Dr. Gasch,

Thank you for sending me your point-by-point response to the individual referee comments on your recent EMBO Journal submission, and your proposal for how to address them in a revision. I am pleased to say that I found your responses overall well-taken, and would therefore formally invite you to prepare a revised version of the manuscript for further consideration for The EMBO Journal. I appreciate that the various experiments you are planning or have already started conducting should clearly be helpful for strengthening the study and for addressing shared key concerns of all three referees. Furthermore, I think it would be of value if you also added the various data analyses and presentational changes that you recurrently mentioned as things you "could do". In this respect, please note the hierarchical data presentation options at EMBO Press (www.embopress.org/page/journal/14602075/authorguide#expandedview), with main data, expanded view data, Appendix data, and source data, as well as data only included in the referee response letter, which we also make publicly available upon publication as part of our transparent review process. Should you have any questions in this regard, other questions regarding the revision work, or should you need an extended revision period, please do not hesitate to contact me.

Regarding the arguments related to Elfving et al 2014: I have looked at this work and agree that it can hardly be taken as precedent for key conclusions of the present work. Nevertheless, I realize that their analysis of Msn2 binding sites and Msn2-dependent transcription did identify DOT6 (and DOT1), and that their text refers to DOT1 as "a repressor of ribosome biogenesis genes", making it likely that Elfving et al may have indeed mixed it up with DOT6 here (both genes being obviously also "Disruptors of Telomeric silencing"). Either way, I agree that these authors did not go significantly into this, but it may be valuable to nevertheless acknowledge this possibility when "adding clearer statements that Elfving et al and Chasman et al manuscripts had previously other that Msn2/4 may affect the repression of rESR genes", as proposed on page 10 of your response.

In summary, we shall be happy to further pursue a revision of this study, modified and extended along the lines suggested in your response letter. Please keep in mind that it is our policy to allow only a single round of major revision, and please update me in case there should be any unexpected problems with the revisions. As always, competing manuscript published during the course of this revision will not affect our final decision on your study. Finally, please note the detailed information and guidelines on how to prepare a revision below (and in our online Guide to Authors) - closely adhering to them shall greatly facilitate the editorial process at the time of resubmission.

Thank you again for the opportunity to consider this work, and I look forward to receiving your revision in due time.

With kind regards,

Hartmut

- 2) Each figure legend must specify
 - size of the scale bars that are mandatory for all micrograph panels
 - the statistical test used to generate error bars and P-values
 - the type error bars (e.g., S.E.M., S.D.)
 - the number (n) and nature (biological or technical replicate) of independent experiments underlying each data point
 - Figures may not include error bars for experiments with $n < 3$; scatter plots showing individual data points should be used instead.
- 3) Revised manuscript text (including main tables, and figure legends for main and EV figures) has to be submitted as editable text file (e.g., .docx format). We encourage highlighting of changes (e.g., via text color) for the referees' reference.
- 4) Each main and each Expanded View (EV) figure should be uploaded as individual production-quality files (preferably in .eps, .tif, .jpg formats). For suggestions on figure preparation/layout, please refer to our Figure Preparation Guidelines: <http://bit.ly/EMBOPressFigurePreparationGuideline>
- 5) Point-by-point response letters should include the original referee comments in full together with your detailed responses to them (and to specific editor requests if applicable), and also be uploaded as editable (e.g., .docx) text files.
- 6) Please complete our Author Checklist, and make sure that information entered into the checklist is also reflected in the manuscript; the checklist will be available to readers as part of the Review Process File. A download link is found at the top of our Guide to Authors: embopress.org/page/journal/14602075/authorguide
- 7) All authors listed as (co-)corresponding need to deposit, in their respective author profiles in our submission system, a unique ORCID identifier linked to their name. Please see our Guide to Authors for detailed instructions.
- 8) Please note that supplementary information at EMBO Press has been superseded by the 'Expanded View' for inclusion of additional figures, tables, movies or datasets; with up to five EV Figures being typeset and directly accessible in the HTML version of the article. For details and guidance, please refer to: embopress.org/page/journal/14602075/authorguide#expandedview
- 9) To facilitate reproducibility and cross-laboratory adoption of methodologies, please structure the Materials & Methods section as outlined in our guide to authors, including a completed Reagents and Tools Table that can be downloaded from our author guidelines as well (<https://www.embopress.org/page/journal/14602075/authorguide#structuredmethods>).
- 10) Digital image enhancement is acceptable practice, as long as it accurately represents the original data and conforms to community standards. If a figure has been subjected to significant electronic manipulation, this must be clearly noted in the figure legend and/or the 'Materials and Methods' section. The editors reserve the right to request original versions of figures and the original images that were used to assemble the figure. Finally, we generally encourage uploading of numerical as well as gel/blot image source data; for details see: embopress.org/page/journal/14602075/authorguide#sourcedata

At EMBO Press, we ask authors to provide source data for the main manuscript figures. Our source data coordinator will contact you to discuss which figure panels we would need source data for and will also provide you with helpful tips on how to upload and organize the files.

In the interest of ensuring the conceptual advance provided by the work, we recommend submitting a revision within 3 months (11th May 2025). Please discuss the revision progress ahead of this time with the editor if you require more time to complete the revisions. Use the link below to submit your revision:

Link Not Available

Referee #1:

Across species, cellular stress responses are often associated with the modulation of transcriptional and translational capacity, which appears to be important for the reallocation of cellular energy and resources from cell growth/proliferation to selective responses that confer stress tolerance. However, the molecular details of this reallocation are not fully understood. Kocik and

Gash addressed this question utilizing the yeast environmental stress response (ESR) and high salt stress as a model system. They identified the stress-responsive transcription factor Msn2/Msn4 as a key coordinator of the ESR, regulating the induction of defense genes and the repression of growth-related genes in response to stress conditions. The Msn2/4-dependent response appears to slow cell growth rate under salt stress conditions and is required for the acquisition of stress tolerance to secondary hydrogen peroxide insult after the salt stress. The authors also found that Msn2/4 is an upstream regulator of the Dot6/Tod6-dependent transcriptional repression of growth-promoting genes through Dot6 expression and nuclear translocation. The findings here seem to be an important addition to the model of the ESR that the authors have established through their previous work. Overall, the single cell analysis and RNA-seq data with yeast genetics are solid. However, the proposed mechanisms for regulation of rESR genes by Msn2/4 are less convincing. To further support their conclusion, concerns listed below should be addressed.

Major points.

- 1) Since it is hard to interpret the difference between the strains in the growth curve in the acute stress response phase (30-60 min after NaCl) in Figure 1B, the authors should show the bar graphs with statistical analysis for this early phase as shown in Figure 1C for the acclimation phase. Also, there seems to be no clear description of why the quad mutant exhibited the phenocopy of the Msn2/4 mutant only in the late phase but not in the early phase.
- 2) Figure 4, it would be unclear to readers not familiar with the yeast gene expression system (like this reviewer) whether exogenously expressed Dot6-GFP is under the control of the endogenous Dot6 gene promoter or some other promoter which is co-introduced with Dot6-GFP. If the latter is the case, promoter specific transcriptional regulation of Dot6-GFP is unlikely to be the cause of the reduction of Dot6-GFP expression in the MSN2/4 mutant, since the reporter expression should be independent of endogenous Dot6 gene expression. Is it possible that Msn2/4 regulates Dot6-GFP protein levels by regulating the protein stability of Dot6-GFP or the translation rate of Dot6-GFP?
- 3) Related to the above question, the Msn2/4 mutant exhibited weaker Dot6 nuclear translocation (Figure 4D). Given the subcellular localization of Dot6 is regulated by phosphorylation, is it possible that the MSN2/4 affects phosphorylation of Dot6, e.g., through the activity of PKA or mTOR signaling?
- 4) Figure 5, it is unclear whether MSN2 and MSN4 binding to each rESR gene promoter is experimentally confirmed or just speculated from the transcription factor binding site. It seems that the authors used the YEASTRACT+ database as described in Methods, but a more detailed explanation of how the MSN2/4 binding is determined should be added in the main text.
- 5) Overall, experimental evidence for direct binding of MSN2/4 to the Dot6 promoter and the effect of MSN2/4 on Dot6 mRNA expression is weak. The authors only showed the data from previous works with different conditions (Fig. EV3). Experiments to show the binding of MSN2 to the Dot6 promoter, such as Chromatin-IP in the current conditions, would be needed. Also, changes in Dot6 mRNA levels do not appear to be significant (Fig. 6). Immunoblotting of endogenous Dot6 would support the argument that the reduction in Dot6 mRNA levels in the MSN2/4 mutant does reduce functional Dot6 protein in the cell.
- 6) MSN2/4 mutant showed reduced Dot6 expression/activity (Fig. 4) and de-repression of rESR genes that partially overlap with the Dot6/Tod6 target. These observations seem to contradict the increased post stress growth rate of Msn2/4 mutant (Fig. 2). Does this mean that the cost of iESR induction outweighs the saved cost of rESR repression? This point could be discussed further.

Minor points.

- 1) In HIGHLIGHTS, the authors make the point "Cells lacking repressors of growth promoting genes have a slower post-stress growth rate". As the authors mention in the text, this is the finding of their previous paper (Bergen et al. eLife 2022) and should not be highlighted in this work.
- 2) Figure 7, according to the authors' conclusion, the dashed line from Msn2/4 to the promoter for ribosome and growth gene should be a negative regulation ("T shape" allow).
- 3) Page 21, line 18, there may be typo (... , slows growth I the absence....).

Referee #2:

Msn2 and Dot6/Tod6 have been the subject of numerous studies demonstrating their importance in stress adaptation and recovery from quiescence. In this study, the authors demonstrate that induction of genes targeted by Msn2 (iESR) but also the inhibition of growth promoting genes (rESR) by Dot6/Tod6 are key points that determine both the future capacity to grow and to resist to future stress. In support of this conclusion, the authors show that deletion of Msn2 can increase growth rate during recovery at the expense of future stress resistance, and consequently Msn2-deficient cells grow significantly faster than wild-type cells during salt acclimation, but have defects in stress resistance (peroxide resistance). The authors propose that Msn2 is not only required for the expression of key stress-related genes involved in stress adaptation, but also for the repression of growth-promoting genes through direct regulation of Dot6 levels and through indirect/direct action of Msn2.

The observation that Msn2 deleted cells have higher growth rate during recovery after stress, in contrast to Dot6/Tod6 mutants, and a defect in peroxide resistance represents the major interest of this study presented in Figure 1, 2. Furthermore, the fact that both Msn2 and Dot6/Tod6 deletion have no growth phenotype in exponential growth phase avoids misinterpretation bias and helps to support their conclusions.

The other parts of this work (Figures 3, 4, 5, 6) add no or only a modest conceptual advance without important mechanistic insights and some conclusions are less well supported by the data presented (mainly the role of Msn2 in repressing rESR

genes). Importantly, some data are redundant with previously published studies, e.g. the authors already published in a previous study that deletion of *Dot6/Tod6* reduces growth rate after stress (although they presented this data as new data in HIGHLIGHTS "Cells lacking repressors of growth-promoting genes have a slower post-stress growth rate"). Moreover, it has already been suggested that *Msn2* has an (indirect) effect on growth-promoting genes, for reasons that remain unclear and were not really clarified in this study. As described in more detail below, new experiments testing mutants under different growth/stress conditions need to be included in the first part of this study to improve the interest of the manuscript, as well as a more cautious interpretation of the genome-wide data, which is mainly based on Venn diagram analysis and the criteria chosen by the authors to define *Msn2/4* target genes. The model also needs to be modified, as the data in this manuscript never showed that *Msn2* regulates the *DOT6* paralog, *TOD6* (see more details in major concern 4).

Major concerns:

1/ To increase the impact of this study for the wider readership of The EMBO Journal, and because the author uses the general term "yeast stress response" in the title, it would be important to test not only salt stress but also growth recovery after other physiological stress conditions (pH, heat shock, starvation) with the experimental set-up used in Figure 1-2. It would also be important to test single *Dot6* and *Tod6* deleted cells as well as cells mutated for the *Msn2* binding site at the *Dot6* promoter for the reasons explained below in points 3-4. It would also be interesting to test other mutant strains such as the *Msn26A* allele described in Elfving et al, 2014 or deletion of *STB3* or *XBP1*, since these two factors are involved as *Dot6/Tod6* in recovery from quiescence and in repression of growth or cell cycle transcriptional program by promoting recruitment of HDAC Rpd3 at specific set of promoters (PMID: 26300265). Interestingly, a previous study has already shown that *Msn2* also regulates and binds to the promoter of the repressor *XBP1* (Elfving et al, 2014). These simple experiments will be very useful to extend and support the conclusion drawn in this study regarding the interplay between *Msn2* and *Dot6/Tod6* during the yeast stress response.

2/ In Figure 2, the effect of the delay in the establishment of peroxide tolerance appears weak and curiously not significant at 20 minutes. Moreover, in the extended view of Figure 1 (panel C), it appears that maximum tolerance is not established in some wild type until 60 minutes, similar to what was described in some older studies and for the double mutant in the main figures 2. Would the authors like to comment on this point? Furthermore, it's clear that the figure in the extended view of Figure 1 (E,F,G,H) does not follow the EMBO guidelines for figures, as it seems that a large number of drop tests from different plates have been fused (mainly in panel H). These data are important in this study and the authors need to show the raw data of the different replicates for these drop test in Expanded View Figure 1-H and also the raw data that allow to make the figure shown in Figure 2.

3/ At the beginning of the paragraph on page 15, the authors say "A remaining question is if the weaker *Dot6* activation in the *msn2Δmsn4Δ* mutant leads to weaker repression of *Dot6* target genes. To directly address this question, we followed dynamic changes to the transcriptome before and in ten-minute increments after salt stress in the different strains. We first identified 489 genes whose response to salt treatment was altered in the *dot6Δtod6Δ* strain compared to wild-type (FDR < 0.05) in at least two timepoints (see Methods)." The data presented in this paper only show that *Msn2* is a *DOT6* regulator and not *TOD6*. Therefore, to study the importance of *Msn2* in regulating *Dot6* levels, the authors can't only use the double mutant *Dot6/Tod6* for the experiments in Figure 1, 2 and for the transcriptomic analysis in Figure 5, but they need to also test also single *DOT6* mutant and a strain mutated for the binding site of *Msn2* at the *DOT6* promoter. These experiments are essential for understanding the importance of *Msn2* on *Dot6* level regulation.

4/ Following on from the previous comments, it does not seem correct to show in the final model (Figure 6) that *Msn2* regulates both *Dot6* and *Tod6* for two main reasons. Firstly, there are no data in this study showing that *Msn2* regulates *Tod6* and secondly, the authors ignore that the major transcriptional regulator of growth promoting genes, *Sfp1*, has already been shown to directly regulate *Tod6* expression (PMID: 30804227). The latter point is particularly interesting as it suggests that the two stress sensitive TFs, *Sfp1* and *Msn2*, regulate *rESR* and *iESR* respectively, and the two paralogs *Tod6* and *Dot6*. Such regulation would maintain a pool of *Dot6* and *Tod6* in a wide range of growth conditions. In addition, it might be interesting to plot the mRNA levels of *Dot6* and *Tod6* on the same graph in Figure 7.

5/ In Figure 4, the authors show that *Msn2* has an effect on *Dot6* levels and localisation, but given that the two paralogs are thought to have similar function, it seems important to measure *Tod6* localisation using a similar experimental setup. In addition, *Msn2* is unlikely to affect the level of *Tod6*, unlike *Dot6*, and for this reason it will be easier to examine changes in *Tod6* localisation by microscopy than *Dot6* in *Msn2*-deficient cells.

6/ Finally, I have probably the most important concerns with the analysis and conclusions in Figure 5-6. My first concern relates to analysis of *Msn2* targets. The authors use YeasTRACT, which is a really useful tool, but some large-scale data have to be taken with caution, as do many databases that pool genome-wide data from large-scale studies of varying quality. Indeed, according to Yeastract, *Msn2/4* directly bind to more than 2100 of the 5500 RNAPII promoters, but it seems very unlikely that *Msn2/4* bind to more than a third of the coding gene promoters. Similarly, by using a very poor stringent cut-off to identify promoters with *Msn2* binding site (they select all genes containing a CCCCT STRE 1000bp upstream TSS), they identified more than 2000 genes containing this motif. Furthermore, a recent study shows that the presence of the STRE motif is a poor predictor of *Msn2* binding in vivo (PMID: 32553192, PMID: 38109289), as the disordered region associated with

Msn2 mainly defines its binding to promoters containing an STRE motif in living cells, suggesting that the presence of the STRE motif does not accurately define Msn2 binding. In this sense, in a study aimed at defining the target of Msn2, it was reported that "while approximately 11,500 STRE sequences are present in yeast genome, with 3160 lying in the promoters of almost 2000 genes, Msn2 fails to bind most of these sites." (Elfving et al, 2014, PMID: 24598258). In the latter study, Elfving and colleagues defined a list of 268 targets (including Dot6) using various analyses including RNAPII and Msn2 ChIP-seq in different mutants and nucleosome profiling, which is likely to be more appropriate than the list used in this manuscript. Nevertheless, the authors use these poorly defined categories to analyze their transcriptomic data, mainly by defining subcategories using a Venn diagram, and the conclusion appears confusing. From my point of view, this Venn diagram analysis needs to be repeated with a more precise definition of the promoter and the number of STRE motifs per promoter, knowing that most studies that carefully assess Msn2/4 binding suggest that Msn2/4 binds to roughly 300 RNAPII promoters. Furthermore, if the authors want to show a direct effect on transcription of Msn2, they need to measure RNA synthesis or RNAPII binding and not mRNA accumulation.

7/ I also have major concerns about redundant conclusions between the Kocik et al. manuscript and other older studies.

-The fact that the absence of Dot6/Tod6 leads to slower acclimation to salt has already been shown (Bergen et al. 2022, PMID: 36350693) and can't be presented as new data in HIGHLIGHT.

-Similarly, it has already been shown in previous studies that deletion of DOT6/TOD6 prevent proper repression of rESR has been shown in numerous studies (PMID: 31395866, PMID: 21730963, PMID: 35310337, among others) and may not be present as a new result: "These results indicate a primary role for Dot6/Tod6 in repression of their RiBi target genes.", this sentence must be accompanied by references to other studies.

- Finally, Elfving and colleagues (PMID: 24598258), in an elegant study aimed at defining the role of Msn2/4 during stress, already show that "Msn2 promotes both transcriptional activation and transcriptional repression" and propose "We do find that many of the genes repressed upon activation of Msn2 are highly enriched for those involved in ribosome biogenesis. However, few of these genes are bound by Msn2, at least under conditions of nutrient downshift. Rather, we observed that Msn2 activates transcription of DOT1, which encodes a repressor of ribosome biogenesis genes (see Supplementary Tables S1 and S2). Moreover, we find that Msn2 binds to and activates transcription of XBP1, which encodes a repressor of a number of genes required for cell cycle progression. Accordingly, Msn2, while a primary purveyor of the ESR, may indirectly repress the growth-associated genes encompassed in the ESR."

This conclusion is very similar to the one made in the present manuscript (but Elfving and colleagues made a mistake in the name of DOT1 instead of DOT6 (in agreement with this point, DOT1 is not present in their Supplementary Table S1 showing the list of promoters bound by Msn2, in contrast to DOT6).

In conclusion, the data presented in Figures 5 and 6 mainly use old data and the conclusions are redundant with older studies. If this paper is accepted for revision, the authors must at least improve the quality of their analysis, try to validate the binding of Msn2 at the Dot6 promoter by ChIP of Msn2 in a strain mutated for the Msn2 motif at the DOT6 promoter, and more carefully report previous observations already made by other laboratories.

Minor comments :

8/ The authors propose that the timing of the delay correlates with delayed Ctt1 protein production in the dot6 Δ tod6 Δ mutant. In a previous study, the authors showed that increasing Ctt1 mRNA levels using a Gal-estradiol-inducible promoter is sufficient to establish peroxide tolerance (PMID: 22851651), but that the absence of Dot6/Tod6 also increases Ctt1 mRNA levels, but not sufficiently to increase total Ctt1 protein levels (Bergen et al. 2022, PMID: 36350693). To confirm their suggestion that the delay in the acquisition of peroxide tolerance is dependent on Ctt1, it may be interesting to use this inducible Gal system to artificially increase Ctt1 mRNA and protein levels. It is also possible that this delay is independent of Ctt1 accumulation.

9/ Perhaps it's my fault and these data are available, but I can't find an Excel table reporting the source data of Figure 5, allowing me to see which gene of all RNAPII-transcribed genes was associated as an Msn2 target, STRE motif, GATGAG motif according to authors.

10/ The figure 4 and one part of associated text have minor interest and could be remove in supplemental Figure.

11/ Probably beyond the scope of this study, but it would be interesting for the authors to discuss study from the Barkai lab and/or to use the genetic tool developed in Kafri et al, 2016 to study growth rate recovery during adaptation and acquisition of peroxide resistance. Kafri and colleagues use different strains expressing one to ~20 copies of mCherry under the control of a strong promoter to challenge the physiological effect of overexpression of an inert protein or unstable mRNA (cells forced to transcribe unstable DAmP transcripts). According to the model and the result in Figure 2, it would be expected that such overexpression would also delay the acquisition of resistance. Furthermore, these constructions will also help to understand whether the effect is related to transcription and/or translation. Perhaps the author can at least discuss this point.

Referee #3:

This study investigates how yeast cells relocate transcriptional and translational resources among gene groups during environmental stress. The authors explore the interplay between the transcription factors Msn2/Msn4 and Dot6/Tod6, concluding that Msn2/Msn4 are the primary regulators in resource redirection. Beyond activating stress-response genes, Msn2/Msn4 play a pivotal role in repressing growth-related genes via three mechanisms: 1) Regulating the transcription of DOT6/TOD6. 2)

Promoting the nuclear localization of Dot6/Tod6. 3) Regulating the transcription of a subset of rESR genes. While the proposed modeling is intriguing, the supporting evidence is insufficient, and many underlying mechanisms remain unclear. Additional experiments are necessary to solidify these conclusions.

Specific Comments:

- 1) In Figure 6, the authors show that *msn2Dmsn4D* mutants have lower DOT6 mRNA levels compared to wild-type cells. However, time-course data indicate that *Msn2/Msn4* are only minor contributors, as both mutant and wild-type cells upregulate DOT6 mRNA to comparable levels post-stress (60 min), albeit with a slight delay in the mutants. The role of *Msn2/Msn4* in regulating DOT6 expression is overstated.
- 2) In Figure 4, the reduction of Dot6 protein levels in *msn2Dmsn4D* mutants before stress is insufficiently explained. The authors attribute this to the absence of stochastic bursts of ESR induction, but single-cell data show consistently low Dot6 abundance across nearly all mutant cells before stress (Figure 4A). This observation is difficult to reconcile with stochastic bursts. A potential experiment to address this would involve mutating the *Msn2/Msn4* binding elements in the DOT6 promoter and assessing whether Dot6 protein levels remain low under no-stress conditions.
- 3) The presentation of results in Figure 5 is unclear. For example, the sentence "Two thirds (68%) of these genes contain upstream Dot6/Tod6 binding elements...and 249 genes showed a repression defect..." raises questions: How many genes overlap between these two groups? How many of these genes also exhibit *Msn2/Msn4* binding? How many *Msn2/Msn4* binding genes lack upstream STRE elements? The data and descriptions should be reorganized to allow clear comparisons of specific features across gene groups.
- 4) Numerous paragraphs in the Results section end with "see Discussion," but it is challenging to locate where these data are discussed. It would be more effective to integrate relevant discussions directly within the results paragraphs.

Minor Comments:

- 1) Please add page numbers and line numbers to the manuscript for easier reference.
- 2) Do the authors have any hypotheses regarding how *Msn2/Msn4* binding represses rESR genes? Including such speculation would strengthen the interpretation of results.
- 3) In the sentence "conversely, ppGpp over-production, leading to stronger... slow growth I the absence of stress...", it should be "in the absence of stress".

The original comments from the reviewers are listed in black text with our response in italicized blue text.

We thank all three reviewers for their constructive comments. We addressed all of the comments with new experiments, more detailed analysis of the transcriptomic data, and updates to the text. We also shifted the manuscript's focus a bit to the importance of regulated resource reallocation and the multi-factorial role of Msn2/4 in mediating that. To this end, we added a new modeling analysis at the end of the paper to explore conditions under which cells would evolutionarily maintain the Msn2/4 response, which comes at a cost during a single-stress response but is essential for surviving combinatorial stresses. We also updated the title accordingly, to "The importance of regulated resource reallocation during dynamic environmental shifts in yeast." We believe that the manuscript is significantly stronger and offers new insights into microbial stress responses and principles conserved in other organisms.

Referee #1 (Report for Author)

Across species, cellular stress responses are often associated with the modulation of transcriptional and translational capacity, which appears to be important for the reallocation of cellular energy and resources from cell growth/proliferation to selective responses that confer stress tolerance. However, the molecular details of this reallocation are not fully understood. Kocik and Gash addressed this question utilizing the yeast environmental stress response (ESR) and high salt stress as a model system. They identified the stress-responsive transcription factor Msn2/Msn4 as a key coordinator of the ESR, regulating the induction of defense genes and the repression of growth-related genes in response to stress conditions. The Msn2/4-dependent response appears to slow cell growth rate under salt stress conditions and is required for the acquisition of stress tolerance to secondary hydrogen peroxide insult after the salt stress. The authors also found that Msn2/4 is an upstream regulator of the Dot6/Tod6-dependent transcriptional repression of growth-promoting genes through Dot6 expression and nuclear translocation. The findings here seem to be an important addition to the model of the ESR that the authors have established through their previous work. Overall, the single cell analysis and RNA-seq data with yeast genetics are solid. However, the proposed mechanisms for regulation of rESR genes by Msn2/4 are less convincing. To further support their conclusion, concerns listed below should be addressed.

We appreciate this reviewer's assessment that this work is an important addition to the field and that our analysis is solid.

Major points.

1) Since it is hard to interpret the difference between the strains in the growth curve in the acute stress response phase (30-60 min after NaCl) in Figure 1B, the authors should show the bar graphs with statistical analysis for this early phase as shown in Figure 1C for the acclimation phase. Also, there seems to be no clear description of why the quad mutant exhibited the phenocopy of the Msn2/4 mutant only in the late phase but not in the early phase.

*We added new data and figure panel 1D that shows the response of these mutants to multiple different stresses. For all five stresses tested, the *msn2Δmsn4Δ* mutant grows faster than wild-type during the acclimation phase, and *dot6Δtod6Δ* mutant grows slower, and the quad mutant lacking all four regulators grows equal to or faster than wild-type. The response of the quad mutant immediately after each stress was not consistent across conditions, and it was difficult to quantify differences statistically due to limited data points and non-exponential growth during this phase. Therefore, since this was not a major point of the paper, we simply deleted it. Our*

main point – that the quad mutant recovers faster than the DOT6/TOD6 mutant – is now bolstered by the addition of other stresses.

2) Figure 4, it would be unclear to readers not familiar with the yeast gene expression system (like this reviewer) whether exogenously expressed Dot6-GFP is under the control of the endogenous Dot6 gene promoter or some other promoter which is co-introduced with Dot6-GFP. If the latter is the case, promoter specific transcriptional regulation of Dot6-GFP is unlikely to be the cause of the reduction of Dot6-GFP expression in the MSN2/4 mutant, since the reporter expression should be independent of endogenous Dot6 gene expression. Is it possible that Msn2/4 regulates Dot6-GFP protein levels by regulating the protein stability of Dot6-GFP or the translation rate of Dot6-GFP?

We apologize for the confusion: the DOT6-GFP coding sequence was integrated into the yeast genome and is expressed from the native DOT6 promoter. We clarified this point on lines 236-238. We believe the most parsimonious explanation is that Msn2/4 can influence DOT6 abundance through promoter binding; however, we added a statement that the effect could also be indirect, perhaps through regulated Dot6 turnover (lines 272-273).

3) Related to the above question, the Msn2/4 mutant exhibited weaker Dot6 nuclear translocation (Figure 4D). Given the subcellular localization of Dot6 is regulated by phosphorylation, is it possible that the MSN2/4 affects phosphorylation of Dot6, e.g., through the activity of PKA or mTOR signaling?

This is an interesting topic. Msn2/4 induce the expression of other stress-responsive regulators, including both positive and negative regulators of PKA signaling (Gasch et al. 2000, Segal et al. 2003, and likely others). Furthermore, our past work shows that Msn2 physically interact with the stress-activated phosphodiesterase Pde2 that regulates PKA during stress (MacGilvray et al. 2018). We are investigating effects of PDE2 deletion on Msn2 and Dot6 nuclear localization through other work that we hope to save for a different manuscript; notably, deletion of PDE2 does not affect Dot6 protein levels. We added clarifications throughout the text that Msn2/4 could also have indirect effects Dot6 abundance.

4) Figure 5, it is unclear whether MSN2 and MSN4 binding to each rESR gene promoter is experimentally confirmed or just speculated from the transcription factor binding site. It seems that the authors used the YEASTRACT+ database as described in Methods, but a more detailed explanation of how the MSN2/4 binding is determined should be added in the main text.

The updated Fig 5 indicates where there is evidence of direct binding of transcription factors to gene promoters as summarized in Yeastract (and confirmed with our own ChIP-seq data from Huebert et al.). A separate column in Fig 5 shows the presence of Msn2/4 “STRE” binding sites within now-500bp upstream of genes in the figure. We added a clarification to the Results and Methods that ChIP-seq data represent ‘direct binding’ evidence in Yeastract. An important note is that we focus on statistical enrichment of genes with direct-binding evidence across groups of genes, which is less subject to single-gene errors in database calls. The exception to this is for DOT6 promoter binding by Msn2/4, for which we show evidence from multiple studies, including NaCl and our own past work (now Fig EV5).

5) Overall, experimental evidence for direct binding of MSN2/4 to the Dot6 promoter and the effect of MSN2/4 on Dot6 mRNA expression is weak. The authors only showed the data from previous works with different conditions (Fig. EV3). Experiments to show the binding of MSN2 to the Dot6 promoter, such as Chromatin-IP in the current conditions, would be needed. Also,

changes in Dot6 mRNA levels do not appear to be significant (Fig. 6). Immunoblotting of endogenous Dot6 would support the argument that the reduction in Dot6 mRNA levels in the MSN2/4 mutant does reduce functional Dot6 protein in the cell.

*We have added additional experiments and analysis to bolster our claims. Part of the challenge here is that Msn2/4 can bind the DOT6 promoter through the known STRE binding site but can also bind through protein-protein interactions via intrinsically disordered domains, even without direct DNA binding (see papers from N. Barkai lab). Indeed, we tried to delete the single known STRE element from this promoter but Msn2 can still bind to it (Brodsky et al. 2020). We instead took an alternative approach and made a TET-inducible Dot6 construct – the revised manuscript shows that reducing Dot6-GFP protein to levels comparable to the *msn2Δmsn4Δ* mutant produces a significant defect in repression of Dot6 targets during NaCl. We maintain that the simplest explanation is that Msn2/4 directly regulate Dot6: the proteins bind the DOT6 promoter in multiple ChIP-studies including during NaCl stress and in our own past work (now Fig EV5), are required to maintain DOT6 mRNA levels (Fig 6), are important to maintain Dot6 protein levels (Fig 4), and are required for normal repression of Dot6 target genes (Fig 5). However, because we are unable to explicitly test the role of Msn2/4 binding via the STRE element, we reframed the manuscript a bit to de-emphasize this regulatory connection and focus instead on the importance of the repression to the Msn2/4 response. We do not believe that additional ChIP-seq data will expand understanding: most Msn2/4-bound sites are consistent across stresses, and furthermore we show data collected during NaCl as used here (Ni et al, see Fig EV5). We note that the reviewer is incorrect that DOT6 mRNA differences in Fig 6 are not significant – the revised Fig 6 now shows asterisks to indicate timepoints with statistically significant differences in the paired edgeR analysis from our study (FDR < 0.05), and for three different stresses from other studies. Unfortunately, we do not have antibodies against Dot6 protein so we rely on quantifying epitope-tagged Dot6 expressed from the native genomic locus driven by the natural DOT6 promoter.*

6) MSN2/4 mutant showed reduced Dot6 expression/activity (Fig. 4) and de-repression of rESR genes that partially overlap with the Dot6/Tod6 target. These observations seem to contradict the increased post stress growth rate of Msn2/4 mutant (Fig. 2). Does this mean that the cost of iESR induction outweighs the saved cost of rESR repression? This point could be discussed further.

*Yes, this is indeed our point: cells need the Dot6/Tod6 response to accommodate the Msn2/4 transcriptional changes (and perhaps others in the immediate response) during NaCl. Loss of Msn2/4 alleviates the need for that repression, indicated by the faster post-stress growth rate of the *msn2Δmsn4Δ* strain and the quad strain, during NaCl as originally shown and now also after heat shock, low and high pH, and ethanol stress. The *msn2Δmsn4Δ* strain grows faster than the wild-type after stress, even though it does not fully repress rESR genes – this is consistent with our model that the main function of rESR repression is resource reallocation: if those resources are no longer needed, there may be little cost to the milder repression of these genes. We added a clarifying statement to this end on lines 190-193.*

Minor points.

1) In HIGHLIGHTS, the authors make the point "Cells lacking repressors of growth promoting genes have a slower post-stress growth rate". As the authors mention in the text, this is the finding of their previous paper (Bergen et al. eLife 2022) and should not be highlighted in this work.

We apologize that this was not meant to be presented as a new result, but rather a critical part of our integrated model. We removed it from the highlights.

2) Figure 7, according to the authors' conclusion, the dashed line from Msn2/4 to the promoter for ribosome and growth gene should be a negative regulation ("T shape" allow).

We added the repression indicator along with several other updates to the figure, now Figure 8.

3) Page 21, line 18, there may be typo (... , slows growth I the absence....).

Thank you, we corrected the typo.

Referee #2 (Report for Author)

Msn2 and Dot6/Tod6 have been the subject of numerous studies demonstrating their importance in stress adaptation and recovery from quiescence. In this study, the authors demonstrate that induction of genes targeted by Msn2 (iESR) but also the inhibition of growth promoting genes (rESR) by Dot6/Tod6 are key points that determine both the future capacity to grow and to resist to future stress. In support of this conclusion, the authors show that deletion of Msn2 can increase growth rate during recovery at the expense of future stress resistance, and consequently Msn2-deficient cells grow significantly faster than wild-type cells during salt acclimation, but have defects in stress resistance (peroxide resistance). The authors propose that Msn2 is not only required for the expression of key stress-related genes involved in stress adaptation, but also for the repression of growth-promoting genes through direct regulation of Dot6 levels and through indirect/direct action of Msn2.

The observation that Msn2 deleted cells have higher growth rate during recovery after stress, in contrast to Dot6/Tod6 mutants, and a defect in peroxide resistance represents the major interest of this study presented in Figure 1, 2. Furthermore, the fact that both Msn2 and Dot6/Tod6 deletion have no growth phenotype in exponential growth phase avoids misinterpretation bias and helps to support their conclusions.

We are glad this reviewer appreciated the novelty and importance of these results.

The other parts of this work (Figures 3, 4, 5, 6) add no or only a modest conceptual advance without important mechanistic insights and some conclusions are less well supported by the data presented (mainly the role of Msn2 in repressing rESR genes). Importantly, some data are redundant with previously published studies, e.g. the authors already published in a previous study that deletion of Dot6/Tod6 reduces growth rate after stress (although they presented this data as new data in HIGHLIGHTS "Cells lacking repressors of growth-promoting genes have a slower post-stress growth rate"). Moreover, it has already been suggested that Msn2 has an (indirect) effect on growth-promoting genes, for reasons that remain unclear and were not really clarified in this study. As described in more detail below, new experiments testing mutants under different growth/stress conditions need to be included in the first part of this study to improve the interest of the manuscript, as well as a more cautious interpretation of the genome-wide data, which is mainly based on Venn diagram analysis and the criteria chosen by the authors to define Msn2/4 target genes. The model also needs to be modified, as the data in this manuscript never showed that Msn2 regulates the DOT6 paralog, TOD6 (see more details in major concern 4).

We thank the reviewer for their careful reading of the manuscript, and we provide detailed responses below. First, we did not intend for the Highlight of dot6tod6Δ growth effect to represent it as a new result, but rather a critical part of our integrated model – we removed this from the Highlights.

We very much disagree with this reviewer that the remainder of our manuscript is only a modest advance. While several papers, including Elfving et al. 2014 and our own work (see Chasman et al. 2014), noted that msn2Δmsn4Δ cells show altered repression of rESR genes, there had been no evidence that this was anything beyond an indirect response to a perturbed cellular state, as this reviewer mentions. Here, we provide new data including mutant growth rates, single-cell microscopy, and transcriptome responses, integrated into a model of resource allocation: cells mount a costly Msn2/4 response, which is balanced by the Dot6/Tod6 response, to support acquisition of subsequent stress tolerance. The evolutionary rationale for this model can be explained by acquired stress tolerance: cells accept (and manage, in part via Dot6/Tod6 activity) a cost for the Msn2/4 response during the mild stress treatment so as to accelerate acquisition of subsequent stress tolerance. We believe these insights will be of significant interest to the stress biology field, especially considering our updates to the manuscript that address other points of the reviewer with the addition of new experiments, new and deeper analysis of transcriptomic data, and new modeling data to explore conditions required to maintain the costly Msn2/4 response through evolution. Specific responses are outlined below.

Major concerns:

1/ To increase the impact of this study for the wider readership of The EMBO Journal, and because the author uses the general term "yeast stress response" in the title, it would be 1) important to test not only salt stress but also growth recovery after other physiological stress conditions (pH, heat shock, starvation) with the experimental set-up used in Figure 1-2. It would also be important to test 2) single Dot6 and Tod6 deleted cells as well as 3) cells mutated for the Msn2 binding site at the Dot6 promoter for the reasons explained below in points 3-4. It would also be interesting to 4) test other mutant strains such as the Msn26A allele described in Elfving et al, 2014 or deletion of STB3 or XBP1, since these two factors are involved as Dot6/Dot6 in recovery from quiescence and in repression of growth or cell cycle transcriptional program by promoting recruitment of HDAC Rpd3 at specific set of promoters (PMID: 26300265). Interestingly, a previous study has already shown that Msn2 also regulates and binds to the promoter of the repressor XBP1 (Elfving et al, 2014). These simple experiments will be very useful to extend and support the conclusion drawn in this study regarding the interplay between Msn2 and Dot6/Tod6 during the yeast stress response.

We thank the reviewer for these suggestions.

1) We added the response of this suite of mutants to four other stresses, including heat shock, ethanol treatment, basic and acidic pH. In all cases, the msn2Δmsn4Δ strain recovers with faster growth rate, while the dot6/tod6Δ strain responds with slower growth rate. 2) We added a detailed analysis (described more below) to show that Tod6 plays a very minor role under these conditions. First, we added new microscopy data (new Fig EV2) and RNA-seq data (new Fig EV3) that shows that TOD6 plays little role during the NaCl response, at least in our hands: Tod6-GFP shows little movement to the nucleus during NaCl stress, and that movement is independent of MSN2/4; deletion of TOD6 has little impact on RiBi repression during NaCl stress, whereas deletion of DOT6 produces a repression defect; reducing Dot6-GFP protein to

levels seen in *msn2Δmsn4Δ* cells also produces a repression defect even in the presence of TOD6. We have updated the text to clarify that *Tod6* plays little role during NaCl, at least under these conditions. 3) Detailed studies by the Barkai lab show that *Msn2* can bind many promoters, including the DOT6 promoter, through mechanisms beyond direct DNA binding (utilizing dispersed motifs in its intrinsically disordered region, see Brodsky et al. and others). As outlined for Reviewer 1, *Msn2* can still bind the DOT6 promoter without DNA binding; therefore, we are unable to ablate binding through STRE deletion. Therefore, we made two adjustments to the manuscript: first, we engineered a TET-inducible DOT6 strain and show that reducing *Dot6*-GFP levels to that seen in *msn2Δmsn4Δ* cells is enough to produce a repression defect, similar to the defect seen when *MSN2/4* are deleted (and *Dot6* protein is reduced). Second, we shifted focus of the manuscript to the broader points that this reviewer found most interesting – that there is a cost to the *Msn2/4* response that is balanced by the *Dot6* / *rESR* repression, and that *Msn2/4* contributes to that repression in multiple ways. We believe that these changes, in addition to the new modeling section at the end of the paper, address the concerns of this reviewer while increasing focus on the most novel parts of the work. 4) The proposal to test other gene deletions and mutations is an interesting one, but on that we feel is beyond the scope of this work. We have added a much more detailed analysis and description of the regulation of clusters of *Msn2/4* targets (see new Fig 5 and associated discussion), which introduces other transcriptional regulators.

2/ In Figure 2, the effect of the delay in the establishment of peroxide tolerance appears weak and curiously not significant at 20 minutes. Moreover, in the extended view of Figure 1 (panel C), it appears that maximum tolerance is not established in some wild type until 60 minutes, similar to what was described in some older studies and for the double mutant in the main figures 2. Would the authors like to comment on this point? Furthermore, it's clear that the figure in the extended view of Figure 1 (E,F,G,H) does not follow the EMBO guidelines for figures, as it seems that a large number of drop tests from different plates have been fused (mainly in panel H). These data are important in this study and the authors need to show the raw data of the different replicates for these drop test in Expanded View Figure 1-H and also the raw data that allow to make the figure shown in Figure 2.

*The assay we used for acquired stress resistance is somewhat noisy with day-to-day variation, and the earliest measured time point at 20 min has a small effect in the wild-type strain; thus we did not have statistical power to overcome the variance at this earliest time point. This can be seen in the extended figure view. The defects at later time points are clearly reproducible and statistically significant, showing that the *dot6Δtod6Δ* cells have a delay in acquired peroxide tolerance but eventually reach wild-type levels. The supplemental image was meant to show representative data, which requires many individual plates over different time points. We have added intervening white space to the representative figure where images across different plates are displayed. Notably, each mutant is compared to a side-by-side treated and plated wild-type strain, maximizing our statistical power with paired analyses and overcoming day-to-day variation. We added scoring data used to make the figure in the Source Data File.*

3/ At the beginning of the paragraph on page 15, the authors say "A remaining question is if the weaker *Dot6* activation in the *msn2Δmsn4Δ* mutant leads to weaker repression of *Dot6* target genes. To directly address this question, we followed dynamic changes to the transcriptome before and in ten-minute increments after salt stress in the different strains. We first identified 489 genes whose response to salt treatment was altered in the *dot6Δtod6Δ* strain compared to wild-type (FDR < 0.05) in at least two timepoints (see Methods)."

The data presented in this paper only show that Msn2 is a DOT6 regulator and not TOD6. Therefore, to study the importance of Msn2 in regulating Dot6 levels, the authors can't only use the double mutant Dot6/Tod6 for the experiments in Figure 1, 2 and for the transcriptomic analysis in Figure 5, but they need to also test also single DOT6 mutant and a strain mutated for the binding site of Msn2 at the DOT6 promoter. These experiments are essential for understanding the importance of Msn2 on Dot6 level regulation.

*As presented above, we added several new experiments to the manuscript showing that: Tod6 shows little translocation to the nucleus in response to NaCl (Fig EV2A), neither levels of Tod6-GFP (Fig EV2C) nor nuclear translocation (Fig EV2B,D) of Tod6-GFP are affected by MSN2/4 deletion; TOD6 deletion does not lead to a repression defect during NaCl treatment, whereas DOT6 deletion or reduced expression does (Fig EV3); as outlined more below, we redid the transcriptomic analysis and provide much more detailed information. As outlined above, we cannot ablate Msn2/4 binding to the DOT6 promoter by deleting its binding site, because the proteins can bind the promoter without DNA binding (Brodsky et al.) – we instead show that reducing Dot6-GFP protein to levels in *msn2Δmsn4Δ* produces a repression defect, similar to the repression defect seen in *msn2Δmsn4Δ* cells, and we shifted the focus of the manuscript to more completely encompass the multiple direct and indirect mechanisms through which Msn2/4 contribute to resource reallocation needed for their own response. Although we cannot test the effect of the Msn2/4 binding site, we maintain that the most parsimonious explanation includes direct regulation of DOT6 by Msn2/4: Msn2/4 bind to the Dot6 promoter during NaCl and other stresses, the *msn2Δmsn4Δ* mutant has lower DOT6 mRNA after NaCl treatment, the *msn2Δmsn4Δ* mutant has lower Dot6 protein levels, lower Dot6 protein levels lead to a repression defect after NaCl, just as MSN2/4 deletion leads to lower repression of these genes. We have expanded the Discussion to the multiple roles that Msn2/4 may play in regulating rESR repression. Collectively, we believe these changes have adequately addressed the reviewer's concerns.*

4/ Following on from the previous comments, it does not seem correct to show in the final model (Figure 6) that Msn2 regulates both Dot6 and Tod6 for two main reasons. Firstly, there are no data in this study showing that Msn2 regulates Tod6 and secondly, the authors ignore that the major transcriptional regulator of growth promoting genes, Sfp1, has already been shown to directly regulate Tod6 expression (PMID: 30804227). The latter point is particularly interesting as it suggests that the two stress sensitive TFs, Sfp1 and Msn2, regulate rESR and iESR respectively, and the two paralogs Tod6 and Dot6. Such regulation would maintain a pool of Dot6 and Tod6 in a wide range of growth conditions. In addition, it might be interesting to plot the mRNA levels of Dot6 and Tod6 on the same graph in Figure 7.

*We apologize for the confusion raised in the previous figure – we did not intend to state that Msn2/4 directly regulate TOD6, and we now show clear data that they do not. We updated the figure (now Fig 8) to more accurately capture our hypothesis along with new results presented from the revised transcriptomic analysis. We added TOD6 mRNA changes to Fig 6: TOD6 is likely a direct target of DOT6 (see Fig 5 and associated discussion), and thus shows a repression defect in the *msn2Δmsn4Δ* strain.*

5/ In Figure 4, the authors show that Msn2 has an effect on Dot6 levels and localisation, but given that the two paralogs are thought to have similar function, it seems important to measure Tod6 localisation using a similar experimental setup. In addition, Msn2 is unlikely to affect the level of Tod6, unlike Dot6, and for this reason it will be easier to examine changes in Tod6 localisation by microscopy than Dot6 in Msn2-deficient cells.

As stated above we added these experiments and confirm that Msn2/4 are unlikely to directly regulate TOD6.

6/ Finally, I have probably the most important concerns with the analysis and conclusions in Figure 5-6.

My first concern relates to analysis of Msn2 targets. The authors use YeasTRACT, which is a really useful tool, but some large-scale data have to be taken with cautious, as do many databases that pool genome-wide data from large-scale studies of varying quality. Indeed, according to Yeastract, Msn2/4 directly bind to more than 2100 of the 5500 RNAPII promoters, but it seems very unlikely that Msn2/4 bind to more than a third of the coding gene promoters.

In the revised manuscript, we provide a new and more detailed analysis of the transcriptomic data. Briefly, we partitioned the 1306 Msn2/4-affected genes into 10 clusters based on distinctions in their expression patterns. For each group, we analyzed enrichment of genes with direct evidence of upstream Msn2/4 binding (from Yeastract and also our own stringent data in Huebert et al. 2012, see Fig 5 Source Data), direct evidence of upstream binding scored for all transcription factors in Yeastract, and presence of at least one (or >1, in the Source Data file) STRE element or one GATGAG element within now-500bp upstream of each gene. An important point is that, aside of the DOT6 promoter for which we show data (Fig EV5), we focus on statistical enrichment of genes with these elements across the clusters we defined in the new analysis. There is very strong enrichment (or under-enrichment) for these features that cannot be explained by some errors in the Yeastract compilation. Again, the trends and enrichments of Msn2/4 binding are similar when we use our own past ChIP-seq data for which we are confident of robust statistical calls. The revised analysis provides a deeper discussion of the transcriptomic architecture of these genes, which presents refined analysis of the multiple ways that Msn2/4 contribute to rESR repression. We thank the reviewer for the suggestion, which has made this part of the manuscript stronger.

Similarly, by using a very poor stringent cut-off to identify promoters with Msn2 binding site (they select all genes containing a CCCCT STRE 1000bp upstream TSS), they identified more than 2000 genes containing this motif. Furthermore, a recent study shows that the presence of the STRE motif is a poor predictor of Msn2 binding in vivo (PMID: 32553192, PMID: 38109289), as the disordered region associated with Msn2 mainly defines its binding to promoters containing an STRE motif in living cells, suggesting that the presence of the STRE motif does not accurately define Msn2 binding. In this sense, in a study aimed at defining the target of Msn2, it was reported that "while approximately 11,500 STRE sequences are present in yeast genome, with 3160 lying in the promoters of almost 2000 genes, Msn2 fails to bind most of these sites." (Elfvig et al, 2014, PMID: 24598258). In the latter study, Elfvig and colleagues defined a list of 268 targets (including Dot6) using various analyses including RNAPII and Msn2 ChIP-seq in different mutants and nucleosome profiling, which is likely to be more appropriate than the list used in this manuscript.

The reviewer is correct that most genomic sequences matching transcription factor binding sites are not functional sites that are bound in cells. Aside of for DOT6, we did not focus on individual STRE elements or use them to define gene groups. We use statistical tests to show that different gene groups that we defined statistically are enriched or under-enriched with significance over other genes in the genome using the same criteria, thus naturally controlling for the high frequency of this short site in the genome. We did update the analysis to focus on 500 bp upstream of each gene. Notably, trends are also true for genes with 2 or more upstream

STRE elements, which significantly reduces the background. As for DOT6, we cited several studies that showed Msn2/4 binding to the DOT6 promoter, including repeated citations of Elfving et al. 2014.

All that said, an important point to highlight is that among the most interesting results of this analysis are the gene clusters that are statistically significantly under-enriched for upstream STRE binding sites, including the Dot6 targets but also other rESR genes – strikingly, some of these gene groups are heavily enriched for upstream Msn2/4 binding but significantly under-enriched for having upstream STRE elements, which contrasts the induced Msn2/4 clusters. Hence, we propose that there are different regulatory mechanisms for these gene groups, which will require future study to understand mechanistically.

Nevertheless, the authors use these poorly defined categories to analyze their transcriptomic data, mainly by defining subcategories using a Venn diagram, and the conclusion appears confusing. From my point of view, this Venn diagram analysis needs to be repeated with a more precise definition of the promoter and the number of STRE motifs per promoter, knowing that most studies that carefully assess Msn2/4 binding suggest that Msn2/4 binds to roughly 300 RNAPII promoters. Furthermore, if the authors want to show a direct effect on transcription of Msn2, they need to measure RNA synthesis or RNAPII binding and not mRNA accumulation.

The revised analysis is much more robust and reveals interesting new points. We do not believe that Msn2/4 bind only 300 promoters, based on rigorous past work from our own and other labs: Msn2/4 is robustly detected at many more promoters that contain STRE elements and that are dependent on Msn2/4 for induction. It is very well accepted that Msn2/4 are transcriptional regulators that bind upstream promoters, recruit RNA-polymerase, and induce gene expression. Past work from our lab (Chasman et al. 2024 and Nemeč et al. 2019) and others ChIP'ed RNA Polymerase to show that it is recruited to iESR promoters during NaCl stress. There is no evidence presented here or elsewhere to warrant a re-evaluation that these are transcription factors.

7/ I also have major concerns about redundant conclusions between the Kocik et al. manuscript and other older studies.

-The fact that the absence of Dot6/Tod6 leads to slower acclimation to salt has already been shown (Bergen et al. 2022, PMID: 36350693) and can't be presented as new data in HIGHLIGHT.

As stated above, we removed this from the Highlights as it was not intended to present new data but rather an important component of our model. We were clear in our manuscript that this result has been reported by our lab in a previous paper.

-Similarly, it has already been shown in previous studies that deletion of DOT6/TOD6 prevent proper repression of rESR has been shown in numerous studies (PMID: 31395866, PMID: 21730963, PMID: 35310337, among others) and may not be present as a new result: "These results indicate a primary role for Dot6/Tod6 in repression of their RiBi target genes.", this sentence must be accompanied by references to other studies.

The identification of these genes was not meant to be a new result but rather to define genes dependent on Dot6 under our conditions. We updated the text in lines 287-293 to state, "We first defined RiBi targets under our growth conditions, identifying 489 genes whose response to salt treatment was altered in the dot6Δtod6Δ strain compared to wild-type in at least two

timepoints (FDR < 0.05), Fig EV3A, see Methods). Consistent with prior results (Ho et al., 2018; Huber et al., 2011; Kunkel et al., 2019; Kusama et al., 2022; Lippman & Broach, 2009), 82% of the affected genes showed a repression defect, and 82% of those harbored matches to the known Dot6/Tod6 consensus 'GATGAG' within 500 bp upstream ($p=3.3e-58$ hypergeometric test)."

- Finally, Elfving and colleagues (PMID: 24598258), in an elegant study aimed at defining the role of Msn2/4 during stress, already show that "Msn2 promotes both transcriptional activation and transcriptional repression" and propose "We do find that many of the genes repressed upon activation of Msn2 are highly enriched for those involved in ribosome biogenesis. However, few of these genes are bound by Msn2, at least under conditions of nutrient downshift. Rather, we observed that Msn2 activates transcription of DOT1, which encodes a repressor of ribosome biogenesis genes (see Supplementary Tables S1 and S2). Moreover, we find that Msn2 binds to and activates transcription of XBP1, which encodes a repressor of a number of genes required for cell cycle progression. Accordingly, Msn2, while a primary purveyor of the ESR, may indirectly repress the growth-associated genes encompassed in the ESR."

Respectfully, the first quote refers to a handful of the 192 genes identified by Elfving et al. that are repressed by glucose depletion and show upstream Msn2 binding and harbor STRE elements. This is not at issue in our study. Msn2 over-expression used by Elfving et al. is stressful and almost certainly activates the remaining components of the ESR, indirectly due to the resulting stress of MSN2 over-expression. That over-expression of MSN2 is stressful and indirectly regulates rESR genes does not supplant our results. Finally, DOT1 is a regulator of telomere genes, and cell-cycle genes are not a part of our model.

This conclusion is very similar to the one made in the present manuscript (but Elfving and colleagues made a mistake in the name of DOT1 instead of DOT6 (in agreement with this point, DOT1 is not present in their Supplementary Table S1 showing the list of promoters bound by Msn2, in contrast to DOT6).

*With all due respect, the authors did not mention DOT6 and we can find no citations in that paper to any study of DOT6. Furthermore, the work by Elfving et al. mentions nothing about resource allocation or the function of the rESR. Thus, we disagree that our model in this manuscript has been previously presented by Elfving et al. It is also important to note that we repeatedly cited Elfving et al. in the manuscript for their data and conclusions. We added several additional references, and also a statement on lines 324-326 that, "Previous studies reported a repression defect in *msn2Δmsn4Δ* cells, but whether this was merely an indirect consequence of a defective stress response was not clear (Chasman et al., 2014; Elfving et al., 2014).*

In conclusion, the data presented in Figures 5 and 6 mainly use old data and the conclusions are redundant with older studies.

This is not correct: Fig 5 and 6A is all new transcriptomic data generated as part of this manuscript, with deep analysis not previously presented, and new insights about the connection of Msn2/4 to different gene groups that suggest complex regulatory architecture. Furthermore, the new modeling section and our integrative model have not been presented before. These experiments add significantly to the many other new results and insights in this work.

If this paper is accepted for revision, the authors must at least improve the quality of their analysis, try to validate the binding of Msn2 at the Dot6 promoter by ChIP of Msn2 in a strain

mutated for the Msn2 motif at the DOT6 promoter, and more carefully report previous observations already made by other laboratories.

We have addressed these points as outlined above.

Minor comments :

8/ The authors propose that the timing of the delay correlates with delayed Ctt1 protein production in the dot6 Δ tod6 Δ mutant. In a previous study, the authors showed that increasing Ctt1 mRNA levels using a Gal-estradiol-inducible promoter is sufficient to establish peroxide tolerance (PMID: 22851651), but that the absence of Dot6/Tod6 also increases Ctt1 mRNA levels, but not sufficiently to increase total Ctt1 protein levels (Bergen et al. 2022, PMID: 36350693). To confirm their suggestion that the delay in the acquisition of peroxide tolerance is dependent on Ctt1, it may be interesting to use this inducible Gal system to artificially increase Ctt1 mRNA and protein levels. It is also possible that this delay is independent of Ctt1 accumulation.

This is an interesting idea but beyond the scope of this manuscript, since as the reviewer points out we have already shown that artificially inducing CTT1 leads to a coordinate increase in acquired peroxide tolerance.

9/ Perhaps it's my fault and these data are available, but I can't find an Excel table reporting the source data of Figure 5, allowing me to see which gene of all RNAPII-transcribed genes was associated as an Msn2 target, STRE motif, GATGAG motif according to authors.

We provided the source data for the new Fig 5 as well as other figures in the manuscript.

10/ The figure 4 and one part of associated text have minor interest and could be remove in supplemental Figure.

We disagree and consider the single-cell microscopy in Fig 4 to be among the most important results in the paper, showing that cells lacking MSN2/4 have lower Dot6 protein abundance and weaker activation of the regulator. We retain this figure to show the impact of Msn2/4 on Dot6 abundance and activity.

11/ Probably beyond the scope of this study, but it would be interesting for the authors to discuss stufy from the Barkai lab and/or to use the genetic tool developed in Kafri et al, 2016 to study growth rate recovery during adaptation and acquisition of peroxide resistance. Kafri and colleagues use different strains expressing one to ~20 copies of mCherry under the control of a strong promoter to challenge the physiological effect of overexpression of an inert protein or unstable mRNA (cells forced to transcribe unstable DAMP transcripts). According to the model and the result in Figure 2, it would be expected that such overexpression would also delay the acquisition of resistance. Furthermore, these constructions will also help to understand whether the effect is related to transcription and/or translation. Perhaps the author can at least discuss this point.

This is an interesting point and something we have considered. At this point, we will focus on other experimental additions but can add a point about this in the Discussion.

Referee #3 (Report for Author)

This study investigates how yeast cells relocate transcriptional and translational resources among gene groups during environmental stress. The authors explore the interplay between the transcription factors Msn2/Msn4 and Dot6/Tod6, concluding that Msn2/Msn4 are the primary regulators in resource redirection. Beyond activating stress-response genes, Msn2/Msn4 play a pivotal role in repressing growth-related genes via three mechanisms: 1) Regulating the transcription of DOT6/TOD6. 2) Promoting the nuclear localization of Dot6/Tod6. 3) Regulating the transcription of a subset of rESR genes. While the proposed modeling is intriguing, the supporting evidence is insufficient, and many underlying mechanisms remain unclear. Additional experiments are necessary to solidify these conclusions.

We are glad the reviewer found the model intriguing and hope that added experiments will provide additional clarity on the model.

Specific Comments:

1) In Figure 6, the authors show that *msn2Δmsn4Δ* mutants have lower DOT6 mRNA levels compared to wild-type cells. However, time-course data indicate that Msn2/Msn4 are only minor contributors, as both mutant and wild-type cells upregulate DOT6 mRNA to comparable levels post-stress (60 min), albeit with a slight delay in the mutants. The role of Msn2/Msn4 in regulating DOT6 expression is overstated.

*In the revised manuscript we reworked this section to lay out the evidence for direct regulation, but we added that the Msn2/4 impact on Dot6 levels could also be influenced by indirect effects. We maintain that the most parsimonious explanation is that Msn2/4 can directly regulate DOT6: i) Msn2/4 move to the DOT6 promoter during NaCl treatment and other stresses, ii) MSN2/4 are required to maintain DOT6 mRNA after the time when they move to the DOT6 promoter – this is seen most clearly immediately after NaCl stress as shown in Fig 6, iii) *msn2Δmsn4Δ* cells show less Dot6 protein, iv) *msn2Δmsn4Δ* cells show a defect repressing Dot6 targets, v) we now show that reducing Dot6 protein to levels seen in the *msn2Δmsn4Δ* strain produces a defect repressing Dot6 targets after NaCl. As outlined above, we are unable to test the role of the Msn2/4 binding site upstream of DOT6, since the proteins can still bind independent of DNA binding, through their intrinsic disordered domains (Brodsky et al.). To further address concerns of this reviewer, we also refocused the manuscript a bit, with less emphasis on direct regulation of Dot6 and more focus on the multi-faceted manner in which Msn2/4 contribute to rESR repression and the importance of resource reallocation in response to single versus multiple stress exposures.*

2) In Figure 4, the reduction of Dot6 protein levels in *msn2Δmsn4Δ* mutants before stress is insufficiently explained. The authors attribute this to the absence of stochastic bursts of ESR induction, but single-cell data show consistently low Dot6 abundance across nearly all mutant cells before stress (Figure 4A). This observation is difficult to reconcile with stochastic bursts. A potential experiment to address this would involve mutating the Msn2/Msn4 binding elements in the DOT6 promoter and assessing whether Dot6 protein levels remain low under no-stress conditions.

As outlined above, we are unable to easily ablate Msn2/4 binding to the DOT6 promoter, because the factors can still bind via protein interactions outside its DNA binding domain (Brodsky et al.). We added a statement that the effect could also be indirect, e.g. through regulated Dot6 turnover. We note that past work from our lab (Bergen et al. 2022, Gasch et al.

2017) show that most unstressed cells experience bursts of nuclear Msn2/4 that could in theory influence constitutive Dot6 protein levels.

3) The presentation of results in Figure 5 is unclear. For example, the sentence "Two thirds (68%) of these genes contain upstream Dot6/Tod6 binding elements...and 249 genes showed a repression defect..." raises questions: How many genes overlap between these two groups? How many of these genes also exhibit Msn2/Msn4 binding? How many Msn2/Msn4 binding genes lack upstream STRE elements? The data and descriptions should be reorganized to allow clear comparisons of specific features across gene groups.

We completely reworked this analysis and presentation. Briefly, we partitioned the 1306 Msn2/4-affected genes into 10 clusters based on distinctions in their expression patterns. For each group, we analyzed enrichment of genes with direct evidence of upstream Msn2/4 binding (from Yeastract and also our own data in Huebert et al. 2012, in the Source Data file), genes with direct evidence of upstream binding scored for all transcription factors in Yeastract, and presence of at least one (or >1, in the Source Data file) STRE element or one GATGAG element within now-500bp upstream of each gene. This analysis presents finer resolution of gene groups and suggests multiple modes through which Msn2/4 influence rESR gene repression through both direct and indirect mechanisms. As such, we expanded focus of the manuscript to discuss the multifaceted roles of Msn2/4.

4) Numerous paragraphs in the Results section end with "see Discussion," but it is challenging to locate where these data are discussed. It would be more effective to integrate relevant discussions directly within the results paragraphs.

We apologize for the confusion, as we were directing the reader to our integrated model. In most cases, we simply deleted these references since we agree it was confusing to readers.

Minor Comments:

1) Please add page numbers and line numbers to the manuscript for easier reference.

We added page and line numbers to the revised manuscript.

2) Do the authors have any hypotheses regarding how Msn2/Msn4 binding represses rESR genes? Including such speculation would strengthen the interpretation of results.

Our revised analysis suggests additional details on subsets of rESR genes influenced by Msn2/4. Results suggest that Msn2/4 can physically bind many rESR promoters, but these groups are significantly under-enriched for Msn2/4 binding sites. We speculate that Msn2/4 may form alternate contacts with other proteins bound at the promoter. We added a statement to this effect on line 491-494.

3) In the sentence "conversely, ppGpp over-production, leading to stronger... slow growth | the absence of stress...", it should be "in the absence of stress".

Thank you, we corrected the error.

Dr. Audrey P. Gasch
University of Wisconsin-Madison
Laboratory of Genetics
425-g Henry Mall
Madison, WI 53706

2nd Feb 2026

Re: EMBOJ-2024-119732R

The importance of regulated resource reallocation during dynamic environmental shifts in yeast

Dear Audrey,

Thank you for submitting your revised manuscript, and my apologies for the delays in its re-evaluation at this time of the year. We have now heard back from all three original referees, and I am happy to say that all found the study substantially improved and now suitable for publication. Referee 2 retains a few specific/presentational concerns (see below), which I would ask you to incorporate during a final round of minor revision. At this point, please also carefully attend to the following open editorial issues:

- Please adjust the order of the manuscript sections, and also make sure to use the correct section headers: Title page with complete author information, Abstract, Introduction, Results, Discussion, Methods, Data Availability, Acknowledgements, Disclosure and Competing Interests Statement, References, Main Figure Legends, Tables, Expanded Figure Legends.
- Please reduce the number of keywords on the abstract page to five (ideally choosing broad general terms), and make sure to place them right after the abstract.
- Please correct the reference list, making sure that for references with multiple authors, the first up to 10 authors should be listed, followed by 'et al.' after that (please refer to our Guide to Authors for additional information on EMBO J reference format). Please also double-check that each reference is complete with citation year, volume, and page/eLocator numbers - this information is currently missing for several of them; and remove all DOIs except for cases that are preprinted/pre-published and do not yet have any other exact citation information.
- Please rename the Conflict of Interest section into "Disclosure and Competing Interests Statement", in accordance with our updated Guide to Authors (<https://www.embopress.org/competing-interests>)
- In the Data Availability section, please make sure to include hyperlinks to all databases in which data associated with the study have been deposited (suggested wording: "The [structural coordinates | microarray | mass spectrometry] data from this publication have been deposited to the [name of the database] database [URL] and assigned the identifier [accession | permalink | hashtag].").
- Please also upload a Synopsis Image (in JPG format), which would complement the Synopsis Text that you provided, and which might serve as a "visual title" for the synopsis section of your paper. This image could be simply based on Figure 8, however with slightly altered proportions (it would currently be too 'panoramic'). Please make sure that it remains in the modest dimensions of (exactly) 550 pixels wide and 400-600 pixels high.
- Please make sure to individually reference each (main) Figure panel at least once in the text - e.g. Figs 5 and 6 seem to currently only be called out as a whole, this could be fixed e.g. by referencing "Fig 6A-B" instead of "Fig 6".
- Our routine pre-acceptance image checks indicated that one of the yeast colony growth assay strips in Figure EV1 appears to have been duplicated - please check and clarify/correct!
- Regarding Source Data inclusion, please carefully check our detailed guidelines (<https://resource-cms.springernature.com/springer-cms/rest/v1/content/27825826/data/v1>) and fundamentally reorganize the provided files. We require these data organized figure by figure, in separate files or folders (e.g. all the Source data files for figure 1 need to be saved in a single folder and this needs to be zipped and then uploaded as "SD figure 1.zip" file). Furthermore, I noticed that all your Source Data appears to be numerical, while in the Source Data Checklist, you indicated 'Microscopy img' on several occasions (even though no micrographs seem to be present in the figures?) - please double-check & clarify. It is also not clear why SD for Figures 1 and 7 have not been included.
- For "Table S1" and those "Supplemental Datasets" remaining once the Source Data have been reorganized and uploaded as such, please convert all of them into 'Expanded View Datasets' - nomenclature in text and elsewhere: 'Dataset EV1/2/3...' - and

move their legend texts from the main text into the respective XLSX files, into the 'Read Me' tabs that should be renamed as 'Legend' tabs. Please make sure that each of these remaining datasets is referenced from the main text on at least one occasion.

- Finally, during routine pre-acceptance checks, our data editors have raised the following queries regarding figures, data, and legends; I would appreciate if you briefly answered to them in the cover letter of your final submission, and made the requested text modifications with changes/additions highlighted via the "Track changes" option, to facilitate our final checking":

- 1) Please define the annotated p values ****/**/*/* as well as provide the exact p-values for the same in the legend of figure EV1 A-D as appropriate.
- 2) Please note that the exact p values are not provided in the legends of figures 1C, D; 2B, 6B, EV3 C, F
- 3) Please indicate the statistical test used for data analysis in the legends of figures EV1 A-D
- 4) Please note that the box plots need to be defined in terms of minima, maxima, centre, bounds of box and whiskers, and percentile in the legends of figures 3C, 4D, EV2 D, EV3 C, D, F; EV4 A
- 5) Please note that information related to n is missing in the legends of figures 3C, EV1 A-D; EV3 C, D, F; EV4 A

I am therefore returning the manuscript to you for a final round of revision, with the link below for eventual resubmission. Should you have any questions regarding the referee comments or this decision, please do not hesitate to contact me directly.

With kind regards,

Hartmut

*** PLEASE NOTE: All revised manuscript are subject to initial checks for completeness and adherence to our formatting guidelines. Revisions may be returned to the authors and delayed in their editorial re-evaluation if they fail to comply to the following requirements. As a first step please read our guidelines for revised submissions:
<https://link.springer.com/journal/44318/submission-guidelines#cms-Revised-submissions>

1) Every manuscript requires a Data Availability section (even if only stating that no deposited datasets are included). Primary datasets or computer code produced in the current study have to be deposited in appropriate public repositories prior to resubmission, and reviewer access details provided in case that public access is not yet allowed.

4) Each main and each Expanded View (EV) figure should be uploaded as individual production-quality files (preferably in .eps, .tif, .jpg formats). For suggestions on figure preparation/layout, please refer to our Figure Preparation Guidelines:
<https://media.springernature.com/original/springer-cms/rest/v1/content/27825798/data/v1>

6) Please complete our Author Checklist, and make sure that information entered into the checklist is also reflected in the manuscript; the checklist will be available to readers as part of the Review Process File.

8) Please note that supplementary information at EMBO Press has been superseded by the 'Expanded View' for inclusion of additional figures, tables, movies or datasets; with up to five EV Figures being typeset and directly accessible in the HTML version of the article.

9) To facilitate reproducibility and cross-laboratory adoption of methodologies, please structure the Materials & Methods section as outlined in our guide to authors, including a completed Reagents and Tools Table.

10) Digital image enhancement is acceptable practice, as long as it accurately represents the original data and conforms to community standards. If a figure has been subjected to significant electronic manipulation, this must be clearly noted in the figure legend and/or the 'Materials and Methods' section. The editors reserve the right to request original versions of figures and the original images that were used to assemble the figure. Finally, we generally encourage uploading of numerical as well as gel/blot image source data.

In the interest of ensuring the conceptual advance provided by the work, we recommend submitting a revision within 3 months (3rd May 2026). Please discuss the revision progress ahead of this time with the editor if you require more time to complete the revisions. Use the link below to submit your revision:

Link Not Available

Referee #1:

The authors have addressed the reviewer's concerns and revised the manuscript in accordance with all reviewers' suggestions. The revisions, including a more detailed analysis of the transcriptome data, have extended our understanding of the MSN2/4- and Dot6-dependent transcriptional responses under stress conditions. Furthermore, modeling of the competition between wildtype and MSN2/4 mutant cells convincingly explains the evolutionary advantage conferred by the MSN2-mediated response in yeast.

Referee #2:

This revised manuscript includes detailed responses to the many questions posed by the reviewers and addresses my major comments. Overall, the authors have modified the highlights, the title, as well as the final model as requested. In addition, they convincingly defend their model by significantly improving the analysis presented in Figure 5, which was, in my view, a clear weakness of the initial version of the study. They also increase the impact of this work for the broader readership of The EMBO Journal by testing additional stress conditions and by adding a new modeling analysis in the final part of the study.

Overall, I feel that the revised manuscript has been significantly strengthened, and I support its publication.

I have only a few remaining comments related to figure and data presentation that should be addressed before publication:

Figure 2B and Figure 6A: There is no graduation on the x-axis, or at least no dots on the graph (as shown in Extended Figure 6A), which would help the reader to more easily identify the time point after NaCl addition at which a significant defect occurs.

I also have two comments regarding symbols and statistical reporting:

First, unless I have missed it, in Figure 6B I can't find the definition of the "+" symbol, either in the figure itself or in the legend.

More generally, although the authors properly describe the statistical tests used and report the number (n) of independent experiments, they do not indicate the actual P values for each comparison, and the meaning of the symbols (*) varies between panels. For example, in Figure 1D, * indicates $P < 0.01$ and + indicates $P = 0.07$, whereas in Extended Figure 3, * indicates $P < 2 \times 10^{-16}$ and + indicates $P = 0.01-0.03$. In a few panels, reporting the exact P value may not be essential, but in others it would greatly help to qualitatively interpret the data. Moreover, I believe that in The EMBO Journal, authors are generally expected to report exact P values for statistical tests.

Referee #3:

The authors have addressed most of my previous concerns, and the revised manuscript is much improved.